# FRAGMENT-WISE INTERPRETABILITY IN GRAPH NEURAL NETWORKS VIA MOLECULE DECOMPOSITION AND CONTRIBUTION ANALYSIS

## ABSTRACT

Graph neural networks (GNNs) are widely used in the field of predicting molecular properties. However, their black box nature limits their use in critical areas like drug discovery. Moreover, existing explainability methods often fail to reliably quantify the contribution of individual atoms or substructures due to the message-passing dynamics, which entangle local representations with information from the entire graph. As a remedy, we propose SEAL (Substructure Explanation via Attribution Learning), an interpretable GNN that divides the molecular graph into chemically meaningful fragments and limits information flow between them. As a result, contributions of individual substructures reflect the true influence of chemical fragments on prediction. Experiments on both synthetic and real molecular benchmarks demonstrate that SEAL consistently outperforms existing methods and produces explanations that chemists judge to be more intuitive and trustworthy.

## 1 INTRODUCTION

Graph Neural Networks (GNNs) have achieved state-of-the-art performance in molecular property prediction by naturally representing molecules as graphs of atoms and bonds (Wieder et al., 2020). However, their decision-making processes remain opaque, limiting their adoption in applications where interpretability is crucial for scientific discoveries. The lack of interpretability is primarily caused by the message-passing mechanism, which repeatedly exchanges information between nodes (atoms). In each layer, a node aggregates messages from its neighbors, updating its own representation to capture increasingly global molecular context. While this enables the network to comprehend complex molecular interactions, it also entangles information across the graph. As a result, a final embedding of each node reflects not only its own properties but also the cumulative properties of distant atoms, making it difficult to assess the influence of particular substructures on prediction. Moreover, typical global pooling mechanisms further mix information from different nodes, often leading to the oversmoothing problem (Zhang et al., 2023).

To overcome this problem, we introduce **SEAL** (**S**ubstructure **E**xplanation via **A**ttribution **L**earning), a novel interpretable GNN that generates fragment-wise explanations for molecular property prediction. SEAL decomposes molecular graphs into chemically meaningful fragments and quantifies the contribution of each fragment to model predictions through a constrained message-passing architecture that reduces information leakage between fragments. It is achieved by defining two separate sets of parameters: one used for message passing within fragments (intrafragment weights), and another for message passing between different fragments (interfragment weights). By adding a regularization term on the interfragment weights as an additional loss function, we can control the flow of information between fragments depending on the complexity of the task.

Many molecular properties, including solubility, toxicity, and binding affinity, are predominantly determined by the presence and identity of specific functional groups and substructures rather than complex global interactions (von Korff & Sander, 2006; Murcko, 1995). Therefore, decomposing molecular graphs into chemically meaningful fragments aligns abstractions that chemists use to understand molecular behavior (Ponzoni et al., 2023). For instance, in the solubility prediction examples shown in Figure 1, the molecule is divided into several fragments, among which the most polar

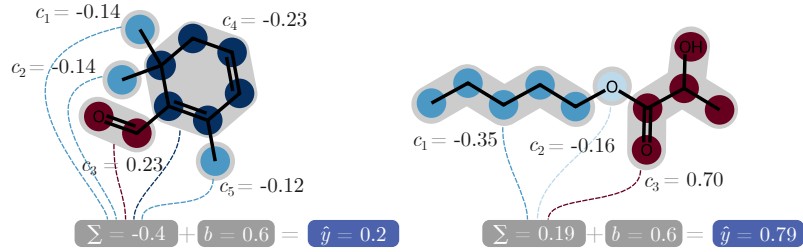

Figure 1: Example explanation generated by SEAL for molecules with low (left side) and high (right side) solubility. In both cases, the molecule is divided into several fragments (marked with gray regions), following our modified BRICS rules described in Section 3.1. The most polar groups contribute positively (red color), and other groups, such as carbon chains, contribute negatively (blue color) to the predicted solubility, which aligns well with the chemical knowledge about factors impacting aqueous solubility (Davis & Leeson, 2023).

groups contribute positively and the hydrophobic groups contribute negatively to the predicted solubility without spreading information to other fragments. These fragment-wise explanations provide more reliable insights than existing GNN explainability methods, which typically assign importance at the level of individual atoms or bonds.

SEAL achieves competitive performance across synthetic and real-world molecular prediction tasks while producing the most accurate explanations among all tested explainers. A user study further confirms that the fragments highlighted by our model are more intuitive to chemists than those returned by other techniques or random assignments. Our contributions can be summarized as follows:

- We propose SEAL, a fragment-based explanation method that decomposes molecular graphs into chemically meaningful substructures, enabling more intuitive insights into model predictions.
- SEAL regularizes inter-fragment message passing to prevent oversmoothing across fragments, preserving local signal and enabling more faithful, fragment-specific contribution estimates.
- SEAL produces explanations preferred by chemists and domain experts while maintaining state-of-the-art predictive performance on molecular property tasks.

## 2 RELATED WORK

**Graph neural networks.** GNNs have become a standard method for analyzing molecular data, often using either a message-passing mechanism (Gilmer et al., 2020) or a transformer-based architecture (Rong et al., 2020; Maziarka et al., 2024). Some of these networks work on fragment graphs where atom groups serve as nodes instead of individual atoms. For example, Cao et al. (2024) proposed a GNN that uses fragment-level message passing for better explainability but still relies on external explainers to determine fragment contributions. Wang et al. (2025) recently introduced FragFormer, a transformer that operates on fragments and employs a variant of the CAM method (Zhou et al., 2016) to explain its predictions. In both models, fragments can contain significant signals coming from other parts of the molecule, potentially reducing local interpretability. In SEAL, we minimize unnecessary message passing between fragments to enhance interpretability.

**Graph-based explainers.** Many explainable AI (XAI) techniques have been proposed to elucidate the predictions of GNNs. Some identify important subgraphs by perturbing the input graph (Ying et al., 2019; Vu & Thai, 2020; Yuan et al., 2021), while other methods analyze the message-passing mechanism in each layer (Feng et al., 2022b; Gui et al., 2023). Often, explaining GNNs is difficult due to the large number of subgraphs and the complex message-passing process. Therefore, Henderson et al. (2021) proposed regularization techniques that disentangle node representations, aiding in generating better explanations. Another approach involves presenting explanations at the fragment level. For instance, Wu et al. (2023) employed BRICS (Degen et al., 2008) to break down

molecules into chemically plausible segments and elucidate predictions by masking entire molecular fragments. In contrast to SEAL, these are post-hoc explanation methods that require additional postprocessing steps to elucidate predictions.

**Interpretable models.** Inherently interpretable methods are also being developed, including prototype-based graph neural networks (Zhang et al., 2022; Rymarczyk et al., 2023) and attention-based models (Xiong et al., 2019; Lee et al., 2023). However, prototypical parts and attention maps on graphs can still be difficult for humans to interpret because of the multitude of explanation patterns that need to be analyzed. (Zhu et al., 2022) introduced HiGNN, a GNN that employs BRICS to generate fragments, forming hierarchical information to enhance predictions. Unfortunately, the fragment information is aggregated using multi-head attention, which complicates interpreting the predictions. To avoid the complexity of prototypical parts or attention maps, SEAL decomposes molecules into a simple sum of scalar fragment contributions.

An extended related work directly comparing features of similar explainers to SEAL is presented in Appendix F.

## 3 SEAL

There are two main differences between the SEAL and the existing GNN models. The first of them, described in Section 3.1, corresponds to the way we aggregate the information from the representation of the atoms. Instead of globally pooling all the representations, we pool them locally within the fragments. The second difference, described in Section 3.2, corresponds to the message passing mechanism, which uses intrafragment and interfragment weights. The latter was regulated with an additional loss.

### 3.1 LOCAL POOLING AND CONTRIBUTION

The interpretability of our model is achieved by redesigning the prediction head in graph-based models. Typically, a readout function in GNNs is used to create a graph-level representation, and then an MLP is applied to make predictions. However, the graph readout aggregates information from all atoms in the graph, hindering the ability to attribute predictions to specific atoms or functional groups.

Our model first aggregates information within graph fragments. We use sum pooling followed by a LayerNorm (Ba et al., 2016) to create the fragment representation from the fragment atom representations. Then, the contribution for each fragment is computed with an MLP, and the final prediction is the sum of all fragment contributions.

Let us define a molecular graph $\mathcal{G} = (\mathcal{V}, \mathcal{E}, X)$, where $\mathcal{V} = \{v_i\}_{i=1}^{N}$ is a set of nodes corresponding to atoms, $\mathcal{E} \subseteq \mathcal{V} \times \mathcal{V}$ is a set of edges corresponding to chemical bonds, $X \in \mathbb{R}^{N \times D}$ is an atom feature matrix, and $D$ is the number of node features. After passing this graph through a sequence of GNN layers, a matrix of atom representations $H \in \mathbb{R}^{N \times M}$ is obtained. Each atom is assigned to exactly one of the $K$ fragments $\mathcal{F}_1, \ldots, \mathcal{F}_K$. Then, the model output is computed as follows:

$$\bar{\mathbf{h}}_i = \sum_{v_j \in \mathcal{F}_i} \mathbf{h}_j, \quad c_i = \mathrm{MLP}\left(\bar{\mathbf{h}}_i\right), \quad \hat{y} = \sum_{i=1}^{K} c_i + b, \tag{1}$$

where $\bar{\mathbf{h}}_i$ is the representation of $i$-th fragment, $c_i$ is the contribution of this fragment, $b$ is a trainable bias term, and $\hat{y}$ is the model prediction. The fragment contributions represent the importance of each fragment. The bias term is essential because every dataset has its own baseline level (arising from constant shifts in the measurement units or structure-independent noise) that cannot be accounted for by a variable number of fragments. Without this bias, the model would need to redistribute this baseline across fragments, reducing the clarity and interpretability of the resulting contributions.

**Fragmentation.** In all experiments, we use a modified BRICS-based fragmentation approach (Degen et al., 2008) inspired by but not identical to Zhang et al. (2021). We isolate side chains attached to rings, even when they contain a single atom. Additionally, unlike their procedure, we separate

non-ring atoms with degree greater than or equal to four (instead of three), and cut non-ring bonds connecting two rings as well as halogen attachments. These adjustments yield fragments that better preserve chemically meaningful units and improve interpretability. Although SEAL can use any fragmentation method as a preprocessing step, we find BRICS most suitable due to its synthesis-inspired rules and strong empirical performance. For completeness, we also evaluate SEALAtom, where each atom forms its own fragment; in this case, the model still produces per-atom additive contributions rather than relying on global average pooling as in standard GNNs.

## 3.2 INTRAFRAGMENT AND INTERFRAGMENT MESSAGE PASSING

The aggregation of messages from neighboring nodes in GNNs is invariant to node permutations. While this mechanism is effective in extracting meaningful information from molecular graphs needed for making correct predictions, the information from each node is easily diffused in the graph, hurting the model's ability to localize crucial atoms and leading to the known problem of oversmoothing.

To mitigate the problem of leaking unnecessary information to neighboring nodes, we propose a new graph neural network variant that operates on pre-fragmented graphs, controlling the information exchanged between fragments. In our implementation, graph layers have separate weights for intrafragment and interfragment edges, $W_{\text{intra}} \in \mathbb{R}^{M' \times M}$ and $W_{\text{inter}} \in \mathbb{R}^{M' \times M}$, respectively. This enables the network to extract relevant information within molecular fragments and block information leaks to neighboring fragments. The new representation of the $i$-th atom $\mathbf{h}'_i \in \mathbb{R}^{M'}$ is computed in the SEAL layer as follows:

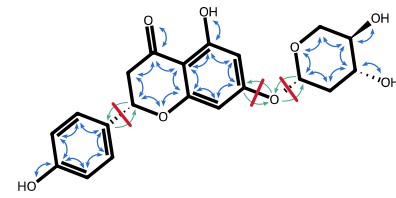

intrafragment weights
interfragment weights
reduced by regularization

$$\mathbf{h}'_i = W\mathbf{h}_i + W_{\text{intra}} \operatorname*{mean}_{j \in \mathcal{N}_{\text{in}}(i)} \mathbf{h}_j + W_{\text{inter}} \operatorname*{mean}_{j \in \mathcal{N}_{\text{out}}(i)} \mathbf{h}_j, \tag{2}$$

where $\mathcal{N}_{\text{in}}(i)$ is a set of neighbors of the $i$-th node within the same fragment, and $\mathcal{N}_{\text{out}}(i)$ is the set of its neighbors outside the fragment. If any of these sets is empty, the corresponding term is removed from the formula.

Figure 2: Message passing in SEAL reduces information exchanged between fragments using different weights for intrafragment (blue arrows) and interfragment (green arrows) edges. The latter are reduced by regularization (red lines).

To avoid leakage of information that is not crucial for prediction, we introduce a regularization term to the loss function, which is the $L_1$ norm of the interfragment weights $W_{\text{inter}}$ (Figure 2). This term is controlled by a hyperparameter $\lambda$ that should be chosen on a case-by-case basis, but typically higher values lead to more interpretable results. The loss function in our model is:

$$\mathcal{L} = \mathcal{L}_{\text{pred}} + \lambda \sum_{l=1}^{L} \left\| W_{\text{inter}}^{(l)} \right\|_1 \tag{3}$$

where $\mathcal{L}_{\text{pred}}$ is the prediction error loss function (mean squared error for regression and cross entropy for classification), $W_{\text{inter}}^{(l)}$ are the interfragment weights in the $l$-th layer.

To balance the trade-off between model performance and interpretability, we perform a 10-fold cross-validation testing multiple values of $\lambda$. The selected model is the one with the highest $\lambda$ values that is not significantly worse than the best model in terms of the target metric (RMSE for regression or AUROC for classification) according to the Wilcoxon signed-rank test.

## 4 RESULTS

In this section, we present the results of experiments conducted on a synthetic benchmark and real-world datasets, as well as the user study. See Appendix A for training details and implementation.

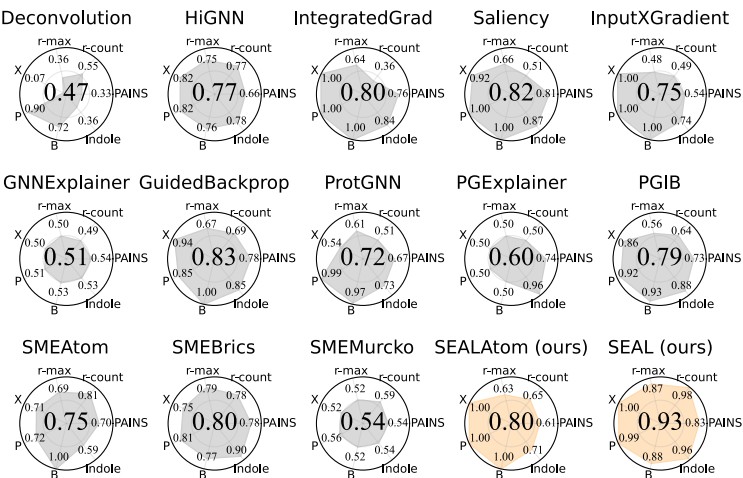

Figure 3: Comparison of explanation quality for the B-XAIC benchmark computed with the Sub-graph Explanation (SE) score for each synthetic dataset (B, P, etc.). SEAL achieves an average score of 0.93, surpassing baseline methods. At the same time, SEAL achieves comparable performance, with an average AUROC of $0.99 \pm 0.01$ (see Appendix B for details).

## 4.1 SYNTHETIC DATASET BENCHMARK

Real-world molecular datasets only offer graph-level labels without assigning importance to specific atoms. Therefore, we chose to first use a synthetic dataset that allows for controlled and reliable assessment of eXplainable AI (XAI) methods by providing direct ground-truth explanations. We evaluate our method on the B-XAIC benchmark (Proszewska et al., 2025), which is designed to compare GNN-based XAI methods in the molecular domain. The dataset includes various tasks focused on identifying different substructures: boron atoms (B), phosphorus atoms (P), halogens (X), indole rings, and pan-assay interference compounds (PAINS). The remaining two tasks focus on counting rings or atoms within rings. Each task has a known ground truth explanation, enabling a precise evaluation of the model's explanation quality.

**Metrics.** To evaluate both model performance and explanation faithfulness, we use a two-part evaluation strategy. For classification, we report standard metrics such as AUROC, F1 Score, and Accuracy. For interpretability of explanations, we use two metrics proposed by Proszewska et al. (2025). **Subgraph Explanations (SE)** is the AUROC evaluating the agreement between model explanation and ground-truth explanation for positive examples. **Null Explanations (NE)** is the percentage of outliers in explained node attributions computed using the interquartile range method for the negative examples.

**Models and baselines.** We benchmark our method against a diverse set of GNN explanation techniques, spanning both mask-based, gradient-based and self-explainable approaches: GNNEx-plainer (Ying et al., 2019), Saliency Maps (Simonyan et al., 2014), InputXGradients (Shrikumar et al., 2016), Integrated Gradients (Sundararajan et al., 2017), Deconvolution (Mahendran & Vedaldi, 2016), (Shrikumar et al., 2016), Guided Backpropagation (Springenberg et al., 2014), PGExplainer (Luo et al., 2020), HiGNN (Zhu et al., 2022), ProtGNN (Zhang et al., 2022), PGIB (Seo et al., 2023) and SME (Wu et al., 2023) denoted as SMEAtom, SMEBrics and SMEMurcko in dependence of fragmentation used in explanation.

The evaluation of model performance is conducted for GCN (Kipf, 2016), GAT (Veličković et al., 2017), GIN (Xu et al., 2018), ProtGNN (Zhang et al., 2022), HiGNN (Zhu et al., 2022), PGIB (Seo et al., 2023) and SME (Wu et al., 2023). Explanation results for post-hoc gradient methods are reported only for the GIN model, as it demonstrates the strongest performance across tasks. Similarly, in ProtGNN and PGIB, we used GIN as the backbone and Saliency as the explainer. Hyperparameters for all models, including SEAL, were optimized through random search. The search space and the optimal hyperparameters found are listed in Appendix A.

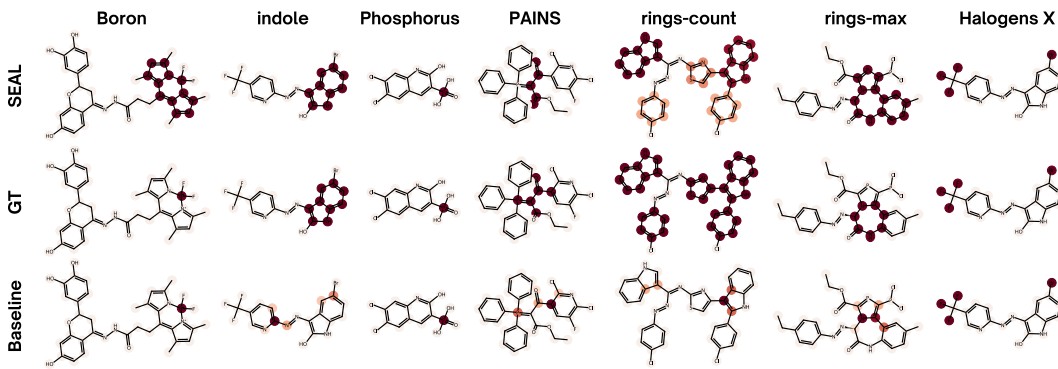

Figure 4: Node-level explanation examples for selected synthetic compounds from the B-XAIC dataset. Each column corresponds to a compound drawn from one of the tasks. The rows (from top to bottom) correspond to the SEAL explanation, the ground-truth explanation, and the explanation generated by the best Baseline (according to SE score). The more intense the red color, the greater the contribution of a substructure or atom. For clarity, the gray regions indicating specific fragments were omitted.

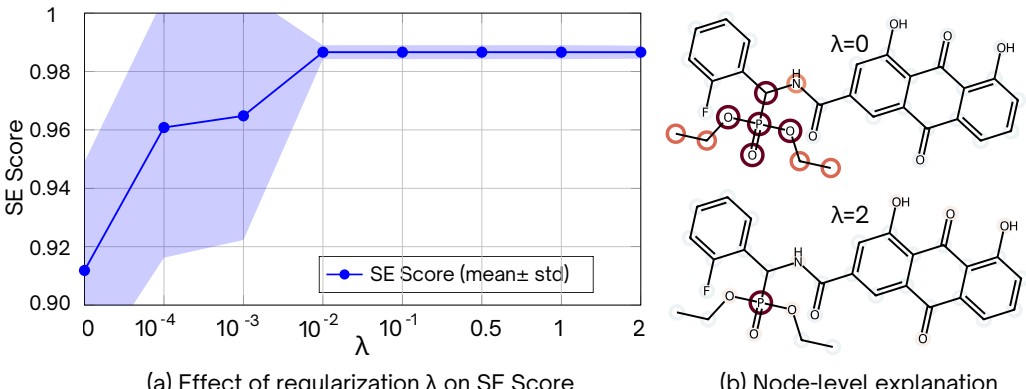

(a) Effect of regularization λ on SE Score  (b) Node-level explanation

Figure 5: Effect of regularization on explanation quality in the phosphorus detection task. **(a)** Plot showing the relationship between $\lambda$ and SE. **(b)** Visual comparison of explanations generated with two different values of $\lambda$. High $\lambda$ values prevent the attribution of high contribution to neighboring fragments.

**Results.** AUROC for all baseline models analyzed equals $0.95 \pm 0.07$, where the maximum baseline score is $0.98 \pm 0.02$ achieved by GIN, and the minimum is $0.88 \pm 0.1$. SEAL achieves $0.99 \pm 0.01$ AUROC, as presented in Table 11 of the Appendix. While achieving competitive classification performance, SEAL adds the capability of explaining its predictions. Figure 3 shows the results of the explanation evaluation, where our method yields significantly higher SE scores than other explainers on challenging tasks such as PAINS, rings-count, indole, and rings-max. In the halogens (X) and phosphorus (P) tasks, our performance is on par with that of the strongest baselines, reflecting the relative ease of localizing single-atom substructures. A slight decline in performance appears in the boron (B) task due to its frequent appearance in complex substructures that our extended BRICS decomposition cannot efficiently segment (see Figure 4). The largest ring pattern, similar to boron, predominantly occurs in larger substructures, but it also presents an additional challenge due to its highly imbalanced nature, with a low percentage of positive samples across the dataset. This limitation does not occur in SEALAtom, which focuses on a single atom. Performance of SEALAtom is consistently strong in the single-atom detection task. However, it faces challenges in more complex tasks. Nevertheless, the overall performance remains comparable to that of alternative explainers. Finally, full evaluation of the positive and negative examples is provided in Table 14 and Table 15 of the Appendix B.

Figure 4 presents example explanations generated by our model for randomly selected molecules. It includes both correct explanations and failure cases where larger fragments (than the ground-truth label) are highlighted. The figure also compares the ground-truth annotations with the outputs of the best-performing baseline method (according to the SE score) selected separately for each task. SEAL effectively highlights chemically meaningful subgraphs, whereas other approaches tend to assign the prediction to only a few atoms, distributing smaller weights across the entire graph. More examples can be found in Appendix E.

**Regularization** SEAL dynamically adapts its $\lambda$ parameter to maximize interpretability without sacrificing performance. We perform an ablation study by varying the regularization parameter $\lambda$, which determines how much message passing is restricted in our model. We discover that the optimal value of $\lambda$ depends on the specific task. In some cases, limiting message propagation improves explanation by preventing information from leaking across irrelevant parts of the graph. For example, in the phosphorus (P) task, increasing $\lambda$ leads to a notable improvement in subgraph explanation quality, as shown in Figure 5. This indicates that stronger regularization helps the model concentrate on localized substructures without causing over-smoothing. In contrast, other tasks, such as PAINS detection, require the information to flow across distant parts of the graph. In these cases, we find that the best explanation performance occurs when $\lambda = 0$. Notably, low $\lambda$ values cause information leakage into adjacent fragments, whereas higher $\lambda$ values provide more focused and faithful explanations.

## 4.2 EVALUATION ON REAL-WORLD DATASETS

Evaluating explanation performance on real-world molecular datasets remains a challenging task. Unlike synthetic benchmarks, these datasets generally do not provide ground-truth explanations that identify which atoms or substructures are responsible for the prediction. Additionally, most molecular properties relevant to real-world applications are significantly more complex, often involving long-range interactions between fragments or features based on the spatial distribution of atoms. To benchmark our method with real-world compounds, we follow the same setup as used for the synthetic dataset.

**Datasets.** We evaluate our method on four real-world molecular property prediction datasets. While three are standard datasets from TDC (Huang et al., 2021), we also include MUTAG (Kazius et al., 2005), which serves as the sole real-world dataset with available ground truth explanations (-NO2 and -NH2 chemical groups contribute to mutagenic property). MUTAG is a binary classification for identifying if a molecule is mutagenic or not. hERG inhibition (Karim et al., 2021) is a binary classification task that includes molecular structures labeled as hERG blockers or non-blockers, a property critical for cardiac safety assessment in drug development. CYP450 2C9 inhibition (Veith et al., 2009) is a binary classification task that focuses on the inhibition of the cytochrome P450 2C9 enzyme, which is central to drug metabolism. Aqueous Solubility (AqSol) (Sorkun et al., 2019) is a regression task that contains compounds with measured solubility in water.

**Metrics.** To evaluate explanations in the absence of ground truth annotations across different methods and fairly compare them with our model, we decided to evaluate on standard positive and negative fidelity. For all models, we mask node features at the input level, ensuring a fair comparison. **Positive Fidelity** is defined as the prediction change after masking the most important nodes indicated by the explainer, and **Negative Fidelity** is the prediction change after retaining only the most important nodes and masking everything else. For MUTAG, we use the same SE metric that is also used in our synthetic tasks and in other XAI works testing on MUTAG (Bui et al., 2024).

For classification tasks, fidelity is measured by the proportion of times the predicted class changes after masking. We evaluate masking at thresholds of 10%, 20%, and 30% of nodes, ensuring that the most relevant atoms are included in explanations without exceeding the specified percentage. However, our method operates on fragments, and it is impossible to select exactly 10% of the atoms of the molecule. Therefore, we select the percentage of atoms in the most relevant fragments that is closest to 10% (e.g. 13%) and mask the same amount of most relevant atoms generated by the baseline methods to keep the sparsity budget fixed for fair comparison. The advantage of our model is that the prediction is a sum of contributions, so we can directly mask contributions instead of

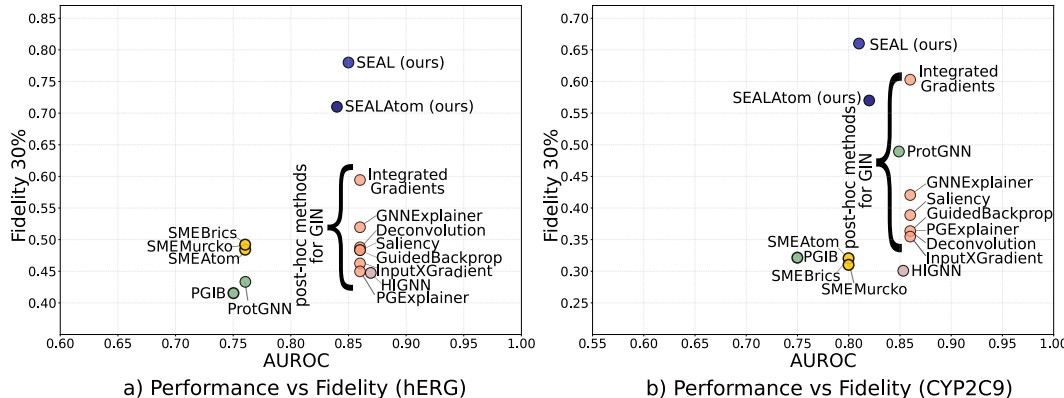

Figure 6: Relationship between explanation quality (Positive Fidelity 30% of masking) and performance (AUROC) for various models on real-world molecular datasets (hERG and CYP2C9). SEAL outperforms other methods in terms of explanation quality, while maintaining a strong performance comparable to that of HiGNN and GIN models. Detailed results, presented in Appendix B, confirm that high explanation quality in SEAL does not come at the cost of performance.

Table 1: Results of model explanations on the real-world MUTAG dataset and corresponding prediction performance. Explanations are evaluated using Subgraph Explanation (SE) and Null Explanation (NE) metrics. Performance of the model is measured by AUROC, F1, and Accuracy. Note that all post-hoc gradient explanations are derived from a shared GIN backbone, and the SME models utilize an identical backbone architecture for generating explanations.

| | | MUTAG | | | |
|---|---|---|---|---|---|
| Model | SE ↑ | ‖ | AUROC ↑ | F1 ↑ | Accuracy ↑ |
| Deconvolution | $0.84 \pm 0.01$ | ‖ | $\mathbf{0.87} \pm 0.01$ | $0.81 \pm 0.00$ | $\mathbf{0.81} \pm 0.01$ |
| GuidedBackprop | $0.38 \pm 0.11$ | ‖ | $\mathbf{0.87} \pm 0.01$ | $0.81 \pm 0.00$ | $\mathbf{0.81} \pm 0.01$ |
| IntegratedGradients | $0.56 \pm 0.18$ | ‖ | $\mathbf{0.87} \pm 0.01$ | $0.81 \pm 0.00$ | $\mathbf{0.81} \pm 0.01$ |
| Saliency | $0.48 \pm 0.07$ | ‖ | $\mathbf{0.87} \pm 0.01$ | $0.81 \pm 0.00$ | $\mathbf{0.81} \pm 0.01$ |
| InputXGradient | $0.45 \pm 0.06$ | ‖ | $\mathbf{0.87} \pm 0.01$ | $0.81 \pm 0.00$ | $\mathbf{0.81} \pm 0.01$ |
| PGExplainer | $0.29 \pm 0.08$ | ‖ | $\mathbf{0.87} \pm 0.01$ | $0.81 \pm 0.00$ | $\mathbf{0.81} \pm 0.01$ |
| GNNExplainer | $0.48 \pm 0.03$ | ‖ | $\mathbf{0.87} \pm 0.01$ | $0.81 \pm 0.00$ | $\mathbf{0.81} \pm 0.01$ |
| HiGNN | $0.55 \pm 0.00$ | ‖ | $\mathbf{0.87} \pm 0.01$ | $0.80 \pm 0.02$ | $0.80 \pm 0.02$ |
| SMEAtom | $0.76 \pm 0.04$ | ‖ | $0.81 \pm 0.01$ | $\mathbf{0.81} \pm 0.01$ | $\mathbf{0.81} \pm 0.01$ |
| SMEBrics | $0.55 \pm 0.00$ | ‖ | $0.81 \pm 0.01$ | $\mathbf{0.81} \pm 0.01$ | $\mathbf{0.81} \pm 0.01$ |
| SMEMurcko | $0.47 \pm 0.03$ | ‖ | $0.81 \pm 0.01$ | $\mathbf{0.81} \pm 0.01$ | $\mathbf{0.81} \pm 0.01$ |
| PGIB | $0.46 \pm 0.05$ | ‖ | $0.50 \pm 0.02$ | $0.75 \pm 0.03$ | $0.75 \pm 0.03$ |
| ProtGNN | $0.47 \pm 0.06$ | ‖ | $0.86 \pm 0.01$ | $0.78 \pm 0.01$ | $0.78 \pm 0.01$ |
| SEALAtom (ours) | $0.71 \pm 0.05$ | ‖ | $0.79 \pm 0.02$ | $0.73 \pm 0.03$ | $0.73 \pm 0.02$ |
| SEAL (ours) | $\mathbf{0.88} \pm 0.01$ | ‖ | $0.85 \pm 0.01$ | $0.80 \pm 0.02$ | $0.79 \pm 0.01$ |

masking the input graph nodes and features (which usually leads to out-of-distribution samples). An ablation study on various masking strategies in SEAL is presented in Appendix C.

**Results.** Figure 6 shows the relationship between predictive AUROC and the quality of explanations measured by positive fidelity on real-world datasets (hERG, CYP2C9). Our SEAL models achieve AUROC values very close to the best-performing baselines, while outperforming other methods in terms of explanation quality. This shows that our method achieves predictive performance on par with the strongest baselines while offering much more quality in explanation. All results on different metrics and methods are indicated in Appendix B in which we also report the comparison of Scaffold instead of Random splits for Fidelity measurement using masking thresholds of 10%-70%. Table 1 presents a consistent trend, where our model on the MUTAG dataset again outperforms competing approaches in explanation quality. These results are particularly signifi-

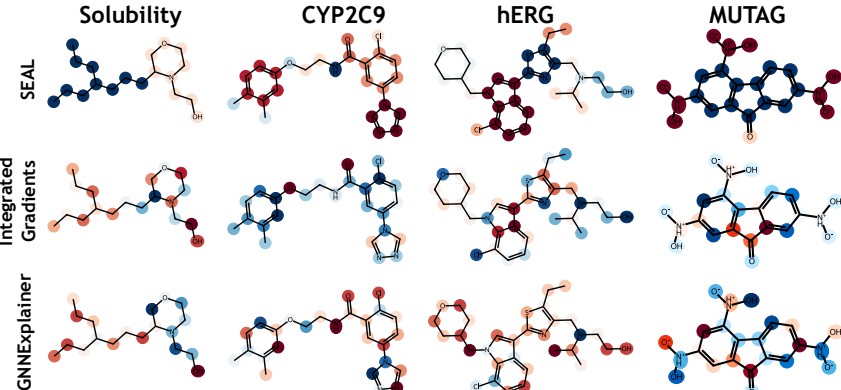

Figure 7: Node-level explanation examples for selected compounds from the Aqueous Solubility, CYP 2C9, hERG and MUTAG datasets. Each column corresponds to a compound from one of the datasets. The rows (from top to bottom) correspond to explanations of SEAL, a gradient-based method (Integrated Gradients), and a perturbation-based method (GNNExplainer). The more intense the color, the greater the contribution (red - positive, blue - negative) of a substructure or atom. SEAL highlights entire substructures with a single color, which corresponds to how chemists analyze molecules in terms of their properties. Only SEAL was able to find -NO2 groups in MUTAG example.

cant as they demonstrate that our framework is not limited to synthetic datasets but also generalizes effectively to the complexities of real-world molecular graphs.

For regression tasks like Solubility, evaluating explanation quality is more difficult, and not all explainers are well-defined in this context. Nevertheless, our method attains reasonable fidelity values compared to other explanation methods. These results are detailed in Appendix B.

**Qualitative examples.** In Figure 7, we present qualitative visualizations of explanations generated by our model compared to the top-performing baselines for the AqSolDB, CYP2C9, hERG MUTAG datasets. While other methods tend to produce scattered or noisy explanations, our model yields more compact and interpretable substructures. These results show that our approach captures chemically plausible explanations that are easier to interpret and often more localized, especially in tasks like solubility, where polarity and solubility driving fragments are correctly emphasized. More examples can be found in Appendix E.

**Discussion.** Across all evaluated tasks, our model consistently demonstrates strong performance, both in terms of the prediction performance and explanation faithfulness, while providing an added benefit of interpretability. We got strong and comparative results compared to the GNN baselines. Furthermore, we also outperform the other explainer techniques, in terms of positive and negative fidelity. Moreover, because SEAL is inherently interpretable, it does not require extensive computing or memory resources, as confirmed in the Complexity Analysis in Appendix G.

By combining strong quantitative results with interpretability aligned with chemical intuition, SEAL proves to be a reliable tool for understanding model decisions across both real and synthetic molecular data. However, fidelity is not a perfect metric because it compares model predictions for the real molecule and its masked counterpart, which has some nodes or their features removed. This artificial reference point is an out-of-distribution sample for the model, so its prediction should be approached with caution. To further support these findings and assess the practical usefulness of the explanations, also keeping in mind that fidelity is not the most informative metric, we conducted a follow-up user study with expert chemists. This enables us to determine whether the generated explanations are not only mathematically accurate but also chemically meaningful and trustworthy in real-world applications.

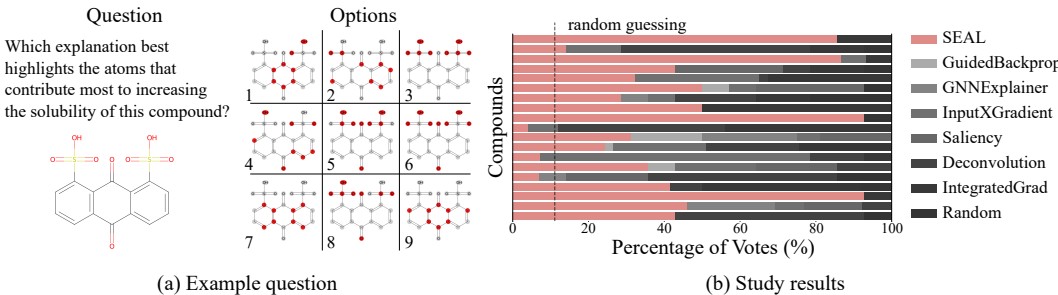

(a) Example question          (b) Study results

Figure 8: User study on the quality of explanations. (a) One example question out of 19 questions in the survey. (b) Distribution of votes per explanation method across all 19 questions. Each bar represents a compound divided between preferred methods (marked with different colors). SEAL produced explanations that chemists preferred the most in 14 out of 19 questions.

### 4.3 USER STUDY

To test whether the explanations produced by SEAL are intuitive to domain experts, we conducted a user study comprising 19 questions that featured various randomly selected compounds. The task for the participants was to indicate the explanation that highlights the atoms contributing most positively to the molecule's solubility. Each question included nine different explanations: one generated by SEAL, six from other explainers, and two random controls, presented in a random order. One control sampled atoms at random, and the other control contained random BRICS fragments to assess whether the preference is based solely on the selection of functional groups familiar to chemists. All presented explanations contained approximately half of the molecule's atoms. Figure 8a shows an example question from the survey. All 14 participants were experts with a minimum of a master's degree in chemistry. They were blinded to the name of the explanation technique, so that their answer was based only on the atoms selected by each method.

SEAL was chosen more often than other explanations in 14 of 19 questions, significantly outperforming all other methods. For the remaining questions, the following methods were chosen most often: Deconvolution and IntegratedGradients (for 5 questions), and InputXGradients (for 3 questions, with possible ties for first place). Other methods (Saliency, GNNExplainer, and Guided Backprop) did not win in any of the questions. The distribution of votes between methods in each question is shown in Figure 8b. All compounds and visualizations that were used for this user study are listed in the Appendix D. The user study confirms that our method, SEAL, provides explanations that align more closely with human intuition and chemical understanding. It was favored over other techniques, emphasizing its ability to produce meaningful and understandable atom-level attributions.

## 5 CONCLUSIONS

In this work, we introduce SEAL, a new approach to GNNs for predicting molecular properties that shifts the focus from atoms and bonds to chemically meaningful fragments. By explicitly controlling the passing of messages within and between fragments, SEAL prevents the leakage of unnecessary information and provides explanations that more closely align with how chemists reason about molecules. Experiments on synthetic and real-world datasets demonstrate that SEAL maintains competitive predictive accuracy and delivers more faithful, intuitive, fragment-level interpretations. A user study further shows that chemists consistently find explanations of SEAL more useful than those of existing methods. Thus, SEAL provides a practical approach to enhancing interpretability in molecular modeling without compromising predictive performance.

### REPRODUCIBILITY STATEMENT

The implementation of our model and the code for reproducing experiments can be found in the supplementary material. The code will be publicly available under an MIT license upon the publication of the paper. All experiments were conducted on an NVIDIA Grace Hopper GH200, NVIDIA Grace CPU 72-Core @ 3.1 GHz, 16GB RAM, CUDA toolkit 12.4. Our experiments were carried

out in Python 3.11, with Pytorch 2.5.1, Pytorch Geometric 2.6.1 for training, and RDKit (2024.9.6) for preprocessing molecules. The full Python environment is available in the code repository.

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

# A  TRAINING DETAILS

## A.1  EXPERIMENTAL DETAILS

We trained the networks with a batch size of 256, using the AdamW optimizer, and employed early stopping after 30 epochs. Additionally, a warm-up period was implemented for the first 50 epochs (with 10 epochs for tasks that required fewer epochs, such as atom-specific tasks from the synthetic dataset). For our model, we used 10-fold cross-validation to select the optimal $\lambda$ using the Wilcoxon

signed-rank test. We used MAE and AUROC as target evaluation metrics for hyperparameter searching and the Wilcoxon test. A weight decay of 0.0001 was applied to all models and tasks. Seed was set to 0 during training, while for explanation extraction and evaluation, it was set to 123. All experiment results were obtained using a 5-fold split approach. The B-XAIC benchmark proposed a fixed train-test set, and we followed this recommendation. For the datasets from TDC, we sampled five testing sets using seeds from 0 to 4, following the benchmark recommendation. By default, we report values from the TDC Random Split, unless Scaffold Split is explicitly indicated. The ranges of hyperparameters are shown in Table 2.

Table 2: Hyperparameter search space used during model optimization.

| Hyperparameter | Values |
| --- | --- |
| Hidden dimensions | [64, 128, 256, 512, 1024] |
| GNN layers | [2, 3, 4] |
| Learning rate | [0.001, 0.003, 0.0001, 0.0003] |
| Dropout rate | [0.0, 0.1, 0.2, 0.3, 0.4, 0.5] |
| $\lambda$ | $[2, 1, 0.5, 10^{-1}, 10^{-2}, 10^{-3}, 10^{-4}, 0]$ |

The hyperparameters selected for the synthetic datasets are listed in Table 9, whereas those for the real-world datasets are presented in Table 10.

## A.2 DATA PREPROCESSING

In our experiments, we standardize target values in our regression task (Solubility), but we do not perform any preprocessing in classification tasks. The atom features used for training include one-hot encoded atom types [C, N, O, F, Cl, Br, P, S, B, I, Other]; we do not use any bond features.

## B EXTENDED RESULTS

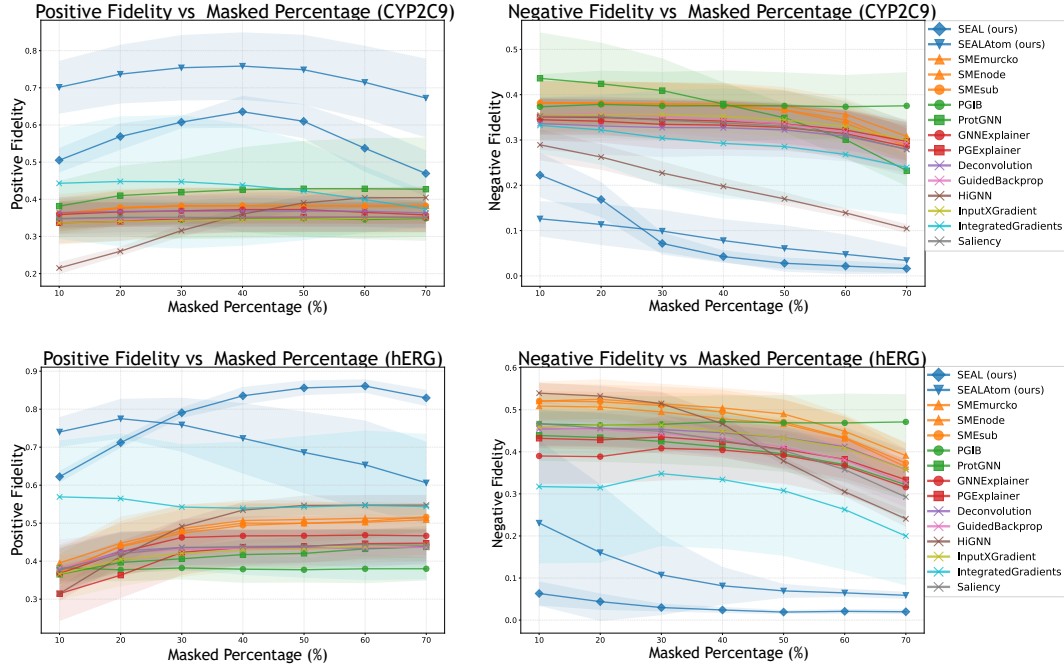

Figure 9: Positive (left column) and Negative (right column) Fidelity scores for CYP2C9 (first row) and hERG (second row) datasets evaluated across different masking percentages (10% to 70%). The results are reported under the Scaffold split and SEAL (ours) models outperform other XAI methods in terms of Positive and Negative fidelity metric at each masking level.

The performance of SEAL with different regularization values $\lambda$ for the synthetic benchmark is presented in Tables 11, Table 12 and Table 13. Detailed results for the subgraph explanation metric are shown in Table 14, and for the null explanation metric in Table 15. For real-world datasets, the evaluation of classification tasks is presented in Table 16, Table 17, while for the regression task in Table 18. For datasets from TDC, we report Random and Scaffold splits. The values of the fidelity metric for these datasets are presented in Table 19, Table 20, Table 21 and Table 22. The comparison of how the Fidelity changes, in dependence of the percentage of molecule masked (from 10% to 70%), we visualize the results in Figure 9.

## C  ABLATION STUDY

### C.1  MASKING STRATEGY

In our fidelity evaluation, we analyze how masking different types of contributions affects the model's interpretability. For each fidelity type (positive, negative), we evaluate the impact of masking the top 10%, 20%, and 30% of nodes or contributions. This allows us to compare how well explanations identify the most influential substructures without exceeding a predefined threshold.

Unlike standard explainers that only operate on node masks, our model allows for masking specific contribution scores directly at the level of the model's architecture by setting $c_i = 0$ for a given fragment. However, a challenge with this approach is that sometimes, even at the beginning of the ranking, a single large important substructure can surpass the 10% node threshold. To fairly compare all the methods, we decide to mask the same amount of atoms for each molecule among the all methods. We need to carefully select the masking strategy: whether to focus on absolute contributions or to selectively mask only positive or negative influences. However, the optimal strategy may vary depending on the task and model sensitivity, whether one chooses to use or omit masking of contributions, and whether masking is guided by absolute, positive only, or negative only importance scores. The results comparing these masking strategies are reported in Table 4 for hERG, in Table 5 for CYP2C9, and in Table 6 for Solubility. These results contain the following naming convention:

- mask-abs: zeroing features, mask contributions - based on maximum absolute value,
- mask: zeroing features, mask contributions - based on maximum or minimum value,
- abs: zeroing features - based on maximum absolute value,
- zero: zeroing features - based on maximum or minimum value.

### C.2  ZERO-INIT

We evaluated different strategies for mitigating information leakage to gather faithful explanations. We compare the proposed dynamic regularization $\lambda$ against a static initialization. Static initialization approach denoted as Zero-Init, initializes the weights $W_{\text{inter}}$ close to zero. Weights are randomly initialized from the normal distribution with a mean of 0 and a standard deviation of $10^{-5}$. We did not apply regularization, which means $\lambda = 0$. We have focused our analysis on the Phosphorus (P) task of the B-XAIC benchmark, as this task exhibits the highest information leakages, as we have observed.

As shown in Table 3, applying the regularization $\lambda$ increases the explanation performance, and the model achieves a 0.99 score on the Subgraph Explanation (SE) metric. In contrast, initialization with the zeros (Zero-Init) does not prevent information leakage. This demonstrates that initialization alone is insufficient to prevent information leakage. Our experiment is supported by the qualitative results in Figure 10, where we show how the contribution values are distributed across the graph, without successfully constraining the information flow for the Zero-Init approach.

### C.3  $\lambda$ CONTRIBUTION

To encourage sparsity in identifying the most relevant substructures, we extended the $\lambda$ constraints. We experimented not only using $\lambda$ as a regularizer for $W_{\text{inter}}$ in message-passing flow, but also

Table 3: Performance of SEAL on SE (Subgraph Explanation) metric for Phosphorus task (P) on B-XAIC benchmark. Standards $\lambda$ parameters denoted as SEAL $\lambda$ and SEAL without regularization, but $W_{\text{inter}} \approx 0$ initialized

| Method | SE |
|---|---|
| SEAL (Zero-Init) | $0.91 \pm 0.04$ |
| SEAL ($\lambda = 2$) | $0.99 \pm 0.00$ |
| SEAL ($\lambda = 1$) | $0.99 \pm 0.00$ |
| SEAL ($\lambda = 0.5$) | $0.99 \pm 0.00$ |
| SEAL ($\lambda = 10^{-1}$) | $0.99 \pm 0.00$ |
| SEAL ($\lambda = 10^{-2}$) | $0.99 \pm 0.00$ |
| SEAL ($\lambda = 10^{-3}$) | $0.96 \pm 0.04$ |
| SEAL ($\lambda = 10^{-4}$) | $0.96 \pm 0.04$ |
| SEAL ($\lambda = 0$) | $0.88 \pm 0.01$ |

Node-level explanation

Figure 10: Effect of regularization on explanation quality in the phosphorus detection task (P). Visual comparison of explanations for $\lambda = 2$ which effectively blocks the information passing and Zero-Init ($W_{\text{inter}} \approx 0$), $\lambda = 0$ where information flow goes from Phosphorus to neighbour atoms.

directly in scalar fragment contributions $c_i$, under these conditions forward pass and loss function are defined as:

$$\bar{\mathbf{h}}_i = \sum_{v_j \in \mathcal{F}_i} \mathbf{h}_j, \quad c_i = \text{MLP}\left(\bar{\mathbf{h}}_i\right), \quad \hat{y} = \sum_{i=1}^{K} c_i + b, \tag{4}$$

where $c_i$ is the scalar contribution to the prediction obtained from the fragment representation, the loss function with extended $\lambda$ constraints is defined as:

$$\mathcal{L} = \mathcal{L}_{\text{pred}} + \lambda_{MP} \sum_{l=1}^{L} \left\| W_{\text{inter}}^{(l)} \right\|_1 + \lambda_{\text{CONTR}} \sum_{i=1}^{K} |c_i| \tag{5}$$

where $\lambda_{\text{MP}}$ controls the flow between fragments as in the original defined loss, and $\lambda_{\text{CONTR}}$ tends to create sparse contributions $c_i$.

To assess the need for this dual $\lambda$ approach, we performed an ablation study for different $\lambda_{\text{MP}}$ and $\lambda_{\text{CONTR}}$ hyperparameter values. The results visualized in Figure 11 compare the performance of the model (AUROC) and explanations (SE) on two tasks from B-XAIC (rings-count, PAINS).

While observing the performance presented in Figure 11, a clear pattern appears: regularization $\lambda_{\text{CONTR}}$ has no influence on the prediction as well as the explanation. As shown in heatmaps, the color gradient of performance changes smoothly and is almost entirely aligned along the $\lambda_{\text{MP}}$ axis. For different $\lambda_{\text{MP}}$ parameters, we see significant changes in Subgraph Explanation (SE) and AUROC

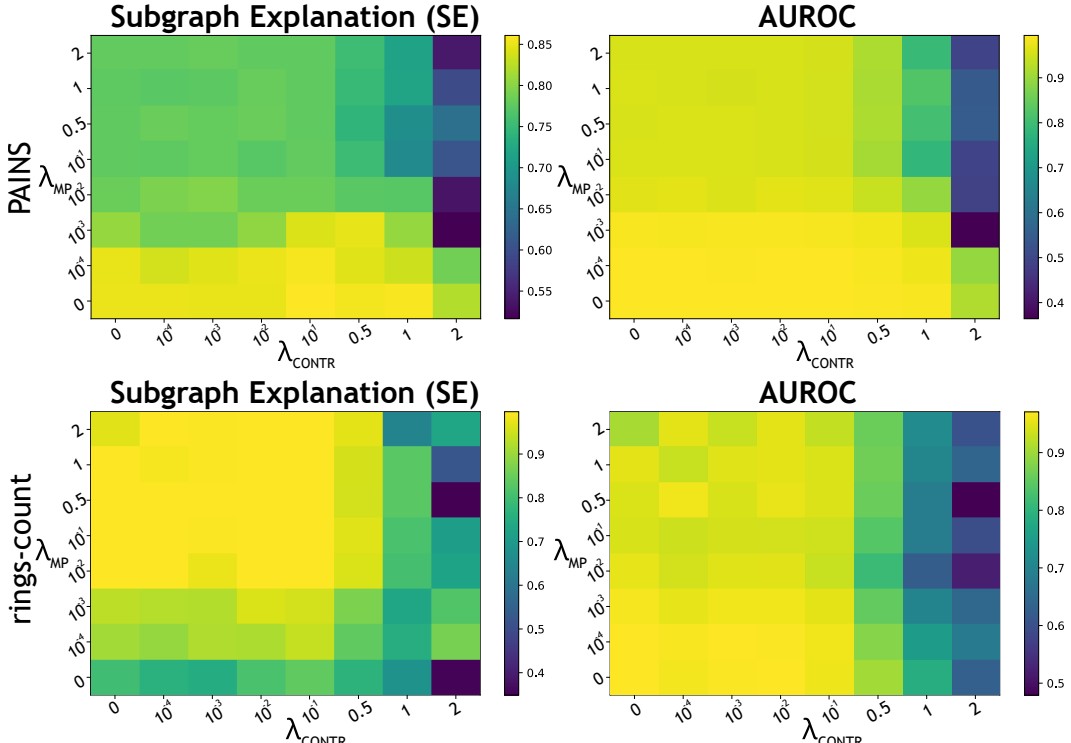

Figure 11: Ablation study on $\lambda$ regularization. The heatmaps display the Subgraph Explanation (SE) and AUROC metric for rings-count and PAINS tasks from B-XAIC, across a grid of $\lambda_{\text{MP}}$ and $\lambda_{\text{CONTR}}$ values. $\lambda_{\text{MP}}$ points to the regularization between fragments on the message passing mechanism, and $\lambda_{\text{CONTR}}$ points to the regularization on the contributions $c_i$.

metrics. Changing $\lambda_{\text{CONTR}}$ makes no difference in the results, worth to note is that, higher $\lambda_{\text{CONTR}}$ makes model collaps what causes underperforming. This observation suggests that regularizing fragment contribution by the $\lambda_{\text{CONTR}}$ is not necessary; the main role is taken by the regularization between fragments by the $\lambda_{\text{MP}}$.

## D    USER STUDY

All of the molecules that were included in our user study are presented in Figures 13-19. Each explanation is annotated with the name of the method that produced this explanation (the names were not included in the survey, and the order of the explanations was randomized). Some methods resulted in the same explanation, which is why some of the figures have multiple method names. In these situations, we had to generate more random explanations to maintain a consistent number of options across questions. Methods that took part in these experiments: SEAL, GuidedBackprop, GNNExplainer, InputXGradient, Saliency, Deconvolution, IntegratedGradients, two Random methods, the first where we sample from nodes, the second where we sample for substructures generated by BRICS.

## E    VISUALIZATIONS

Figures 20, 22, 24, 26, 28, 30, and 32 display examples of explanations generated by the SEAL model for the tasks in the synthetic dataset for the positive target class. The explanations for the negative class, where the substructure is not present in the compound, are illustrated in Figures 21, 23, 25, 27, 29, 31, and 33. The explanations for the real-world datasets are available in Figures 34-37.

Table 4: Model explanations performance using different type of masking strategy in SEAL architecture for hERG dataset. Evaluating using Fidelity metrics at 10%, 20%, and 30% masking thresholds.

| Model | Fidelity$_{10}+\uparrow$ | Fidelity$_{10}-\downarrow$ | Fidelity$_{20}+\uparrow$ | Fidelity$_{20}-\downarrow$ | Fidelity$_{30}+\uparrow$ | Fidelity$_{30}-\downarrow$ |
|---|---|---|---|---|---|---|
| | | | hERG | | | |
| SEAL-mask-abs | $0.36 \pm 0.01$ | $0.18 \pm 0.00$ | $0.37 \pm 0.02$ | $0.18 \pm 0.01$ | $0.38 \pm 0.02$ | $0.15 \pm 0.01$ |
| SEAL-mask | $0.57 \pm 0.01$ | $0.00 \pm 0.00$ | $0.66 \pm 0.01$ | $0.00 \pm 0.00$ | $0.76 \pm 0.01$ | $0.00 \pm 0.00$ |
| SEAL-abs | $0.49 \pm 0.04$ | $0.46 \pm 0.04$ | $0.49 \pm 0.04$ | $0.46 \pm 0.04$ | $0.49 \pm 0.04$ | $0.44 \pm 0.05$ |
| SEAL-zero | $0.59 \pm 0.05$ | $0.45 \pm 0.04$ | $0.58 \pm 0.10$ | $0.43 \pm 0.06$ | $0.58 \pm 0.12$ | $0.40 \pm 0.09$ |
| SEAL-mask-abs | $0.36 \pm 0.01$ | $0.18 \pm 0.01$ | $0.37 \pm 0.02$ | $0.18 \pm 0.01$ | $0.39 \pm 0.02$ | $0.15 \pm 0.01$ |
| SEAL-mask | $0.57 \pm 0.02$ | $0.00 \pm 0.00$ | $0.65 \pm 0.02$ | $0.00 \pm 0.00$ | $0.75 \pm 0.01$ | $0.00 \pm 0.00$ |
| SEAL-abs | $0.54 \pm 0.04$ | $0.48 \pm 0.10$ | $0.55 \pm 0.03$ | $0.47 \pm 0.11$ | $0.55 \pm 0.03$ | $0.45 \pm 0.12$ |
| SEAL-zero | $0.64 \pm 0.03$ | $0.44 \pm 0.15$ | $0.66 \pm 0.06$ | $0.41 \pm 0.18$ | $0.67 \pm 0.10$ | $0.38 \pm 0.20$ |
| SEAL-mask-abs | $0.37 \pm 0.01$ | $0.18 \pm 0.01$ | $0.37 \pm 0.01$ | $0.17 \pm 0.00$ | $0.39 \pm 0.02$ | $0.15 \pm 0.01$ |
| SEAL-mask | $0.57 \pm 0.01$ | $0.00 \pm 0.00$ | $0.66 \pm 0.02$ | $0.00 \pm 0.00$ | $0.76 \pm 0.01$ | $0.00 \pm 0.00$ |
| SEAL-abs | $0.52 \pm 0.04$ | $0.49 \pm 0.05$ | $0.52 \pm 0.04$ | $0.48 \pm 0.06$ | $0.52 \pm 0.04$ | $0.47 \pm 0.07$ |
| SEAL-zero | $0.62 \pm 0.03$ | $0.47 \pm 0.08$ | $0.62 \pm 0.07$ | $0.45 \pm 0.10$ | $0.61 \pm 0.09$ | $0.42 \pm 0.11$ |
| SEAL-mask-abs | $0.37 \pm 0.01$ | $0.20 \pm 0.02$ | $0.38 \pm 0.01$ | $0.19 \pm 0.01$ | $0.40 \pm 0.01$ | $0.16 \pm 0.01$ |
| SEAL-mask | $0.59 \pm 0.01$ | $0.00 \pm 0.00$ | $0.68 \pm 0.01$ | $0.00 \pm 0.00$ | $0.77 \pm 0.01$ | $0.00 \pm 0.00$ |
| SEAL-abs | $0.50 \pm 0.03$ | $0.50 \pm 0.03$ | $0.51 \pm 0.02$ | $0.49 \pm 0.04$ | $0.51 \pm 0.02$ | $0.48 \pm 0.04$ |
| SEAL-zero | $0.59 \pm 0.05$ | $0.49 \pm 0.04$ | $0.60 \pm 0.07$ | $0.48 \pm 0.05$ | $0.58 \pm 0.07$ | $0.45 \pm 0.07$ |
| SEAL-mask-abs | $0.37 \pm 0.01$ | $0.19 \pm 0.02$ | $0.38 \pm 0.01$ | $0.18 \pm 0.01$ | $0.39 \pm 0.01$ | $0.15 \pm 0.01$ |
| SEAL-mask | $0.59 \pm 0.01$ | $0.00 \pm 0.00$ | $0.68 \pm 0.01$ | $0.00 \pm 0.00$ | $0.78 \pm 0.01$ | $0.00 \pm 0.00$ |
| SEAL-abs | $0.47 \pm 0.03$ | $0.48 \pm 0.03$ | $0.48 \pm 0.02$ | $0.47 \pm 0.03$ | $0.49 \pm 0.02$ | $0.46 \pm 0.04$ |
| SEAL-zero | $0.59 \pm 0.03$ | $0.47 \pm 0.05$ | $0.58 \pm 0.07$ | $0.45 \pm 0.07$ | $0.56 \pm 0.09$ | $0.43 \pm 0.09$ |
| SEAL-mask-abs | $0.38 \pm 0.03$ | $0.24 \pm 0.01$ | $0.39 \pm 0.02$ | $0.22 \pm 0.01$ | $0.41 \pm 0.02$ | $0.20 \pm 0.02$ |
| SEAL-mask | $0.63 \pm 0.03$ | $0.02 \pm 0.01$ | $0.72 \pm 0.03$ | $0.01 \pm 0.01$ | $0.80 \pm 0.03$ | $0.01 \pm 0.01$ |
| SEAL-abs | $0.43 \pm 0.04$ | $0.48 \pm 0.01$ | $0.45 \pm 0.04$ | $0.48 \pm 0.01$ | $0.47 \pm 0.03$ | $0.47 \pm 0.02$ |
| SEAL-zero | $0.58 \pm 0.04$ | $0.48 \pm 0.01$ | $0.59 \pm 0.05$ | $0.47 \pm 0.02$ | $0.57 \pm 0.05$ | $0.44 \pm 0.04$ |
| SEAL-mask-abs | $0.42 \pm 0.02$ | $0.28 \pm 0.04$ | $0.44 \pm 0.02$ | $0.28 \pm 0.04$ | $0.46 \pm 0.02$ | $0.27 \pm 0.03$ |
| SEAL-mask | $0.63 \pm 0.01$ | $0.09 \pm 0.03$ | $0.71 \pm 0.01$ | $0.07 \pm 0.02$ | $0.78 \pm 0.01$ | $0.05 \pm 0.02$ |
| SEAL-abs | $0.46 \pm 0.02$ | $0.49 \pm 0.01$ | $0.49 \pm 0.01$ | $0.49 \pm 0.01$ | $0.50 \pm 0.02$ | $0.49 \pm 0.01$ |
| SEAL-zero | $0.56 \pm 0.05$ | $0.49 \pm 0.01$ | $0.57 \pm 0.05$ | $0.48 \pm 0.01$ | $0.55 \pm 0.06$ | $0.47 \pm 0.02$ |
| SEAL-mask-abs | $0.46 \pm 0.02$ | $0.32 \pm 0.02$ | $0.47 \pm 0.01$ | $0.32 \pm 0.02$ | $0.48 \pm 0.01$ | $0.31 \pm 0.02$ |
| SEAL-mask | $0.67 \pm 0.02$ | $0.15 \pm 0.02$ | $0.74 \pm 0.02$ | $0.15 \pm 0.01$ | $0.78 \pm 0.03$ | $0.14 \pm 0.01$ |
| SEAL-abs | $0.43 \pm 0.02$ | $0.49 \pm 0.02$ | $0.48 \pm 0.02$ | $0.49 \pm 0.02$ | $0.49 \pm 0.02$ | $0.48 \pm 0.02$ |
| SEAL-zero | $0.52 \pm 0.02$ | $0.49 \pm 0.02$ | $0.53 \pm 0.02$ | $0.48 \pm 0.02$ | $0.51 \pm 0.02$ | $0.48 \pm 0.02$ |

The row groups correspond to $\lambda = 2$, $\lambda = 1$, $\lambda = 0.5$, $\lambda = 10^{-1}$, $\lambda = 10^{-2}$, $\lambda = 10^{-3}$, $\lambda = 10^{-4}$, and $\lambda = 0$ respectively.

Table 5: Model explanations performance using different type of masking strategy in SEAL architecture for CYP2C9 dataset. Evaluating using Fidelity metrics at 10%, 20%, and 30% masking thresholds.

| | Model | Fidelity$_{10}+\uparrow$ | Fidelity$_{10}-\downarrow$ | Fidelity$_{20}+\uparrow$ | Fidelity$_{20}-\downarrow$ | Fidelity$_{30}+\uparrow$ | Fidelity$_{30}-\downarrow$ |
|---|---|---|---|---|---|---|---|
| | | CYP2C9 | | | | | |
| $\lambda = 2$ | SEAL-mask-abs | $0.36 \pm 0.01$ | $0.19 \pm 0.01$ | $0.37 \pm 0.01$ | $0.18 \pm 0.01$ | $0.39 \pm 0.02$ | $0.15 \pm 0.02$ |
| | SEAL-mask | $0.52 \pm 0.02$ | $0.04 \pm 0.05$ | $0.57 \pm 0.03$ | $0.03 \pm 0.04$ | $0.66 \pm 0.03$ | $0.01 \pm 0.01$ |
| | SEAL-abs | $0.41 \pm 0.11$ | $0.40 \pm 0.11$ | $0.42 \pm 0.11$ | $0.39 \pm 0.11$ | $0.45 \pm 0.10$ | $0.38 \pm 0.12$ |
| | SEAL-zero | $0.49 \pm 0.08$ | $0.37 \pm 0.15$ | $0.51 \pm 0.08$ | $0.35 \pm 0.15$ | $0.54 \pm 0.07$ | $0.32 \pm 0.16$ |
| $\lambda = 1$ | SEAL-mask-abs | $0.34 \pm 0.02$ | $0.20 \pm 0.02$ | $0.35 \pm 0.01$ | $0.19 \pm 0.02$ | $0.36 \pm 0.02$ | $0.16 \pm 0.02$ |
| | SEAL-mask | $0.50 \pm 0.02$ | $0.05 \pm 0.06$ | $0.55 \pm 0.02$ | $0.04 \pm 0.05$ | $0.64 \pm 0.03$ | $0.01 \pm 0.01$ |
| | SEAL-abs | $0.40 \pm 0.13$ | $0.40 \pm 0.12$ | $0.41 \pm 0.13$ | $0.40 \pm 0.12$ | $0.42 \pm 0.13$ | $0.38 \pm 0.12$ |
| | SEAL-zero | $0.45 \pm 0.10$ | $0.39 \pm 0.13$ | $0.46 \pm 0.11$ | $0.38 \pm 0.13$ | $0.47 \pm 0.11$ | $0.35 \pm 0.14$ |
| $\lambda = 0.5$ | SEAL-mask-abs | $0.35 \pm 0.01$ | $0.20 \pm 0.02$ | $0.36 \pm 0.01$ | $0.19 \pm 0.02$ | $0.37 \pm 0.01$ | $0.15 \pm 0.01$ |
| | SEAL-mask | $0.50 \pm 0.02$ | $0.05 \pm 0.04$ | $0.56 \pm 0.01$ | $0.04 \pm 0.03$ | $0.64 \pm 0.02$ | $0.01 \pm 0.01$ |
| | SEAL-abs | $0.35 \pm 0.02$ | $0.36 \pm 0.04$ | $0.37 \pm 0.02$ | $0.35 \pm 0.04$ | $0.40 \pm 0.05$ | $0.33 \pm 0.05$ |
| | SEAL-zero | $0.44 \pm 0.05$ | $0.34 \pm 0.05$ | $0.46 \pm 0.05$ | $0.32 \pm 0.05$ | $0.48 \pm 0.05$ | $0.28 \pm 0.05$ |
| $\lambda = 10^{-1}$ | SEAL-mask-abs | $0.38 \pm 0.02$ | $0.19 \pm 0.02$ | $0.38 \pm 0.02$ | $0.18 \pm 0.02$ | $0.38 \pm 0.02$ | $0.16 \pm 0.01$ |
| | SEAL-mask | $0.57 \pm 0.02$ | $0.00 \pm 0.00$ | $0.63 \pm 0.02$ | $0.00 \pm 0.00$ | $0.72 \pm 0.02$ | $0.00 \pm 0.00$ |
| | SEAL-abs | $0.53 \pm 0.12$ | $0.52 \pm 0.10$ | $0.55 \pm 0.12$ | $0.50 \pm 0.10$ | $0.57 \pm 0.11$ | $0.48 \pm 0.10$ |
| | SEAL-zero | $0.56 \pm 0.08$ | $0.51 \pm 0.11$ | $0.59 \pm 0.08$ | $0.49 \pm 0.10$ | $0.60 \pm 0.08$ | $0.45 \pm 0.12$ |
| $\lambda = 10^{-2}$ | SEAL-mask-abs | $0.35 \pm 0.02$ | $0.20 \pm 0.02$ | $0.36 \pm 0.02$ | $0.20 \pm 0.01$ | $0.36 \pm 0.02$ | $0.18 \pm 0.01$ |
| | SEAL-mask | $0.53 \pm 0.02$ | $0.00 \pm 0.00$ | $0.60 \pm 0.02$ | $0.00 \pm 0.00$ | $0.69 \pm 0.02$ | $0.00 \pm 0.00$ |
| | SEAL-abs | $0.47 \pm 0.14$ | $0.40 \pm 0.08$ | $0.48 \pm 0.15$ | $0.40 \pm 0.08$ | $0.50 \pm 0.15$ | $0.38 \pm 0.08$ |
| | SEAL-zero | $0.50 \pm 0.09$ | $0.39 \pm 0.12$ | $0.52 \pm 0.10$ | $0.38 \pm 0.12$ | $0.53 \pm 0.11$ | $0.36 \pm 0.14$ |
| $\lambda = 10^{-3}$ | SEAL-mask-abs | $0.38 \pm 0.02$ | $0.22 \pm 0.03$ | $0.39 \pm 0.01$ | $0.21 \pm 0.02$ | $0.39 \pm 0.02$ | $0.20 \pm 0.02$ |
| | SEAL-mask | $0.53 \pm 0.02$ | $0.06 \pm 0.04$ | $0.58 \pm 0.02$ | $0.05 \pm 0.04$ | $0.65 \pm 0.02$ | $0.04 \pm 0.03$ |
| | SEAL-abs | $0.41 \pm 0.09$ | $0.38 \pm 0.09$ | $0.42 \pm 0.10$ | $0.37 \pm 0.08$ | $0.43 \pm 0.09$ | $0.36 \pm 0.08$ |
| | SEAL-zero | $0.47 \pm 0.06$ | $0.36 \pm 0.09$ | $0.49 \pm 0.08$ | $0.35 \pm 0.10$ | $0.51 \pm 0.09$ | $0.31 \pm 0.10$ |
| $\lambda = 10^{-4}$ | SEAL-mask-abs | $0.38 \pm 0.03$ | $0.26 \pm 0.04$ | $0.41 \pm 0.04$ | $0.25 \pm 0.03$ | $0.43 \pm 0.04$ | $0.24 \pm 0.03$ |
| | SEAL-mask | $0.52 \pm 0.02$ | $0.13 \pm 0.05$ | $0.58 \pm 0.02$ | $0.11 \pm 0.03$ | $0.63 \pm 0.03$ | $0.10 \pm 0.04$ |
| | SEAL-abs | $0.38 \pm 0.07$ | $0.41 \pm 0.11$ | $0.41 \pm 0.09$ | $0.39 \pm 0.09$ | $0.44 \pm 0.10$ | $0.36 \pm 0.07$ |
| | SEAL-zero | $0.41 \pm 0.05$ | $0.40 \pm 0.13$ | $0.45 \pm 0.06$ | $0.38 \pm 0.12$ | $0.47 \pm 0.07$ | $0.34 \pm 0.11$ |
| $\lambda = 0$ | SEAL-mask-abs | $0.37 \pm 0.01$ | $0.25 \pm 0.02$ | $0.41 \pm 0.02$ | $0.23 \pm 0.02$ | $0.43 \pm 0.03$ | $0.22 \pm 0.01$ |
| | SEAL-mask | $0.52 \pm 0.00$ | $0.11 \pm 0.02$ | $0.59 \pm 0.01$ | $0.09 \pm 0.01$ | $0.65 \pm 0.01$ | $0.07 \pm 0.01$ |
| | SEAL-abs | $0.30 \pm 0.01$ | $0.32 \pm 0.02$ | $0.33 \pm 0.01$ | $0.31 \pm 0.02$ | $0.35 \pm 0.02$ | $0.29 \pm 0.02$ |
| | SEAL-zero | $0.39 \pm 0.02$ | $0.28 \pm 0.02$ | $0.43 \pm 0.02$ | $0.26 \pm 0.02$ | $0.47 \pm 0.03$ | $0.21 \pm 0.02$ |

Table 6: Model explanations performance using different type of masking strategy in SEAL architecture for Solubility dataset. Evaluating using Fidelity metrics at 10%, 20%, and 30% masking thresholds.

| | Model | Fidelity$_{10}+\uparrow$ | Fidelity$_{10}-\downarrow$ | Fidelity$_{20}+\uparrow$ | Fidelity$_{20}-\downarrow$ | Fidelity$_{30}+\uparrow$ | Fidelity$_{30}-\downarrow$ |
|---|---|---|---|---|---|---|---|
| | | Solubility | | | | | |
| $\lambda = 2$ | SEAL-mask-abs | $0.45 \pm 0.03$ | $0.26 \pm 0.10$ | $0.46 \pm 0.03$ | $0.25 \pm 0.10$ | $0.49 \pm 0.04$ | $0.22 \pm 0.09$ |
| | SEAL-mask | $0.42 \pm 0.03$ | $0.30 \pm 0.10$ | $0.45 \pm 0.03$ | $0.29 \pm 0.09$ | $0.49 \pm 0.03$ | $0.27 \pm 0.08$ |
| | SEAL-abs | $1.12 \pm 0.25$ | $1.04 \pm 0.42$ | $1.20 \pm 0.28$ | $0.96 \pm 0.38$ | $1.34 \pm 0.32$ | $0.84 \pm 0.33$ |
| | SEAL-zero | $0.98 \pm 0.23$ | $1.20 \pm 0.47$ | $1.07 \pm 0.26$ | $1.11 \pm 0.42$ | $1.21 \pm 0.30$ | $0.98 \pm 0.37$ |
| $\lambda = 1$ | SEAL-mask-abs | $0.44 \pm 0.04$ | $0.22 \pm 0.04$ | $0.46 \pm 0.04$ | $0.21 \pm 0.04$ | $0.48 \pm 0.05$ | $0.19 \pm 0.04$ |
| | SEAL-mask | $0.41 \pm 0.03$ | $0.27 \pm 0.04$ | $0.44 \pm 0.03$ | $0.27 \pm 0.04$ | $0.47 \pm 0.04$ | $0.24 \pm 0.04$ |
| | SEAL-abs | $1.04 \pm 0.31$ | $0.83 \pm 0.39$ | $1.11 \pm 0.34$ | $0.78 \pm 0.36$ | $1.22 \pm 0.38$ | $0.69 \pm 0.31$ |
| | SEAL-zero | $0.90 \pm 0.30$ | $0.99 \pm 0.42$ | $0.98 \pm 0.33$ | $0.93 \pm 0.38$ | $1.10 \pm 0.37$ | $0.82 \pm 0.33$ |
| $\lambda = 0.5$ | SEAL-mask-abs | $0.44 \pm 0.02$ | $0.22 \pm 0.03$ | $0.46 \pm 0.02$ | $0.21 \pm 0.03$ | $0.48 \pm 0.02$ | $0.19 \pm 0.03$ |
| | SEAL-mask | $0.41 \pm 0.03$ | $0.26 \pm 0.03$ | $0.44 \pm 0.02$ | $0.26 \pm 0.03$ | $0.48 \pm 0.03$ | $0.24 \pm 0.03$ |
| | SEAL-abs | $1.08 \pm 0.31$ | $0.89 \pm 0.41$ | $1.15 \pm 0.33$ | $0.83 \pm 0.38$ | $1.26 \pm 0.37$ | $0.74 \pm 0.34$ |
| | SEAL-zero | $0.91 \pm 0.26$ | $1.08 \pm 0.50$ | $1.01 \pm 0.28$ | $1.00 \pm 0.46$ | $1.13 \pm 0.33$ | $0.89 \pm 0.40$ |
| $\lambda = 10^{-1}$ | SEAL-mask-abs | $0.44 \pm 0.05$ | $0.27 \pm 0.03$ | $0.45 \pm 0.06$ | $0.25 \pm 0.03$ | $0.47 \pm 0.06$ | $0.23 \pm 0.02$ |
| | SEAL-mask | $0.45 \pm 0.04$ | $0.24 \pm 0.04$ | $0.48 \pm 0.04$ | $0.24 \pm 0.03$ | $0.52 \pm 0.04$ | $0.23 \pm 0.02$ |
| | SEAL-abs | $0.77 \pm 0.20$ | $0.67 \pm 0.29$ | $0.83 \pm 0.23$ | $0.62 \pm 0.26$ | $0.92 \pm 0.27$ | $0.55 \pm 0.23$ |
| | SEAL-zero | $0.69 \pm 0.20$ | $0.78 \pm 0.33$ | $0.75 \pm 0.23$ | $0.73 \pm 0.30$ | $0.83 \pm 0.25$ | $0.65 \pm 0.26$ |
| $\lambda = 10^{-2}$ | SEAL-mask-abs | $0.42 \pm 0.02$ | $0.23 \pm 0.03$ | $0.43 \pm 0.02$ | $0.22 \pm 0.03$ | $0.44 \pm 0.02$ | $0.20 \pm 0.03$ |
| | SEAL-mask | $0.42 \pm 0.01$ | $0.21 \pm 0.02$ | $0.45 \pm 0.02$ | $0.21 \pm 0.02$ | $0.48 \pm 0.01$ | $0.21 \pm 0.02$ |
| | SEAL-abs | $0.64 \pm 0.12$ | $0.50 \pm 0.13$ | $0.68 \pm 0.13$ | $0.49 \pm 0.13$ | $0.73 \pm 0.14$ | $0.46 \pm 0.13$ |
| | SEAL-zero | $0.59 \pm 0.11$ | $0.48 \pm 0.11$ | $0.64 \pm 0.12$ | $0.47 \pm 0.11$ | $0.72 \pm 0.14$ | $0.44 \pm 0.11$ |
| $\lambda = 10^{-3}$ | SEAL-mask-abs | $0.48 \pm 0.04$ | $0.29 \pm 0.03$ | $0.51 \pm 0.04$ | $0.25 \pm 0.03$ | $0.55 \pm 0.05$ | $0.22 \pm 0.03$ |
| | SEAL-mask | $0.45 \pm 0.04$ | $0.33 \pm 0.03$ | $0.50 \pm 0.05$ | $0.31 \pm 0.03$ | $0.54 \pm 0.05$ | $0.27 \pm 0.03$ |
| | SEAL-abs | $1.24 \pm 0.20$ | $1.26 \pm 0.33$ | $1.37 \pm 0.21$ | $1.14 \pm 0.30$ | $1.55 \pm 0.25$ | $0.98 \pm 0.25$ |
| | SEAL-zero | $1.10 \pm 0.23$ | $1.43 \pm 0.32$ | $1.23 \pm 0.25$ | $1.29 \pm 0.29$ | $1.40 \pm 0.27$ | $1.11 \pm 0.24$ |
| $\lambda = 10^{-4}$ | SEAL-mask-abs | $0.59 \pm 0.06$ | $0.41 \pm 0.04$ | $0.62 \pm 0.06$ | $0.39 \pm 0.04$ | $0.66 \pm 0.06$ | $0.36 \pm 0.03$ |
| | SEAL-mask | $0.56 \pm 0.06$ | $0.47 \pm 0.04$ | $0.60 \pm 0.06$ | $0.45 \pm 0.04$ | $0.64 \pm 0.07$ | $0.42 \pm 0.03$ |
| | SEAL-abs | $0.78 \pm 0.14$ | $0.64 \pm 0.16$ | $0.84 \pm 0.15$ | $0.58 \pm 0.13$ | $0.91 \pm 0.17$ | $0.52 \pm 0.09$ |
| | SEAL-zero | $0.72 \pm 0.10$ | $0.73 \pm 0.22$ | $0.79 \pm 0.10$ | $0.66 \pm 0.19$ | $0.88 \pm 0.12$ | $0.58 \pm 0.15$ |
| $\lambda = 0$ | SEAL-mask-abs | $0.53 \pm 0.06$ | $0.46 \pm 0.04$ | $0.59 \pm 0.06$ | $0.42 \pm 0.04$ | $0.63 \pm 0.07$ | $0.37 \pm 0.04$ |
| | SEAL-mask | $0.54 \pm 0.05$ | $0.47 \pm 0.04$ | $0.61 \pm 0.05$ | $0.43 \pm 0.04$ | $0.65 \pm 0.06$ | $0.38 \pm 0.04$ |
| | SEAL-abs | $0.49 \pm 0.03$ | $0.55 \pm 0.07$ | $0.54 \pm 0.04$ | $0.53 \pm 0.08$ | $0.58 \pm 0.05$ | $0.48 \pm 0.07$ |
| | SEAL-zero | $0.53 \pm 0.03$ | $0.49 \pm 0.07$ | $0.60 \pm 0.03$ | $0.46 \pm 0.06$ | $0.65 \pm 0.04$ | $0.41 \pm 0.05$ |

# F  EXTENDED RELATED WORK

SEAL differs substantially from existing explainability approaches for graph neural networks. Model-level explanation methods such as XGNN (Yuan et al., 2020) and MAGE (Yu & Gao, 2025) operate by generating synthetic graphs rather than attributing predictions on a specific molecular input, making them unsuitable for fragment-level interpretability. Among self-explainable GNNs, including KerGNN (Feng et al., 2022a), ProtGNN (Zhang et al., 2022), and PGIB (Seo et al., 2023), none provide instance-specific explanations at the level of molecular fragments, as they rely on prototypes or kernels rather than decomposing a prediction across chemically meaningful substructures. The closest work is HiGNN (Zhu et al., 2022), whose hierarchical architecture makes post-hoc inspection easier, but it does not enforce fragment-wise interpretability within the model and therefore remains only partially interpretable. SME (Wu et al., 2023) also uses BRICS-derived fragments but functions purely as a post-hoc masking explainer. Other common explainers such as GNNExplainer (Ying et al., 2019), PGExplainer (Luo et al., 2020), SubgraphX (Yuan et al., 2021), and PGM-Explainer (Vu & Thai, 2020) identify important subgraphs by masking or perturbing the input graph, which alters molecular structures and often produces out-of-distribution graphs. DEGREE (Feng et al., 2022b) is the only post-hoc method that yields a form of prediction decomposition, though it is not fragment-based and still relies on the representations learned by a standard GNN. Methods like FlowX (message-flow analysis, Gui et al. (2023)) and gradient-based techniques operate on the original graph but their signals become unreliable under oversmoothing. Importantly, none of these approaches, including hierarchical or prototype-based models, explicitly address oversmoothing or regulate cross-fragment message mixing, whereas SEAL provides inherent fragment-level interpretability and direct architectural control over information flow. For feature comparison, see Figure 7.

Table 7: Comparison of XAI GNN models with emphasis on SEAL key features. 'frag.' is a fragment-based explanation, 'interp.' is an inherently interpretable model, 'instance' is an instance-level explanation (as opposed to model-level), 'molecular' is an explainer adapted to molecular domain, 'decomp.' is an explainer which decomposes prediction into subgraph contributions, 'control' is an explainer with an oversmoothing control mechanism, and 'original' is an explainer that does not alter the original input graph.

| Model | frag. | interp. | instance | molecular | decomp. | control | original |
|---|---|---|---|---|---|---|---|
| SEAL (ours) | Yes | Yes | Yes | Yes | Yes | Yes | Yes |
| HiGNN | Yes | Partially | Yes | Yes | No | No | Yes |
| ProtGNN | No | Yes | Yes | No | No | No | Yes |
| KerGNN | No | Yes | Yes | No | No | No | Yes |
| PGIB | No | Yes | Yes | No | No | No | No |
| GNNExplainer | No | No | Yes | No | No | No | No |
| PGExplainer | No | No | Yes | No | No | No | No |
| SubgraphX | No | No | Yes | No | No | No | No |
| PGM-Explainer | No | No | Yes | No | No | No | No |
| GraphSHAP | No | No | Yes | No | No | No | No |
| FlowX | No | No | Yes | No | No | No | Partially |
| Gradient Methods | No | No | Yes | No | No | No | Yes |
| DEGREE | No | No | Yes | No | Yes | No | Yes |
| SME | Yes | No | Yes | Yes | No | No | No |
| MotifExplainer | Yes | No | Yes | No | No | No | Partially |
| MAGE | Yes | No | No | No | No | No | No |
| XGNN | No | No | No | No | No | No | No |

# G  COMPLEXITY ANALYSIS

We denote the standard GCN forward pass complexity as: $O(L|E|F + LNF^2)$ where $L$ is layers, $|E|$ is edges, $N$ is nodes, and $F$ is feature dimension. SEAL performs a node update using three components: a root transformation $W$ and aggregations over intra- and inter-fragment neighbors with $W_{\text{intra}}$ and $W_{\text{inter}}$. The time complexity is therefore $O(|E|F + 3NF^2) \approx O(|E|F + NF^2)$,

which is asymptotically equivalent to a standard GCN layer. The memory complexity is $O(3F^2) \approx O(F^2)$, due to the three weight matrices, which dominate the storage requirements, which is also equivalent asymptotically to standard GCN.

Each of the methods listed in Table 8 has an asymptotic forward complexity approximately equivalent to our SEAL layer and a standard GCN $O(|E|F + NF^2)$. Some methods introduce additional mechanisms that increase practical computational costs: for example, HiGNN uses fragment interaction blocks, while prototype-based methods such as ProtGNN and PGIB adopt a technique to resolve the time complexity issue in MCTS. Despite some cases, the core message-passing and linear transformations dominate the time complexity, so the asymptotic cost remains comparable to SEAL and GCN. The comparison of the complexity of the explanation for each method is presented in Table 8, where our SEAL method needs only constant time to obtain the explanation, as it is directly interpretable from the sum of the contributions.

In Figure 12 we present the time measurement of each method, which consists of a single inference run to obtain results and the explanation, averaged over all graphs. The tests were run on a CPU with 64GB of RAM, without a GPU. Experiments were done on the same configurations (hidden dimension equals 256, layers equals 3) for different models. Our model has an increased number of parameters ($405K$) in comparison to GIN ($334K$) and GCN ($136K$). As we expected, SEAL needs only a forward pass, which takes one of the smallest times (2.1 ms), since it is interpretable from the design. Gradient-based methods require an additional backward pass, which increases computational cost. Among self-explainable models, PGIB incorporates optimizations to reduce MCTS overhead. Similarly, for ProtGNN, which offers variants both with and without MCTS, we report results for the faster configuration, but it is worth noting that the version with MCTS was the slowest explanation. Finally, while PGExplainer requires an initial training phase, its inference time is significantly faster than GNNExplainer.

Table 8: Comparison of Explanation Complexity for the Gradients Based methods: Saliency, InputXGradients, GuidedBackprop, Deconvolution, Integrated Gradients. Masked-Based Methods: GNNExplainer, PGExplainer. Self-Explainable: ProtGNN, PGIB. Fragment-Based Methods: SME, HiGNN, SEAL. We use the following notation $S$ Integration steps, $T$ optimization epochs, $W$: Complexity of MLP, $M$: Number of Prototypes, $K$ Number of fragments produced by BRICS.

| Method | Explanation Complexity |
|---|---|
| Saliency | $O(L|E|F + LNF^2)$ |
| InputXGradients | $O(L|E|F + LNF^2)$ |
| GuidedBackprop | $O(L|E|F + LNF^2)$ |
| Deconvolution | $O(L|E|F + LNF^2)$ |
| IntegratedGradients | $O(S(L|E|F + LNF^2))$ |
| GNNExplainer | $O(T(L|E|F + LNF^2))$ |
| PGExplainer | $O(T(L|E|F + LNF^2) + O(T|E|W)$ |
| ProtGNN | $O(MF)$ |
| PGIB | $O(MF)$ |
| SME* | $O(2^K(L|E|F + LNF^2))$ |
| HiGNN | $O(KF)$ |
| **SEAL** | **O(1)** |

*In practice, SME limits the exponential search space ($2^K$) to a fixed number of samples (e.g., 100)

# H   USE OF LLMS

In this study, large language models (LLMs) like Claude Sonnet 4 and ChatGPT 4o were used to rewrite sections of the text. The authors reviewed and verified the generated content.

Table 9: Hyperparameters found for SEAL, SEALAtom, GAT, GCN, GIN, ProtGNN, HiGNN, SME and PGIB in synthetic dataset evaluation.

| Model | B | P | PAINS | X | indole | rings-count | rings-max |
|---|---|---|---|---|---|---|---|
| SEAL | | | | | | | |
| Hidden dimensions | 1024 | 1024 | 512 | 1024 | 512 | 1024 | 256 |
| GNN layers | 4 | 4 | 3 | 2 | 4 | 2 | 4 |
| Learning rate | 0.0001 | 0.003 | 0.003 | 0.003 | 0.0003 | 0.003 | 0.003 |
| Dropout | 0.4 | 0.1 | 0.1 | 0.1 | 0.1 | 0.1 | 0.2 |
| $\lambda$ | 2 | 2 | 0 | 2 | $10^{-4}$ | $10^{-3}$ | 2 |
| SEALAtom | | | | | | | |
| Hidden dimensions | 1024 | 1024 | 256 | 1024 | 256 | 512 | 1024 |
| GNN layers | 4 | 4 | 4 | 2 | 4 | 4 | 4 |
| Learning rate | 0.0001 | 0.003 | 0.003 | 0.003 | 0.003 | 0.0003 | 0.0003 |
| Dropout | 0.4 | 0.1 | 0.2 | 0.1 | 0.2 | 0.1 | 0.1 |
| $\lambda$ | 2 | 2 | 0 | 2 | $10^{-4}$ | $10^{-4}$ | 2 |
| GAT | | | | | | | |
| Hidden dimensions | 256 | 1024 | 256 | 1024 | 256 | 1024 | 256 |
| GNN layers | 3 | 4 | 3 | 4 | 3 | 4 | 3 |
| Learning rate | 0.0003 | 0.0001 | 0.0001 | 0.0001 | 0.0001 | 0.003 | 0.0001 |
| Dropout | 0.4 | 0.4 | 0 | 0.4 | 0 | 0.1 | 0 |
| GCN | | | | | | | |
| Hidden dimensions | 1024 | 1024 | 512 | 1024 | 512 | 512 | 1024 |
| GNN layers | 4 | 4 | 4 | 4 | 4 | 4 | 4 |
| Learning rate | 0.0001 | 0.0001 | 0.0003 | 0.0001 | 0.0003 | 0.0003 | 0.0003 |
| Dropout | 0.4 | 0.4 | 0.1 | 0.4 | 0.1 | 0.1 | 0.1 |
| GIN | | | | | | | |
| Hidden dimensions | 1024 | 1024 | 1024 | 1024 | 512 | 256 | 1024 |
| GNN layers | 4 | 4 | 4 | 4 | 4 | 3 | 4 |
| Learning rate | 0.0001 | 0.0001 | 0.0003 | 0.0001 | 0.0003 | 0.001 | 0.0003 |
| Dropout | 0.4 | 0.4 | 0.1 | 0.4 | 0.1 | 0.5 | 0.1 |
| ProtGNN | | | | | | | |
| Hidden dimensions | 1024 | 1024 | 1024 | 1024 | 512 | 256 | 1024 |
| GNN layers | 4 | 4 | 4 | 4 | 4 | 3 | 4 |
| Learning rate | 0.0001 | 0.0001 | 0.0003 | 0.0001 | 0.0003 | 0.001 | 0.0003 |
| Dropout | 0.4 | 0.4 | 0.1 | 0.4 | 0.1 | 0.5 | 0.1 |
| HiGNN | | | | | | | |
| Hidden dimensions | 128 | 128 | 256 | 128 | 128 | 256 | 128 |
| GNN layers | 4 | 4 | 4 | 4 | 4 | 4 | 4 |
| Learning rate | 0.0003 | 0.0003 | 0.0003 | 0.0003 | 0.0003 | 0.0003 | 0.0003 |
| Dropout | 0.4 | 0.4 | 0.1 | 0.4 | 0.4 | 0.5 | 0.1 |
| SME | | | | | | | |
| Hidden dimensions | 256 | 256 | 256 | 256 | 256 | 128 | 256 |
| GNN layers | 3 | 3 | 4 | 3 | 3 | 2 | 3 |
| Learning rate | 0.0003 | 0.0003 | 0.003 | 0.0003 | 0.0003 | 0.0001 | 0.0001 |
| Dropout | 0.3 | 0.3 | 0.2 | 0.3 | 0.3 | 0.4 | 0.2 |
| PGIB | | | | | | | |
| Hidden dimensions | 256 | 256 | 1024 | 256 | 512 | 1024 | 256 |
| GNN layers | 3 | 3 | 4 | 3 | 4 | 4 | 4 |
| Learning rate | 0.0001 | 0.0001 | 0.0003 | 0.0001 | 0.0001 | 0.003 | 0.0001 |
| Dropout | 0.2 | 0.2 | 0.1 | 0.2 | 0.2 | 0.1 | 0.4 |

Table 10: Hyperparameters found for SEAL, SEALAtom, GAT, GCN, GIN, ProtGNN, HiGNN, SME and PGIB in real-world dataset evaluation.

| Model | CYP | hERG | Solubility | MUTAG |
|---|---|---|---|---|
| **SEALAtom** | | | | |
| Hidden dimensions | 512 | 1024 | 512 | 512 |
| GNN layers | 4 | 4 | 4 | 4 |
| Learning rate | 0.0003 | 0.0003 | 0.0003 | 0.0003 |
| Dropout | 0.1 | 0.1 | 0.1 | 0.1 |
| $\lambda$ | 0 | 0 | 0 | 0 |
| **SEAL** | | | | |
| Hidden dimensions | 512 | 512 | 1024 | 256 |
| GNN layers | 4 | 4 | 4 | 3 |
| Learning rate | 0.0003 | 0.0003 | 0.003 | 0.0001 |
| Dropout | 0.1 | 0.1 | 0.1 | 0.0 |
| $\lambda$ | 2.0 | 0.0001 | 0.0001 | 0.001 |
| **GAT** | | | | |
| Hidden dimensions | 256 | 256 | 128 | 256 |
| GNN layers | 3 | 3 | 3 | 3 |
| Learning rate | 0.0001 | 0.0001 | 0.0003 | 0.0001 |
| Dropout | 0 | 0 | 0.3 | 0.0 |
| **GCN** | | | | |
| Hidden dimensions | 256 | 1024 | 1024 | 1024 |
| GNN layers | 4 | 4 | 4 | 4 |
| Learning rate | 0.003 | 0.003 | 0.003 | 0.0003 |
| Dropout | 0.2 | 0.1 | 0.1 | 0.1 |
| **GIN** | | | | |
| Hidden dimensions | 512 | 512 | 1024 | 256 |
| GNN layers | 4 | 4 | 4 | 3 |
| Learning rate | 0.0003 | 0.0003 | 0.0003 | 0.0001 |
| Dropout | 0.1 | 0.1 | 0.1 | 0.0 |
| **ProtGNN** | | | | |
| Hidden dimensions | 256 | 256 | - | 1024 |
| GNN layers | 4 | 3 | - | 3 |
| Learning rate | 0.003 | 0.003 | - | 0.003 |
| Dropout | 0.2 | 0.5 | - | 0.0 |
| **HiGNN** | | | | |
| Hidden dimensions | 128 | 256 | 512 | 512 |
| GNN layers | 4 | 4 | 4 | 4 |
| Learning rate | 0.003 | 0.0003 | 0.0003 | 0.0003 |
| Dropout | 0.2 | 0.1 | 0.1 | 0.1 |
| **SME** | | | | |
| Hidden dimensions | 256 | 1024 | 1024 | 256 |
| GNN layers | 4 | 4 | 2 | 4 |
| Learning rate | 0.003 | 0.0001 | 0.0003 | 0.003 |
| Dropout | 0.4 | 0.3 | 0.4 | 0.2 |
| **PGIB** | | | | |
| Hidden dimensions | 512 | 512 | - | 1024 |
| GNN layers | 2 | 3 | - | 3 |
| Learning rate | 0.001 | 0.0001 | - | 0.001 |
| Dropout | 0.4 | 0.2 | - | 0.5 |

Figure 12: Comparison of computational efficiency across different explanation approaches. Measurement execution time (in ms) required for model inference and explanation generation. The methods are categorized into: Gradient-based: Saliency, InputXGradient, GuidedBackprop, Deconvolution, Integrated Gradients, Masked-based: GNNExplainer, PGExplainer, Self-Explainable: ProtGNN, PGIB, and Fragment-based: SME, HiGNN, and our SEAL.

Table 11: AUROC score of various graph neural network architectures on the B-XAIC benchmark.

| Model | rings-count | rings-max | X | P | B | Indole | PAINS |
|---|---|---|---|---|---|---|---|
| | | | AUROC $\uparrow$ | | | | |
| GIN | $\mathbf{1.00} \pm 0.00$ | $0.93 \pm 0.02$ | $\mathbf{1.00} \pm 0.00$ | $\mathbf{1.00} \pm 0.00$ | $\mathbf{1.00} \pm 0.00$ | $\mathbf{1.00} \pm 0.00$ | $\mathbf{0.99} \pm 0.00$ |
| GCN | $\mathbf{1.00} \pm 0.00$ | $0.82 \pm 0.01$ | $\mathbf{1.00} \pm 0.00$ | $\mathbf{1.00} \pm 0.00$ | $\mathbf{1.00} \pm 0.00$ | $0.99 \pm 0.00$ | $0.97 \pm 0.00$ |
| GAT | $0.88 \pm 0.01$ | $0.75 \pm 0.02$ | $\mathbf{1.00} \pm 0.00$ | $\mathbf{1.00} \pm 0.00$ | $\mathbf{1.00} \pm 0.00$ | $0.97 \pm 0.00$ | $0.92 \pm 0.01$ |
| HIGNN | $0.97 \pm 0.00$ | $0.91 \pm 0.01$ | $\mathbf{1.00} \pm 0.00$ | $\mathbf{1.00} \pm 0.00$ | $\mathbf{1.00} \pm 0.00$ | $0.99 \pm 0.00$ | $\mathbf{0.99} \pm 0.00$ |
| ProtGNN | $0.98 \pm 0.01$ | $0.68 \pm 0.06$ | $0.94 \pm 0.03$ | $0.98 \pm 0.04$ | $0.79 \pm 0.19$ | $0.98 \pm 0.01$ | $0.88 \pm 0.10$ |
| SME | $\mathbf{1.00} \pm 0.00$ | $0.86 \pm 0.01$ | $\mathbf{1.00} \pm 0.00$ | $\mathbf{1.00} \pm 0.00$ | $\mathbf{1.00} \pm 0.00$ | $0.99 \pm 0.00$ | $0.98 \pm 0.00$ |
| PGIB | $0.83 \pm 0.02$ | $0.76 \pm 0.02$ | $\mathbf{1.00} \pm 0.00$ | $\mathbf{1.00} \pm 0.00$ | $\mathbf{1.00} \pm 0.00$ | $0.97 \pm 0.00$ | $0.90 \pm 0.02$ |
| SEAL ($\lambda = 2$) | $0.97 \pm 0.01$ | $\mathbf{0.99} \pm 0.01$ | $\mathbf{1.00} \pm 0.00$ | $\mathbf{1.00} \pm 0.00$ | $\mathbf{1.00} \pm 0.00$ | $\mathbf{1.00} \pm 0.00$ | $0.95 \pm 0.00$ |
| SEAL ($\lambda = 1$) | $0.97 \pm 0.00$ | $\mathbf{0.99} \pm 0.01$ | $\mathbf{1.00} \pm 0.00$ | $\mathbf{1.00} \pm 0.00$ | $\mathbf{1.00} \pm 0.00$ | $\mathbf{1.00} \pm 0.00$ | $0.96 \pm 0.01$ |
| SEAL ($\lambda = 0.5$) | $0.97 \pm 0.00$ | $\mathbf{0.99} \pm 0.00$ | $\mathbf{1.00} \pm 0.00$ | $\mathbf{1.00} \pm 0.00$ | $\mathbf{1.00} \pm 0.00$ | $\mathbf{1.00} \pm 0.00$ | $0.96 \pm 0.00$ |
| SEAL ($\lambda = 10^{-1}$) | $0.98 \pm 0.00$ | $\mathbf{0.99} \pm 0.00$ | $\mathbf{1.00} \pm 0.00$ | $\mathbf{1.00} \pm 0.00$ | $\mathbf{1.00} \pm 0.00$ | $\mathbf{1.00} \pm 0.00$ | $0.96 \pm 0.00$ |
| SEAL ($\lambda = 10^{-2}$) | $0.98 \pm 0.00$ | $\mathbf{0.99} \pm 0.01$ | $\mathbf{1.00} \pm 0.00$ | $\mathbf{1.00} \pm 0.00$ | $\mathbf{1.00} \pm 0.00$ | $\mathbf{1.00} \pm 0.00$ | $0.96 \pm 0.01$ |
| SEAL ($\lambda = 10^{-3}$) | $0.98 \pm 0.01$ | $\mathbf{0.99} \pm 0.00$ | $\mathbf{1.00} \pm 0.00$ | $\mathbf{1.00} \pm 0.00$ | $\mathbf{1.00} \pm 0.00$ | $\mathbf{1.00} \pm 0.00$ | $\mathbf{0.99} \pm 0.00$ |
| SEAL ($\lambda = 10^{-4}$) | $0.99 \pm 0.00$ | $\mathbf{0.99} \pm 0.01$ | $\mathbf{1.00} \pm 0.00$ | $\mathbf{1.00} \pm 0.00$ | $\mathbf{1.00} \pm 0.00$ | $\mathbf{1.00} \pm 0.00$ | $\mathbf{0.99} \pm 0.00$ |
| SEAL ($\lambda = 0$) | $0.99 \pm 0.00$ | $0.98 \pm 0.00$ | $\mathbf{1.00} \pm 0.00$ | $\mathbf{1.00} \pm 0.00$ | $\mathbf{1.00} \pm 0.00$ | $\mathbf{1.00} \pm 0.00$ | $\mathbf{0.99} \pm 0.00$ |
| SEALAtom ($\lambda = 2$) | $0.83 \pm 0.01$ | $0.66 \pm 0.02$ | $\mathbf{1.00} \pm 0.00$ | $\mathbf{1.00} \pm 0.00$ | $\mathbf{1.00} \pm 0.00$ | $0.74 \pm 0.01$ | $0.71 \pm 0.01$ |
| SEALAtom ($\lambda = 1$) | $0.82 \pm 0.01$ | $0.66 \pm 0.02$ | $\mathbf{1.00} \pm 0.00$ | $\mathbf{1.00} \pm 0.00$ | $\mathbf{1.00} \pm 0.00$ | $0.75 \pm 0.01$ | $0.71 \pm 0.01$ |
| SEALAtom ($\lambda = 0.5$) | $0.82 \pm 0.02$ | $0.66 \pm 0.01$ | $\mathbf{1.00} \pm 0.00$ | $\mathbf{1.00} \pm 0.00$ | $\mathbf{1.00} \pm 0.00$ | $0.74 \pm 0.01$ | $0.71 \pm 0.01$ |
| SEALAtom ($\lambda = 10^{-1}$) | $0.81 \pm 0.02$ | $0.65 \pm 0.02$ | $\mathbf{1.00} \pm 0.00$ | $\mathbf{1.00} \pm 0.00$ | $\mathbf{1.00} \pm 0.00$ | $0.74 \pm 0.01$ | $0.71 \pm 0.02$ |
| SEALAtom ($\lambda = 10^{-2}$) | $0.86 \pm 0.02$ | $0.66 \pm 0.01$ | $\mathbf{1.00} \pm 0.00$ | $\mathbf{1.00} \pm 0.00$ | $\mathbf{1.00} \pm 0.00$ | $0.75 \pm 0.02$ | $0.70 \pm 0.01$ |
| SEALAtom ($\lambda = 10^{-3}$) | $0.93 \pm 0.01$ | $0.69 \pm 0.01$ | $\mathbf{1.00} \pm 0.00$ | $\mathbf{1.00} \pm 0.00$ | $\mathbf{1.00} \pm 0.00$ | $0.95 \pm 0.03$ | $0.82 \pm 0.03$ |
| SEALAtom ($\lambda = 10^{-4}$) | $0.96 \pm 0.02$ | $0.74 \pm 0.01$ | $\mathbf{1.00} \pm 0.00$ | $\mathbf{1.00} \pm 0.00$ | $\mathbf{1.00} \pm 0.00$ | $\mathbf{1.00} \pm 0.00$ | $0.96 \pm 0.01$ |
| SEALAtom ($\lambda = 0$) | $0.97 \pm 0.00$ | $0.93 \pm 0.01$ | $\mathbf{1.00} \pm 0.00$ | $\mathbf{1.00} \pm 0.00$ | $\mathbf{1.00} \pm 0.00$ | $\mathbf{1.00} \pm 0.00$ | $\mathbf{0.99} \pm 0.00$ |

Table 12: F1 score of various graph neural network architectures on the B-XAIC benchmark.

| Model | rings-count | rings-max | X | P | B | Indole | PAINS |
|---|---|---|---|---|---|---|---|
| | | | F1 Score ↑ | | | | |
| GIN | **1.00** ± 0.00 | **0.96** ± 0.00 | **1.00** ± 0.00 | **1.00** ± 0.00 | **1.00** ± 0.00 | **0.99** ± 0.00 | 0.97± 0.00 |
| GCN | 0.98 ± 0.00 | 0.93 ± 0.01 | **1.00** ± 0.00 | **1.00** ± 0.00 | **1.00** ± 0.00 | 0.97 ± 0.00 | 0.93 ± 0.00 |
| GAT | 0.79 ± 0.03 | 0.92 ± 0.01 | **1.00** ± 0.00 | **1.00** ± 0.00 | **1.00** ± 0.00 | 0.92 ± 0.01 | 0.85 ± 0.01 |
| HIGNN | 0.92 ± 0.01 | 0.95 ± 0.00 | **1.00** ± 0.00 | **1.00** ± 0.00 | **1.00** ± 0.00 | 0.95 ± 0.01 | 0.96 ± 0.01 |
| ProtGNN | 0.94 ± 0.01 | 0.92 ± 0.00 | 0.86 ± 0.01 | 0.94 ± 0.05 | 0.98 ± 0.01 | 0.91 ± 0.03 | 0.86 ± 0.05 |
| SME | **1.00** ± 0.00 | 0.95 ± 0.00 | **1.00** ± 0.00 | **1.00** ± 0.00 | **1.00** ± 0.00 | **0.99** ± 0.00 | **0.98** ± 0.00 |
| PGIB | 0.77 ± 0.01 | 0.89 ± 0.04 | 0.99 ± 0.00 | **1.00** ± 0.00 | **1.00**± 0.00 | 0.91 ± 0.01 | 0.83 ± 0.02 |
| SEAL ($\lambda = 2$) | 0.90 ± 0.03 | 0.85 ± 0.05 | **1.00** ± 0.00 | **1.00** ± 0.00 | 0.99 ± 0.01 | 0.98 ± 0.00 | 0.86 ± 0.00 |
| SEAL ($\lambda = 1$) | 0.86 ± 0.02 | 0.87 ± 0.01 | **1.00** ± 0.00 | **1.00** ± 0.00 | 0.99 ± 0.01 | 0.98 ± 0.00 | 0.86 ± 0.01 |
| SEAL ($\lambda = 0.5$) | 0.87 ± 0.03 | 0.87 ± 0.02 | **1.00** ± 0.00 | **1.00** ± 0.00 | 0.99 ± 0.01 | 0.98 ± 0.00 | 0.86 ± 0.01 |
| SEAL ($\lambda = 10^{-1}$) | 0.88 ± 0.04 | 0.89 ± 0.01 | **1.00** ± 0.00 | 0.99 ± 0.01 | 0.99 ± 0.01 | 0.98 ± 0.00 | 0.86 ± 0.01 |
| SEAL ($\lambda = 10^{-2}$) | 0.92 ± 0.01 | 0.90 ± 0.02 | **1.00** ± 0.00 | **1.00** ± 0.00 | 0.98 ± 0.02 | **0.99** ± 0.00 | 0.86 ± 0.02 |
| SEAL ($\lambda = 10^{-3}$) | 0.93 ± 0.02 | 0.91 ± 0.02 | **1.00** ± 0.00 | **1.00** ± 0.00 | 0.99 ± 0.01 | **0.99** ± 0.00 | 0.93 ± 0.01 |
| SEAL ($\lambda = 10^{-4}$) | 0.94 ± 0.01 | 0.91 ± 0.02 | **1.00** ± 0.00 | **1.00** ± 0.00 | 0.98 ± 0.01 | **0.99** ± 0.00 | 0.95 ± 0.00 |
| SEAL ($\lambda = 0$) | 0.93 ± 0.01 | 0.88 ± 0.01 | **1.00** ± 0.00 | **1.00** ± 0.00 | **1.00** ± 0.00 | **0.99** ± 0.00 | 0.96 ± 0.01 |
| SEALAtom ($\lambda = 2$) | 0.66 ± 0.02 | 0.31 ± 0.03 | **1.00** ± 0.00 | **1.00** ± 0.00 | **1.00** ± 0.00 | 0.57 ± 0.02 | 0.54 ± 0.02 |
| SEALAtom ($\lambda = 1$) | 0.62 ± 0.02 | 0.31 ± 0.03 | **1.00** ± 0.00 | **1.00** ± 0.00 | **1.00** ± 0.00 | 0.58 ± 0.03 | 0.54 ± 0.02 |
| SEALAtom ($\lambda = 0.5$) | 0.62 ± 0.05 | 0.30 ± 0.03 | **1.00** ± 0.00 | **1.00** ± 0.00 | **1.00** ± 0.00 | 0.56 ± 0.03 | 0.54 ± 0.01 |
| SEALAtom ($\lambda = 10^{-1}$) | 0.56 ± 0.09 | 0.27 ± 0.04 | **1.00** ± 0.00 | **1.00** ± 0.00 | **1.00** ± 0.00 | 0.55 ± 0.03 | 0.48 ± 0.06 |
| SEALAtom ($\lambda = 10^{-2}$) | 0.68 ± 0.01 | 0.28 ± 0.02 | **1.00** ± 0.00 | **1.00** ± 0.00 | **1.00** ± 0.00 | 0.53 ± 0.05 | 0.46 ± 0.01 |
| SEALAtom ($\lambda = 10^{-3}$) | 0.83 ± 0.01 | 0.32 ± 0.03 | **1.00** ± 0.00 | **1.00** ± 0.00 | **1.00** ± 0.00 | 0.87 ± 0.07 | 0.62 ± 0.05 |
| SEALAtom ($\lambda = 10^{-4}$) | 0.87 ± 0.04 | 0.35 ± 0.03 | **1.00** ± 0.00 | **1.00** ± 0.00 | **1.00** ± 0.00 | 0.98 ± 0.00 | 0.85 ± 0.01 |
| SEALAtom ($\lambda = 0$) | 0.88 ± 0.01 | 0.68 ± 0.02 | **1.00** ± 0.00 | **1.00** ± 0.00 | **1.00** ± 0.00 | **0.99** ± 0.00 | 0.94 ± 0.01 |

Table 13: Accuracy score of various graph neural network architectures on the B-XAIC benchmark.

| Model | rings-count | rings-max | X | P | B | Indole | PAINS |
|---|---|---|---|---|---|---|---|
| | | | Accuracy ↑ | | | | |
| GIN | **1.00** ± 0.00 | 0.96 ± 0.00 | **1.00** ± 0.00 | **1.00** ± 0.00 | **1.00** ± 0.00 | **0.99** ± 0.00 | 0.97 ± 0.00 |
| GCN | 0.98 ± 0.00 | 0.93 ± 0.01 | **1.00** ± 0.00 | **1.00** ± 0.00 | **1.00** ± 0.00 | 0.97 ± 0.00 | 0.93 ± 0.00 |
| GAT | 0.81 ± 0.02 | 0.91 ± 0.02 | **1.00** ± 0.00 | **1.00** ± 0.00 | **1.00** ± 0.00 | 0.92 ± 0.01 | 0.86 ± 0.01 |
| HIGNN | 0.92 ± 0.01 | 0.95 ± 0.01 | **1.00** ± 0.00 | **1.00** ± 0.00 | **1.00** ± 0.00 | 0.95 ± 0.01 | 0.96 ± 0.01 |
| ProtGNN | 0.94 ± 0.01 | 0.94 ± 0.00 | 0.86 ± 0.01 | 0.95 ± 0.04 | 0.98 ± 0.00 | 0.91 ± 0.03 | 0.86 ± 0.04 |
| SME | **1.00** ± 0.00 | 0.95 ± 0.01 | **1.00**± 0.00 | **1.00** ± 0.00 | **1.00** ± 0.00 | **0.99** ± 0.00 | **0.98** ± 0.00 |
| PGIB | 0.77 ± 0.01 | 0.89 ± 0.04 | 0.99 ± 0.00 | **1.00**± 0.00 | **1.00** ± 0.00 | 0.91 ± 0.01 | 0.83 ± 0.02 |
| SEAL ($\lambda = 2$) | 0.93 ± 0.02 | 0.98 ± 0.01 | **1.00** ± 0.00 | **1.00** ± 0.00 | **1.00** ± 0.00 | **0.99** ± 0.00 | 0.91 ± 0.00 |
| SEAL ($\lambda = 1$) | 0.91 ± 0.02 | 0.98 ± 0.00 | **1.00** ± 0.00 | **1.00** ± 0.00 | **1.00** ± 0.00 | **0.99** ± 0.00 | 0.91 ± 0.01 |
| SEAL ($\lambda = 0.5$) | 0.92 ± 0.02 | 0.98 ± 0.00 | **1.00** ± 0.00 | **1.00** ± 0.00 | **1.00** ± 0.00 | **0.99**± 0.00 | 0.91 ± 0.01 |
| SEAL ($\lambda = 10^{-1}$) | 0.92 ± 0.03 | **0.99** ± 0.00 | **1.00** ± 0.00 | **1.00** ± 0.00 | **1.00** ± 0.00 | **0.99**± 0.00 | 0.91 ± 0.00 |
| SEAL ($\lambda = 10^{-2}$) | 0.95 ± 0.01 | **0.99** ± 0.00 | **1.00** ± 0.00 | **1.00** ± 0.00 | **1.00** ± 0.00 | **0.99** ± 0.00 | 0.91 ± 0.01 |
| SEAL ($\lambda = 10^{-3}$) | 0.96 ± 0.01 | **0.99** ± 0.00 | **1.00** ± 0.00 | **1.00** ± 0.00 | **1.00** ± 0.00 | **0.99** ± 0.00 | 0.95 ± 0.00 |
| SEAL ($\lambda = 10^{-4}$) | 0.97 ± 0.00 | **0.99** ± 0.00 | **1.00** ± 0.00 | **1.00** ± 0.00 | **1.00** ± 0.00 | **0.99** ± 0.00 | 0.97 ± 0.00 |
| SEAL ($\lambda = 0$) | 0.96 ± 0.00 | **0.99** ± 0.00 | **1.00** ± 0.00 | **1.00** ± 0.00 | **1.00** ± 0.00 | **0.99** ± 0.00 | 0.97 ± 0.00 |
| SEALAtom ($\lambda = 2$) | 0.82 ± 0.01 | 0.93 ± 0.01 | **1.00** ± 0.00 | **1.00** ± 0.00 | **1.00** ± 0.00 | 0.72 ± 0.01 | 0.72 ± 0.01 |
| SEALAtom ($\lambda = 1$) | 0.81 ± 0.01 | 0.93 ± 0.01 | **1.00** ± 0.00 | **1.00** ± 0.00 | **1.00** ± 0.00 | 0.71 ± 0.00 | 0.71 ± 0.02 |
| SEALAtom ($\lambda = 0.5$) | 0.81 ± 0.01 | 0.93 ± 0.01 | **1.00** ± 0.00 | **1.00** ± 0.00 | **1.00** ± 0.00 | 0.71 ± 0.01 | 0.71 ± 0.01 |
| SEALAtom ($\lambda = 10^{-1}$) | 0.80 ± 0.02 | 0.94 ± 0.01 | **1.00** ± 0.00 | **1.00** ± 0.00 | **1.00** ± 0.00 | 0.71 ± 0.01 | 0.73 ± 0.01 |
| SEALAtom ($\lambda = 10^{-2}$) | 0.83 ± 0.01 | 0.94 ± 0.01 | **1.00** ± 0.00 | **1.00** ± 0.00 | **1.00** ± 0.00 | 0.72 ± 0.02 | 0.73 ± 0.00 |
| SEALAtom ($\lambda = 10^{-3}$) | 0.91 ± 0.01 | 0.94 ± 0.01 | **1.00** ± 0.00 | **1.00** ± 0.00 | **1.00** ± 0.00 | 0.91 ± 0.05 | 0.79 ± 0.01 |
| SEALAtom ($\lambda = 10^{-4}$) | 0.93 ± 0.02 | 0.94 ± 0.01 | **1.00** ± 0.00 | **1.00** ± 0.00 | **1.00** ± 0.00 | 0.98 ± 0.00 | 0.91 ± 0.00 |
| SEALAtom ($\lambda = 0$) | 0.93 ± 0.00 | 0.96 ± 0.00 | **1.00** ± 0.00 | **1.00** ± 0.00 | **1.00** ± 0.00 | **0.99** ± 0.00 | 0.96 ± 0.00 |

Table 14: Performance of various model explanations on the B-XAIC benchmark. The subgraph explanation (SE) metric is employed for positive examples containing the relevant pattern.

| Model | rings-count | rings-max | X | P | B | Indole | PAINS |
|---|---|---|---|---|---|---|---|
| | | | SE ↑ | | | | |
| Deconvolution | $0.55 \pm 0.24$ | $0.36 \pm 0.22$ | $0.07 \pm 0.00$ | $0.90 \pm 0.00$ | $0.72 \pm 0.01$ | $0.36 \pm 0.21$ | $0.33 \pm 0.01$ |
| GuidedBackprop | $0.69 \pm 0.05$ | $0.67 \pm 0.02$ | $0.94 \pm 0.01$ | $0.85 \pm 0.11$ | $\mathbf{1.00} \pm 0.00$ | $0.85 \pm 0.03$ | $0.78 \pm 0.02$ |
| IntegratedGrad | $0.36 \pm 0.00$ | $0.64 \pm 0.04$ | $\mathbf{1.00} \pm 0.00$ | $\mathbf{1.00} \pm 0.00$ | $\mathbf{1.00} \pm 0.00$ | $0.84 \pm 0.06$ | $0.76 \pm 0.02$ |
| Saliency | $0.51 \pm 0.04$ | $0.66 \pm 0.03$ | $0.92 \pm 0.02$ | $\mathbf{1.00} \pm 0.00$ | $\mathbf{1.00} \pm 0.00$ | $0.87 \pm 0.02$ | $0.81 \pm 0.01$ |
| InputXGradient | $0.49 \pm 0.03$ | $0.48 \pm 0.03$ | $\mathbf{1.00} \pm 0.00$ | $\mathbf{1.00} \pm 0.00$ | $\mathbf{1.00} \pm 0.00$ | $0.74 \pm 0.05$ | $0.54 \pm 0.03$ |
| GNNExplainer | $0.49 \pm 0.01$ | $0.50 \pm 0.00$ | $0.50 \pm 0.00$ | $0.51 \pm 0.01$ | $0.53 \pm 0.05$ | $0.53 \pm 0.03$ | $0.54 \pm 0.06$ |
| HiGNN | $0.77 \pm 0.00$ | $0.75 \pm 0.00$ | $0.82 \pm 0.00$ | $0.82 \pm 0.01$ | $0.76 \pm 0.01$ | $0.78 \pm 0.02$ | $0.66 \pm 0.02$ |
| ProtGNN | $0.51 \pm 0.04$ | $0.61 \pm 0.06$ | $0.54 \pm 0.11$ | $0.99 \pm 0.01$ | $0.97 \pm 0.02$ | $0.73 \pm 0.11$ | $0.67 \pm 0.10$ |
| PGExplainer | $0.50 \pm 0.00$ | $0.50 \pm 0.00$ | $0.50 \pm 0.00$ | $0.50 \pm 0.00$ | $0.50 \pm 0.00$ | $\mathbf{0.96} \pm 0.02$ | $0.74 \pm 0.20$ |
| PGIB | $0.64 \pm 0.06$ | $0.56 \pm 0.01$ | $0.86 \pm 0.03$ | $0.92 \pm 0.02$ | $0.93 \pm 0.02$ | $0.88 \pm 0.02$ | $0.73 \pm 0.01$ |
| SMEAtom | $0.81 \pm 0.01$ | $0.69 \pm 0.01$ | $0.71 \pm 0.15$ | $0.72 \pm 0.20$ | $\mathbf{1.00} \pm 0.00$ | $0.59 \pm 0.04$ | $0.70 \pm 0.02$ |
| SMEBrics | $0.78 \pm 0.00$ | $0.79 \pm 0.01$ | $0.75 \pm 0.04$ | $0.81 \pm 0.01$ | $0.77 \pm 0.01$ | $0.90 \pm 0.00$ | $0.78 \pm 0.01$ |
| SMEMurcko | $0.59 \pm 0.00$ | $0.52 \pm 0.01$ | $0.52 \pm 0.01$ | $0.56 \pm 0.01$ | $0.52 \pm 0.01$ | $0.54 \pm 0.00$ | $0.54 \pm 0.00$ |
| SEAL ($\lambda = 2$) | $\mathbf{1.00} \pm 0.00$ | $0.87 \pm 0.01$ | $\mathbf{1.00} \pm 0.00$ | $0.99 \pm 0.00$ | $0.88 \pm 0.01$ | $\mathbf{0.96} \pm 0.00$ | $0.77 \pm 0.00$ |
| SEAL ($\lambda = 1$) | $\mathbf{1.00} \pm 0.00$ | $\mathbf{0.88} \pm 0.00$ | $\mathbf{1.00} \pm 0.00$ | $0.99 \pm 0.00$ | $0.88 \pm 0.01$ | $\mathbf{0.96} \pm 0.00$ | $0.77 \pm 0.01$ |
| SEAL ($\lambda = 0.5$) | $\mathbf{1.00} \pm 0.00$ | $0.87 \pm 0.01$ | $\mathbf{1.00} \pm 0.00$ | $0.99 \pm 0.00$ | $0.88 \pm 0.01$ | $\mathbf{0.96} \pm 0.00$ | $0.78 \pm 0.01$ |
| SEAL ($\lambda = 10^{-1}$) | $\mathbf{1.00} \pm 0.00$ | $\mathbf{0.88} \pm 0.00$ | $\mathbf{1.00} \pm 0.00$ | $0.99 \pm 0.00$ | $0.88 \pm 0.01$ | $\mathbf{0.96} \pm 0.00$ | $0.78 \pm 0.01$ |
| SEAL ($\lambda = 10^{-2}$) | $\mathbf{1.00} \pm 0.00$ | $\mathbf{0.88} \pm 0.01$ | $\mathbf{1.00} \pm 0.00$ | $0.99 \pm 0.00$ | $0.88 \pm 0.01$ | $\mathbf{0.96} \pm 0.00$ | $0.78 \pm 0.01$ |
| SEAL ($\lambda = 10^{-3}$) | $0.98 \pm 0.01$ | $0.75 \pm 0.04$ | $\mathbf{1.00} \pm 0.00$ | $0.96 \pm 0.04$ | $0.88 \pm 0.01$ | $\mathbf{0.96} \pm 0.00$ | $0.80 \pm 0.01$ |
| SEAL ($\lambda = 10^{-4}$) | $0.96 \pm 0.01$ | $0.60 \pm 0.06$ | $\mathbf{1.00} \pm 0.00$ | $0.96 \pm 0.04$ | $0.88 \pm 0.01$ | $\mathbf{0.96} \pm 0.00$ | $\mathbf{0.83} \pm 0.02$ |
| SEAL ($\lambda = 0$) | $0.87 \pm 0.04$ | $0.44 \pm 0.03$ | $\mathbf{1.00} \pm 0.00$ | $0.91 \pm 0.04$ | $0.88 \pm 0.01$ | $\mathbf{0.96} \pm 0.00$ | $\mathbf{0.83} \pm 0.01$ |
| SEALAtom ($\lambda = 2$) | $0.74 \pm 0.07$ | $0.54 \pm 0.05$ | $\mathbf{1.00} \pm 0.00$ | $\mathbf{1.00} \pm 0.00$ | $\mathbf{1.00} \pm 0.00$ | $0.62 \pm 0.07$ | $0.49 \pm 0.03$ |
| SEALAtom ($\lambda = 1$) | $0.70 \pm 0.03$ | $0.50 \pm 0.03$ | $\mathbf{1.00} \pm 0.00$ | $\mathbf{1.00} \pm 0.00$ | $\mathbf{1.00} \pm 0.00$ | $0.65 \pm 0.06$ | $0.46 \pm 0.01$ |
| SEALAtom ($\lambda = 0.5$) | $0.70 \pm 0.06$ | $0.53 \pm 0.06$ | $\mathbf{1.00} \pm 0.00$ | $\mathbf{1.00} \pm 0.00$ | $\mathbf{1.00} \pm 0.00$ | $0.62 \pm 0.06$ | $0.46 \pm 0.01$ |
| SEALAtom ($\lambda = 10^{-1}$) | $0.75 \pm 0.05$ | $0.54 \pm 0.06$ | $\mathbf{1.00} \pm 0.00$ | $\mathbf{1.00} \pm 0.00$ | $\mathbf{1.00} \pm 0.00$ | $0.62 \pm 0.07$ | $0.48 \pm 0.03$ |
| SEALAtom ($\lambda = 10^{-2}$) | $0.71 \pm 0.01$ | $0.58 \pm 0.01$ | $\mathbf{1.00} \pm 0.00$ | $\mathbf{1.00} \pm 0.00$ | $\mathbf{1.00} \pm 0.00$ | $0.60 \pm 0.07$ | $0.48 \pm 0.02$ |
| SEALAtom ($\lambda = 10^{-3}$) | $0.66 \pm 0.00$ | $0.59 \pm 0.01$ | $\mathbf{1.00} \pm 0.00$ | $\mathbf{1.00} \pm 0.00$ | $0.95 \pm 0.04$ | $0.68 \pm 0.06$ | $0.58 \pm 0.03$ |
| SEALAtom ($\lambda = 10^{-4}$) | $0.65 \pm 0.00$ | $0.56 \pm 0.01$ | $\mathbf{1.00} \pm 0.00$ | $0.98 \pm 0.04$ | $0.97 \pm 0.06$ | $0.71 \pm 0.06$ | $0.61 \pm 0.01$ |
| SEALAtom ($\lambda = 0$) | $0.61 \pm 0.01$ | $0.63 \pm 0.01$ | $0.95 \pm 0.01$ | $0.85 \pm 0.08$ | $0.91 \pm 0.05$ | $0.75 \pm 0.01$ | $0.64 \pm 0.02$ |

Table 15: Performance of various model explanations on the B-XAIC benchmark. The null explanation (NE) metric is employed for negative examples, checking uniform distribution.

| Model | rings-count | rings-max | X | P | B | Indole | PAINS |
|---|---|---|---|---|---|---|---|
| | | | NE ↑ | | | | |
| Deconvolution | $0.56 \pm 0.06$ | $0.82 \pm 0.01$ | $0.84 \pm 0.01$ | $0.84 \pm 0.01$ | $0.82 \pm 0.01$ | $\mathbf{0.80} \pm 0.01$ | $\mathbf{0.81} \pm 0.01$ |
| GuidedBackprop | $0.32 \pm 0.08$ | $0.19 \pm 0.03$ | $0.35 \pm 0.10$ | $0.51 \pm 0.04$ | $0.45 \pm 0.03$ | $0.33 \pm 0.05$ | $0.28 \pm 0.02$ |
| IntegratedGradients | $0.81 \pm 0.08$ | $0.75 \pm 0.06$ | $0.19 \pm 0.10$ | $0.36 \pm 0.37$ | $0.22 \pm 0.07$ | $0.31 \pm 0.13$ | $0.42 \pm 0.25$ |
| Saliency | $0.48 \pm 0.05$ | $0.41 \pm 0.03$ | $0.37 \pm 0.05$ | $0.55 \pm 0.03$ | $0.50 \pm 0.08$ | $0.42 \pm 0.03$ | $0.35 \pm 0.04$ |
| InputXGradient | $0.53 \pm 0.05$ | $0.49 \pm 0.02$ | $0.23 \pm 0.07$ | $0.69 \pm 0.16$ | $0.39 \pm 0.14$ | $0.49 \pm 0.01$ | $0.40 \pm 0.04$ |
| GNNExplainer | $0.80 \pm 0.07$ | $0.92 \pm 0.06$ | $0.65 \pm 0.02$ | $0.67 \pm 0.01$ | $0.67 \pm 0.00$ | $0.73 \pm 0.09$ | $0.55 \pm 0.26$ |
| HiGNN | $0.56 \pm 0.06$ | $0.31 \pm 0.02$ | $0.19 \pm 0.01$ | $0.14 \pm 0.00$ | $0.16 \pm 0.00$ | $0.38 \pm 0.03$ | $0.58 \pm 0.09$ |
| ProtGNN | $0.43 \pm 0.12$ | $0.40 \pm 0.03$ | $0.53 \pm 0.18$ | $0.64 \pm 0.33$ | $0.33 \pm 0.06$ | $0.46 \pm 0.06$ | $0.40 \pm 0.05$ |
| PGExplainer | $\mathbf{1.00} \pm 0.00$ | $\mathbf{0.99} \pm 0.00$ | $\mathbf{1.00} \pm 0.00$ | $\mathbf{1.00} \pm 0.00$ | $\mathbf{1.00} \pm 0.00$ | $0.16 \pm 0.09$ | $0.42 \pm 0.42$ |
| PGIB | $0.42 \pm 0.04$ | $0.31 \pm 0.05$ | $0.32 \pm 0.04$ | $0.36 \pm 0.06$ | $0.34 \pm 0.05$ | $0.33 \pm 0.03$ | $0.26 \pm 0.02$ |
| SMEAtom | $0.54 \pm 0.11$ | $0.07 \pm 0.01$ | $0.33 \pm 0.11$ | $0.54 \pm 0.11$ | $0.48 \pm 0.09$ | $0.03 \pm 0.01$ | $0.03 \pm 0.01$ |
| SMEBrics | $0.70 \pm 0.05$ | $0.57 \pm 0.00$ | $0.81 \pm 0.01$ | $0.80 \pm 0.01$ | $0.80 \pm 0.01$ | $0.74 \pm 0.00$ | $0.67 \pm 0.01$ |
| SMEMurcko | $\mathbf{1.00} \pm 0.00$ | $0.20 \pm 0.00$ | $0.46 \pm 0.01$ | $0.34 \pm 0.01$ | $0.38 \pm 0.00$ | $0.33 \pm 0.01$ | $0.30 \pm 0.01$ |
| SEAL ($\lambda = 2$) | $0.44 \pm 0.05$ | $0.72 \pm 0.02$ | $0.32 \pm 0.09$ | $0.16 \pm 0.06$ | $0.44 \pm 0.02$ | $0.64 \pm 0.04$ | $0.61 \pm 0.01$ |
| SEAL ($\lambda = 1$) | $0.42 \pm 0.05$ | $0.73 \pm 0.02$ | $0.36 \pm 0.09$ | $0.18 \pm 0.18$ | $0.45 \pm 0.03$ | $0.63 \pm 0.03$ | $0.62 \pm 0.01$ |
| SEAL ($\lambda = 0.5$) | $0.45 \pm 0.04$ | $0.73 \pm 0.04$ | $0.38 \pm 0.08$ | $0.21 \pm 0.16$ | $0.46 \pm 0.05$ | $0.63 \pm 0.04$ | $0.63 \pm 0.01$ |
| SEAL ($\lambda = 10^{-1}$) | $0.53 \pm 0.06$ | $0.72 \pm 0.02$ | $0.32 \pm 0.05$ | $0.22 \pm 0.07$ | $0.48 \pm 0.05$ | $0.68 \pm 0.03$ | $0.63 \pm 0.01$ |
| SEAL ($\lambda = 10^{-2}$) | $0.48 \pm 0.04$ | $0.72 \pm 0.02$ | $0.51 \pm 0.05$ | $0.09 \pm 0.05$ | $0.49 \pm 0.04$ | $0.63 \pm 0.02$ | $0.63 \pm 0.01$ |
| SEAL ($\lambda = 10^{-3}$) | $0.71 \pm 0.10$ | $0.74 \pm 0.01$ | $0.36 \pm 0.07$ | $0.10 \pm 0.01$ | $0.43 \pm 0.07$ | $0.69 \pm 0.05$ | $0.65 \pm 0.01$ |
| SEAL ($\lambda = 10^{-4}$) | $0.70 \pm 0.05$ | $0.73 \pm 0.01$ | $0.33 \pm 0.09$ | $0.10 \pm 0.02$ | $0.26 \pm 0.06$ | $0.65 \pm 0.05$ | $0.70 \pm 0.03$ |
| SEAL ($\lambda = 0$) | $0.59 \pm 0.06$ | $0.70 \pm 0.02$ | $0.60 \pm 0.07$ | $0.16 \pm 0.08$ | $0.38 \pm 0.12$ | $0.57 \pm 0.05$ | $0.68 \pm 0.01$ |
| SEALAtom ($\lambda = 2$) | $0.59 \pm 0.02$ | $0.11 \pm 0.01$ | $0.18 \pm 0.10$ | $0.08 \pm 0.06$ | $0.03 \pm 0.02$ | $0.09 \pm 0.01$ | $0.00 \pm 0.00$ |
| SEALAtom ($\lambda = 1$) | $0.59 \pm 0.04$ | $0.14 \pm 0.05$ | $0.16 \pm 0.12$ | $0.11 \pm 0.06$ | $0.06 \pm 0.02$ | $0.09 \pm 0.01$ | $0.00 \pm 0.00$ |
| SEALAtom ($\lambda = 0.5$) | $0.59 \pm 0.02$ | $0.13 \pm 0.05$ | $0.11 \pm 0.11$ | $0.12 \pm 0.01$ | $0.06 \pm 0.02$ | $0.09 \pm 0.01$ | $0.00 \pm 0.00$ |
| SEALAtom ($\lambda = 10^{-1}$) | $0.60 \pm 0.02$ | $0.07 \pm 0.03$ | $0.12 \pm 0.09$ | $0.09 \pm 0.06$ | $0.02 \pm 0.00$ | $0.09 \pm 0.01$ | $0.00 \pm 0.00$ |
| SEALAtom ($\lambda = 10^{-2}$) | $0.58 \pm 0.05$ | $0.06 \pm 0.05$ | $0.23 \pm 0.06$ | $0.14 \pm 0.12$ | $0.05 \pm 0.03$ | $0.13 \pm 0.05$ | $0.00 \pm 0.00$ |
| SEALAtom ($\lambda = 10^{-3}$) | $0.48 \pm 0.08$ | $0.09 \pm 0.02$ | $0.22 \pm 0.04$ | $0.09 \pm 0.06$ | $0.04 \pm 0.03$ | $0.05 \pm 0.03$ | $0.08 \pm 0.05$ |
| SEALAtom ($\lambda = 10^{-4}$) | $0.34 \pm 0.04$ | $0.06 \pm 0.04$ | $0.14 \pm 0.07$ | $0.05 \pm 0.03$ | $0.03 \pm 0.02$ | $0.04 \pm 0.02$ | $0.05 \pm 0.01$ |
| SEALAtom ($\lambda = 0$) | $0.40 \pm 0.05$ | $0.08 \pm 0.02$ | $0.41 \pm 0.23$ | $0.35 \pm 0.29$ | $0.29 \pm 0.12$ | $0.06 \pm 0.01$ | $0.08 \pm 0.01$ |

Table 16: Comparison of model performance on real-world datasets (hERG and CYP2C9). Results are reported for both Random and Scaffold splitting strategies.

| | Model | Random | | | Scaffold | | |
| --- | --- | --- | --- | --- | --- | --- | --- |
| | | AUROC ↑ | F1 ↑ | Accuracy ↑ | AUROC ↑ | F1 ↑ | Accuracy ↑ |
| hERG | GIN | $0.86 \pm 0.01$ | $0.78 \pm 0.01$ | $0.78 \pm 0.01$ | $0.79 \pm 0.02$ | $0.70 \pm 0.04$ | $0.72 \pm 0.02$ |
| | GAT | $0.70 \pm 0.01$ | $0.64 \pm 0.01$ | $0.65 \pm 0.01$ | $0.75 \pm 0.02$ | $0.66 \pm 0.04$ | $0.71 \pm 0.02$ |
| | GCN | $0.81 \pm 0.03$ | $0.73 \pm 0.02$ | $0.73 \pm 0.02$ | $0.76 \pm 0.02$ | $0.67 \pm 0.04$ | $0.71 \pm 0.03$ |
| | HiGNN | $\mathbf{0.87} \pm 0.01$ | $\mathbf{0.79} \pm 0.01$ | $\mathbf{0.79} \pm 0.01$ | $0.80 \pm 0.01$ | $0.72 \pm 0.01$ | $0.72 \pm 0.01$ |
| | ProtGNN | $0.76 \pm 0.06$ | $0.69 \pm 0.05$ | $0.69 \pm 0.05$ | $0.76 \pm 0.05$ | $0.69 \pm 0.05$ | $0.70 \pm 0.05$ |
| | SME | $0.76 \pm 0.01$ | $0.76 \pm 0.01$ | $0.76 \pm 0.01$ | $0.76 \pm 0.01$ | $\mathbf{0.77} \pm 0.01$ | $\mathbf{0.77} \pm 0.01$ |
| | PGIB | $0.75 \pm 0.02$ | $0.68 \pm 0.02$ | $0.68 \pm 0.02$ | $0.74 \pm 0.01$ | $0.67 \pm 0.01$ | $0.67 \pm 0.01$ |
| | SEAL ($\lambda = 2$) | $0.81 \pm 0.01$ | $0.71 \pm 0.03$ | $0.74 \pm 0.01$ | $0.76 \pm 0.02$ | $0.67 \pm 0.02$ | $0.71 \pm 0.01$ |
| | SEAL ($\lambda = 1$) | $0.81 \pm 0.01$ | $0.73 \pm 0.02$ | $0.75 \pm 0.01$ | $0.76 \pm 0.01$ | $0.67 \pm 0.04$ | $0.71 \pm 0.01$ |
| | SEAL ($\lambda = 0.5$) | $0.81 \pm 0.01$ | $0.73 \pm 0.02$ | $0.75 \pm 0.01$ | $0.76 \pm 0.02$ | $0.68 \pm 0.04$ | $0.71 \pm 0.01$ |
| | SEAL ($\lambda = 10^{-1}$) | $0.79 \pm 0.01$ | $0.70 \pm 0.02$ | $0.73 \pm 0.01$ | $0.76 \pm 0.02$ | $0.67 \pm 0.04$ | $0.71 \pm 0.02$ |
| | SEAL ($\lambda = 10^{-2}$) | $0.80 \pm 0.00$ | $0.70 \pm 0.02$ | $0.73 \pm 0.01$ | $0.75 \pm 0.01$ | $0.65 \pm 0.04$ | $0.70 \pm 0.01$ |
| | SEAL ($\lambda = 10^{-3}$) | $0.81 \pm 0.03$ | $0.71 \pm 0.03$ | $0.74 \pm 0.02$ | $0.78 \pm 0.02$ | $0.69 \pm 0.03$ | $0.72 \pm 0.01$ |
| | SEAL ($\lambda = 10^{-4}$) | $0.85 \pm 0.01$ | $0.76 \pm 0.01$ | $0.77 \pm 0.00$ | $\mathbf{0.81} \pm 0.01$ | $0.71 \pm 0.03$ | $0.74 \pm 0.02$ |
| | SEAL ($\lambda = 0$) | $0.85 \pm 0.01$ | $0.76 \pm 0.01$ | $0.78 \pm 0.01$ | $0.80 \pm 0.02$ | $0.71 \pm 0.03$ | $0.74 \pm 0.01$ |
| | SEALAtom ($\lambda = 2$) | $0.65 \pm 0.01$ | $0.49 \pm 0.06$ | $0.62 \pm 0.01$ | $0.66 \pm 0.02$ | $0.50 \pm 0.10$ | $0.63 \pm 0.03$ |
| | SEALAtom ($\lambda = 1$) | $0.65 \pm 0.01$ | $0.50 \pm 0.08$ | $0.62 \pm 0.02$ | $0.66 \pm 0.02$ | $0.53 \pm 0.11$ | $0.63 \pm 0.02$ |
| | SEALAtom ($\lambda = 0.5$) | $0.65 \pm 0.01$ | $0.50 \pm 0.05$ | $0.62 \pm 0.01$ | $0.65 \pm 0.01$ | $0.52 \pm 0.10$ | $0.63 \pm 0.03$ |
| | SEALAtom ($\lambda = 10^{-1}$) | $0.66 \pm 0.01$ | $0.51 \pm 0.05$ | $0.63 \pm 0.01$ | $0.66 \pm 0.01$ | $0.52 \pm 0.10$ | $0.63 \pm 0.03$ |
| | SEALAtom ($\lambda = 10^{-2}$) | $0.65 \pm 0.01$ | $0.53 \pm 0.02$ | $0.63 \pm 0.01$ | $0.66 \pm 0.01$ | $0.51 \pm 0.10$ | $0.63 \pm 0.03$ |
| | SEALAtom ($\lambda = 10^{-3}$) | $0.71 \pm 0.01$ | $0.60 \pm 0.01$ | $0.67 \pm 0.01$ | $0.66 \pm 0.03$ | $0.54 \pm 0.08$ | $0.64 \pm 0.03$ |
| | SEALAtom ($\lambda = 10^{-4}$) | $0.75 \pm 0.01$ | $0.63 \pm 0.01$ | $0.69 \pm 0.01$ | $0.71 \pm 0.02$ | $0.62 \pm 0.05$ | $0.68 \pm 0.02$ |
| | SEALAtom ($\lambda = 0$) | $0.84 \pm 0.01$ | $0.75 \pm 0.01$ | $0.77 \pm 0.01$ | $0.77 \pm 0.01$ | $0.68 \pm 0.05$ | $0.72 \pm 0.02$ |
| CYP2C9 | GIN | $\mathbf{0.86} \pm 0.01$ | $0.79 \pm 0.01$ | $0.79 \pm 0.01$ | $0.83 \pm 0.01$ | $0.65 \pm 0.03$ | $0.79 \pm 0.02$ |
| | GAT | $0.68 \pm 0.01$ | $0.67 \pm 0.01$ | $0.69 \pm 0.01$ | $0.67 \pm 0.02$ | $0.21 \pm 0.11$ | $0.70 \pm 0.05$ |
| | GCN | $0.84 \pm 0.01$ | $0.78 \pm 0.01$ | $0.78 \pm 0.01$ | $0.80 \pm 0.02$ | $0.57 \pm 0.03$ | $0.77 \pm 0.04$ |
| | HiGNN | $0.85 \pm 0.00$ | $0.76 \pm 0.02$ | $0.75 \pm 0.02$ | $\mathbf{0.84} \pm 0.01$ | $0.75 \pm 0.04$ | $0.74 \pm 0.04$ |
| | ProtGNN | $0.85 \pm 0.00$ | $0.77 \pm 0.01$ | $0.77 \pm 0.01$ | $0.83 \pm 0.01$ | $0.78 \pm 0.02$ | $0.78 \pm 0.02$ |
| | SME | $0.80 \pm 0.02$ | $\mathbf{0.82} \pm 0.01$ | $\mathbf{0.82} \pm 0.01$ | $0.80 \pm 0.01$ | $\mathbf{0.81} \pm 0.02$ | $\mathbf{0.80} \pm 0.02$ |
| | PGIB | $0.75 \pm 0.02$ | $0.72 \pm 0.01$ | $0.72 \pm 0.01$ | $0.79 \pm 0.02$ | $0.74 \pm 0.03$ | $0.74 \pm 0.03$ |
| | SEAL ($\lambda = 2$) | $0.81 \pm 0.01$ | $0.65 \pm 0.02$ | $0.78 \pm 0.01$ | $0.79 \pm 0.02$ | $0.64 \pm 0.01$ | $0.75 \pm 0.02$ |
| | SEAL ($\lambda = 1$) | $0.81 \pm 0.01$ | $0.64 \pm 0.03$ | $0.78 \pm 0.01$ | $0.79 \pm 0.02$ | $0.65 \pm 0.02$ | $0.76 \pm 0.02$ |
| | SEAL ($\lambda = 0.5$) | $0.81 \pm 0.00$ | $0.64 \pm 0.02$ | $0.78 \pm 0.01$ | $0.78 \pm 0.01$ | $0.63 \pm 0.03$ | $0.75 \pm 0.02$ |
| | SEAL ($\lambda = 10^{-1}$) | $0.79 \pm 0.01$ | $0.59 \pm 0.03$ | $0.76 \pm 0.01$ | $0.80 \pm 0.02$ | $0.64 \pm 0.02$ | $0.76 \pm 0.01$ |
| | SEAL ($\lambda = 10^{-2}$) | $0.79 \pm 0.01$ | $0.58 \pm 0.03$ | $0.76 \pm 0.01$ | $0.78 \pm 0.01$ | $0.61 \pm 0.02$ | $0.77 \pm 0.01$ |
| | SEAL ($\lambda = 10^{-3}$) | $0.83 \pm 0.01$ | $0.64 \pm 0.03$ | $0.78 \pm 0.01$ | $0.79 \pm 0.01$ | $0.62 \pm 0.01$ | $0.77 \pm 0.02$ |
| | SEAL ($\lambda = 10^{-4}$) | $0.83 \pm 0.00$ | $0.64 \pm 0.04$ | $0.78 \pm 0.01$ | $0.80 \pm 0.01$ | $0.62 \pm 0.03$ | $0.77 \pm 0.02$ |
| | SEAL ($\lambda = 0$) | $0.83 \pm 0.01$ | $0.64 \pm 0.03$ | $0.79 \pm 0.01$ | $0.82 \pm 0.01$ | $0.65 \pm 0.03$ | $0.78 \pm 0.02$ |
| | SEALAtom ($\lambda = 2$) | $0.63 \pm 0.03$ | $0.43 \pm 0.07$ | $0.70 \pm 0.01$ | $0.63 \pm 0.02$ | $0.40 \pm 0.04$ | $0.69 \pm 0.03$ |
| | SEALAtom ($\lambda = 1$) | $0.64 \pm 0.01$ | $0.38 \pm 0.06$ | $0.70 \pm 0.01$ | $0.62 \pm 0.02$ | $0.35 \pm 0.06$ | $0.69 \pm 0.04$ |
| | SEALAtom ($\lambda = 0.5$) | $0.62 \pm 0.01$ | $0.31 \pm 0.04$ | $0.69 \pm 0.01$ | $0.62 \pm 0.01$ | $0.33 \pm 0.03$ | $0.69 \pm 0.04$ |
| | SEALAtom ($\lambda = 10^{-1}$) | $0.61 \pm 0.00$ | $0.31 \pm 0.04$ | $0.68 \pm 0.01$ | $0.61 \pm 0.01$ | $0.32 \pm 0.01$ | $0.69 \pm 0.04$ |
| | SEALAtom ($\lambda = 10^{-2}$) | $0.70 \pm 0.02$ | $0.45 \pm 0.06$ | $0.72 \pm 0.01$ | $0.67 \pm 0.02$ | $0.46 \pm 0.06$ | $0.71 \pm 0.03$ |
| | SEALAtom ($\lambda = 10^{-3}$) | $0.73 \pm 0.01$ | $0.49 \pm 0.03$ | $0.74 \pm 0.01$ | $0.72 \pm 0.01$ | $0.52 \pm 0.03$ | $0.73 \pm 0.03$ |
| | SEALAtom ($\lambda = 10^{-4}$) | $0.80 \pm 0.01$ | $0.60 \pm 0.02$ | $0.77 \pm 0.01$ | $0.75 \pm 0.02$ | $0.60 \pm 0.03$ | $0.74 \pm 0.02$ |
| | SEALAtom ($\lambda = 0$) | $0.82 \pm 0.01$ | $0.60 \pm 0.03$ | $0.77 \pm 0.01$ | $0.76 \pm 0.02$ | $0.61 \pm 0.03$ | $0.73 \pm 0.03$ |

Table 17: Comparison of model performance on real-world datasets (MUTAG).

| | Model | AUROC ↑ | F1 ↑ | Accuracy ↑ |
|---|---|---|---|---|
| | GIN | $\mathbf{0.87} \pm 0.01$ | $\mathbf{0.81} \pm 0.00$ | $\mathbf{0.81} \pm 0.01$ |
| | GAT | $0.78 \pm 0.00$ | $0.72 \pm 0.01$ | $0.72 \pm 0.01$ |
| | GCN | $0.82 \pm 0.02$ | $0.75 \pm 0.01$ | $0.75 \pm 0.01$ |
| | HIGNN | $0.87 \pm 0.01$ | $0.80 \pm 0.02$ | $0.80 \pm 0.02$ |
| | ProtGNN | $0.86 \pm 0.01$ | $0.78 \pm 0.01$ | $0.78 \pm 0.01$ |
| | SME | $0.81 \pm 0.01$ | $\mathbf{0.81} \pm 0.01$ | $\mathbf{0.81} \pm 0.01$ |
| | PGIB | $0.83 \pm 0.02$ | $0.75 \pm 0.03$ | $0.75 \pm 0.03$ |
| MUTAG | SEAL ($\lambda = 2$) | $0.81 \pm 0.02$ | $0.75 \pm 0.01$ | $0.75 \pm 0.01$ |
| | SEAL ($\lambda = 1$) | $0.81 \pm 0.01$ | $0.75 \pm 0.02$ | $0.75 \pm 0.01$ |
| | SEAL ($\lambda = 0.5$) | $0.81 \pm 0.01$ | $0.75 \pm 0.02$ | $0.75 \pm 0.01$ |
| | SEAL ($\lambda = 10^{-1}$) | $0.81 \pm 0.01$ | $0.76 \pm 0.01$ | $0.75 \pm 0.01$ |
| | SEAL ($\lambda = 10^{-2}$) | $0.84 \pm 0.01$ | $0.79 \pm 0.01$ | $0.79 \pm 0.01$ |
| | SEAL ($\lambda = 10^{-3}$) | $0.85 \pm 0.01$ | $0.80 \pm 0.02$ | $0.79 \pm 0.01$ |
| | SEAL ($\lambda = 10^{-4}$) | $0.85 \pm 0.00$ | $0.79 \pm 0.01$ | $0.78 \pm 0.01$ |
| | SEAL ($\lambda = 0$) | $0.84 \pm 0.01$ | $0.77 \pm 0.03$ | $0.77 \pm 0.02$ |
| | SEALAtom ($\lambda = 2$) | $0.56 \pm 0.03$ | $0.38 \pm 0.13$ | $0.53 \pm 0.03$ |
| | SEALAtom ($\lambda = 1$) | $0.55 \pm 0.04$ | $0.36 \pm 0.13$ | $0.53 \pm 0.03$ |
| | SEALAtom ($\lambda = 0.5$) | $0.55 \pm 0.04$ | $0.38 \pm 0.13$ | $0.53 \pm 0.04$ |
| | SEALAtom ($\lambda = 10^{-1}$) | $0.55 \pm 0.05$ | $0.39 \pm 0.11$ | $0.53 \pm 0.04$ |
| | SEALAtom ($\lambda = 10^{-2}$) | $0.69 \pm 0.02$ | $0.62 \pm 0.03$ | $0.66 \pm 0.01$ |
| | SEALAtom ($\lambda = 10^{-3}$) | $0.70 \pm 0.03$ | $0.61 \pm 0.06$ | $0.66 \pm 0.03$ |
| | SEALAtom ($\lambda = 10^{-4}$) | $0.75 \pm 0.01$ | $0.68 \pm 0.02$ | $0.70 \pm 0.01$ |
| | SEALAtom ($\lambda = 0$) | $0.79 \pm 0.02$ | $0.73 \pm 0.03$ | $0.73 \pm 0.02$ |

Table 18: Comparison of model performance on real-world datasets (Solubility). Results are reported for both Random and Scaffold splitting strategies.

| | | Random | | Scaffold | |
|---|---|---|---|---|---|
| | Model | MAE ↓ | RMSE ↓ | MAE ↓ | RMSE ↓ |
| | GIN | $0.41 \pm 0.01$ | $0.60 \pm 0.02$ | $0.58 \pm 0.03$ | $0.82 \pm 0.06$ |
| | GAT | $0.57 \pm 0.01$ | $0.75 \pm 0.03$ | $0.60 \pm 0.02$ | $0.82 \pm 0.03$ |
| | GCN | $0.49 \pm 0.02$ | $0.67 \pm 0.03$ | $0.59 \pm 0.03$ | $0.82 \pm 0.04$ |
| | HiGNN | $0.38 \pm 0.05$ | $0.55 \pm 0.07$ | $0.53 \pm 0.06$ | $0.72 \pm 0.06$ |
| | SME | $\mathbf{0.32} \pm 0.01$ | $\mathbf{0.46} \pm 0.01$ | $\mathbf{0.39} \pm 0.02$ | $\mathbf{0.54} \pm 0.02$ |
| | SEAL ($\lambda = 2$) | $0.54 \pm 0.04$ | $0.73 \pm 0.05$ | $0.69 \pm 0.06$ | $0.90 \pm 0.05$ |
| | SEAL ($\lambda = 1$) | $0.53 \pm 0.05$ | $0.73 \pm 0.05$ | $0.66 \pm 0.03$ | $0.88 \pm 0.05$ |
| | SEAL ($\lambda = 0.5$) | $0.54 \pm 0.05$ | $0.73 \pm 0.05$ | $0.67 \pm 0.03$ | $0.88 \pm 0.03$ |
| Solubility | SEAL ($\lambda = 10^{-1}$) | $0.54 \pm 0.05$ | $0.73 \pm 0.06$ | $0.66 \pm 0.02$ | $0.87 \pm 0.02$ |
| | SEAL ($\lambda = 10^{-2}$) | $0.53 \pm 0.05$ | $0.73 \pm 0.04$ | $0.63 \pm 0.02$ | $0.84 \pm 0.03$ |
| | SEAL ($\lambda = 10^{-3}$) | $0.48 \pm 0.01$ | $0.68 \pm 0.04$ | $0.61 \pm 0.02$ | $0.82 \pm 0.02$ |
| | SEAL ($\lambda = 10^{-4}$) | $0.47 \pm 0.01$ | $0.66 \pm 0.04$ | $0.60 \pm 0.03$ | $0.81 \pm 0.03$ |
| | SEAL ($\lambda = 0$) | $0.45 \pm 0.01$ | $0.66 \pm 0.03$ | $0.59 \pm 0.02$ | $0.80 \pm 0.02$ |
| | SEALAtom ($\lambda = 2$) | $0.64 \pm 0.01$ | $0.81 \pm 0.01$ | $0.72 \pm 0.03$ | $0.93 \pm 0.03$ |
| | SEALAtom ($\lambda = 1$) | $0.63 \pm 0.01$ | $0.80 \pm 0.01$ | $0.70 \pm 0.03$ | $0.92 \pm 0.03$ |
| | SEALAtom ($\lambda = 0.5$) | $0.61 \pm 0.00$ | $0.78 \pm 0.01$ | $0.68 \pm 0.04$ | $0.90 \pm 0.05$ |
| | SEALAtom ($\lambda = 10^{-1}$) | $0.58 \pm 0.01$ | $0.76 \pm 0.01$ | $0.66 \pm 0.02$ | $0.88 \pm 0.04$ |
| | SEALAtom ($\lambda = 10^{-2}$) | $0.53 \pm 0.01$ | $0.72 \pm 0.01$ | $0.63 \pm 0.02$ | $0.87 \pm 0.04$ |
| | SEALAtom ($\lambda = 10^{-3}$) | $0.49 \pm 0.01$ | $0.69 \pm 0.01$ | $0.60 \pm 0.02$ | $0.85 \pm 0.05$ |
| | SEALAtom ($\lambda = 10^{-4}$) | $0.48 \pm 0.01$ | $0.66 \pm 0.01$ | $0.59 \pm 0.02$ | $0.82 \pm 0.04$ |
| | SEALAtom ($\lambda = 0$) | $0.47 \pm 0.01$ | $0.65 \pm 0.03$ | $0.61 \pm 0.02$ | $0.81 \pm 0.03$ |

Table 19: Performance of model explanations on real-world datasets (CYP2C9). Explanations are evaluated using Fidelity metrics at 10%, 20%, and 30% masking thresholds, representing the proportion of most important atoms (nodes) either removed or retained during the evaluation.

| Model | Fidelity$_{10}+\uparrow$ | Fidelity$_{10}-\downarrow$ | Fidelity$_{20}+\uparrow$ | Fidelity$_{20}-\downarrow$ | Fidelity$_{30}+\uparrow$ | Fidelity$_{30}-\downarrow$ |
|---|---|---|---|---|---|---|
| Deconvolution | $0.34 \pm 0.03$ | $0.37 \pm 0.03$ | $0.35 \pm 0.03$ | $0.36 \pm 0.03$ | $0.36 \pm 0.03$ | $0.35 \pm 0.03$ |
| GuidedBackprop | $0.36 \pm 0.03$ | $0.37 \pm 0.04$ | $0.36 \pm 0.03$ | $0.36 \pm 0.03$ | $0.36 \pm 0.03$ | $0.35 \pm 0.03$ |
| IntegratedGradients | $\mathbf{0.61} \pm 0.19$ | $0.28 \pm 0.15$ | $\mathbf{0.63} \pm 0.21$ | $0.27 \pm 0.15$ | $0.60 \pm 0.23$ | $0.24 \pm 0.15$ |
| Saliency | $0.38 \pm 0.05$ | $0.36 \pm 0.03$ | $0.38 \pm 0.05$ | $0.35 \pm 0.03$ | $0.39 \pm 0.06$ | $0.34 \pm 0.03$ |
| InputXGradient | $0.35 \pm 0.03$ | $0.41 \pm 0.11$ | $0.35 \pm 0.04$ | $0.40 \pm 0.10$ | $0.35 \pm 0.04$ | $0.39 \pm 0.09$ |
| GNNExplainer | $0.41 \pm 0.07$ | $0.32 \pm 0.06$ | $0.42 \pm 0.08$ | $0.31 \pm 0.08$ | $0.42 \pm 0.09$ | $0.28 \pm 0.09$ |
| HiGNN | $0.22 \pm 0.04$ | $0.37 \pm 0.10$ | $0.25 \pm 0.05$ | $0.36 \pm 0.09$ | $0.31 \pm 0.07$ | $0.31 \pm 0.08$ |
| ProtGNN | $0.42 \pm 0.10$ | $0.48 \pm 0.13$ | $0.46 \pm 0.12$ | $0.47 \pm 0.13$ | $0.49 \pm 0.13$ | $0.45 \pm 0.13$ |
| PGExplainer | $0.35 \pm 0.03$ | $0.40 \pm 0.09$ | $0.35 \pm 0.03$ | $0.39 \pm 0.09$ | $0.36 \pm 0.03$ | $0.38 \pm 0.06$ |
| SMEAtom | $0.31 \pm 0.03$ | $0.33 \pm 0.02$ | $0.32 \pm 0.02$ | $0.33 \pm 0.02$ | $0.33 \pm 0.02$ | $0.32 \pm 0.02$ |
| SMEBrics | $0.31 \pm 0.03$ | $0.33 \pm 0.02$ | $0.32 \pm 0.03$ | $0.32 \pm 0.02$ | $0.33 \pm 0.02$ | $0.32 \pm 0.03$ |
| SMEMurcko | $0.30 \pm 0.03$ | $0.33 \pm 0.02$ | $0.31 \pm 0.04$ | $0.32 \pm 0.02$ | $0.32 \pm 0.03$ | $0.32 \pm 0.02$ |
| PGIB | $0.30 \pm 0.05$ | $0.30 \pm 0.04$ | $0.31 \pm 0.05$ | $0.29 \pm 0.05$ | $0.32 \pm 0.04$ | $0.27 \pm 0.06$ |
| SEAL ($\lambda = 2$) | $0.52 \pm 0.02$ | $0.04 \pm 0.05$ | $0.57 \pm 0.03$ | $0.03 \pm 0.04$ | $0.66 \pm 0.03$ | $0.01 \pm 0.01$ |
| SEAL ($\lambda = 1$) | $0.50 \pm 0.02$ | $0.05 \pm 0.06$ | $0.55 \pm 0.02$ | $0.04 \pm 0.05$ | $0.64 \pm 0.03$ | $0.01 \pm 0.01$ |
| SEAL ($\lambda = 0.5$) | $0.50 \pm 0.02$ | $0.05 \pm 0.04$ | $0.56 \pm 0.01$ | $0.04 \pm 0.03$ | $0.64 \pm 0.02$ | $0.01 \pm 0.01$ |
| SEAL ($\lambda = 10^{-1}$) | $0.57 \pm 0.02$ | $\mathbf{0.00} \pm 0.00$ | $\mathbf{0.63} \pm 0.02$ | $\mathbf{0.00} \pm 0.00$ | $\mathbf{0.72} \pm 0.02$ | $\mathbf{0.00} \pm 0.00$ |
| SEAL ($\lambda = 10^{-2}$) | $0.53 \pm 0.02$ | $\mathbf{0.00} \pm 0.00$ | $0.60 \pm 0.02$ | $\mathbf{0.00} \pm 0.00$ | $0.69 \pm 0.02$ | $\mathbf{0.00} \pm 0.00$ |
| SEAL ($\lambda = 10^{-3}$) | $0.53 \pm 0.02$ | $0.06 \pm 0.04$ | $0.58 \pm 0.02$ | $0.05 \pm 0.04$ | $0.65 \pm 0.02$ | $0.04 \pm 0.03$ |
| SEAL ($\lambda = 10^{-4}$) | $0.52 \pm 0.02$ | $0.13 \pm 0.05$ | $0.58 \pm 0.02$ | $0.11 \pm 0.03$ | $0.63 \pm 0.03$ | $0.10 \pm 0.04$ |
| SEAL ($\lambda = 0$) | $0.52 \pm 0.00$ | $0.11 \pm 0.02$ | $0.59 \pm 0.01$ | $0.09 \pm 0.01$ | $0.65 \pm 0.01$ | $0.07 \pm 0.01$ |
| SEALAtom ($\lambda = 2$) | $0.43 \pm 0.08$ | $0.14 \pm 0.06$ | $0.44 \pm 0.08$ | $0.10 \pm 0.05$ | $0.44 \pm 0.08$ | $0.05 \pm 0.03$ |
| SEALAtom ($\lambda = 1$) | $0.34 \pm 0.09$ | $0.16 \pm 0.06$ | $0.35 \pm 0.09$ | $0.12 \pm 0.05$ | $0.35 \pm 0.09$ | $0.06 \pm 0.03$ |
| SEALAtom ($\lambda = 0.5$) | $0.38 \pm 0.09$ | $0.07 \pm 0.07$ | $0.39 \pm 0.10$ | $0.05 \pm 0.05$ | $0.38 \pm 0.09$ | $0.02 \pm 0.03$ |
| SEALAtom ($\lambda = 10^{-1}$) | $0.52 \pm 0.15$ | $0.02 \pm 0.01$ | $0.55 \pm 0.17$ | $0.01 \pm 0.01$ | $0.53 \pm 0.18$ | $0.00 \pm 0.00$ |
| SEALAtom ($\lambda = 10^{-2}$) | $0.35 \pm 0.03$ | $0.24 \pm 0.08$ | $0.35 \pm 0.03$ | $0.23 \pm 0.08$ | $0.34 \pm 0.03$ | $0.19 \pm 0.08$ |
| SEALAtom ($\lambda = 10^{-3}$) | $0.34 \pm 0.04$ | $0.24 \pm 0.07$ | $0.33 \pm 0.04$ | $0.23 \pm 0.07$ | $0.33 \pm 0.04$ | $0.21 \pm 0.07$ |
| SEALAtom ($\lambda = 10^{-4}$) | $0.44 \pm 0.05$ | $0.17 \pm 0.04$ | $0.44 \pm 0.05$ | $0.17 \pm 0.05$ | $0.44 \pm 0.06$ | $0.17 \pm 0.07$ |
| SEALAtom ($\lambda = 0$) | $0.57 \pm 0.04$ | $0.20 \pm 0.03$ | $0.58 \pm 0.04$ | $0.18 \pm 0.04$ | $0.57 \pm 0.04$ | $0.17 \pm 0.05$ |

Table 20: Performance of model explanations on real-world datasets (hERG). Explanations are evaluated using Fidelity metrics at 10%, 20%, and 30% masking thresholds, representing the proportion of most important atoms (nodes) either removed or retained during the evaluation.

| Model | Fidelity$_{10}+\uparrow$ | Fidelity$_{10}-\downarrow$ | Fidelity$_{20}+\uparrow$ | Fidelity$_{20}-\downarrow$ | Fidelity$_{30}+\uparrow$ | Fidelity$_{30}-\downarrow$ |
|---|---|---|---|---|---|---|
| Deconvolution | $0.44 \pm 0.04$ | $0.48 \pm 0.02$ | $0.48 \pm 0.02$ | $0.48 \pm 0.02$ | $0.49 \pm 0.02$ | $0.46 \pm 0.03$ |
| GuidedBackprop | $0.45 \pm 0.02$ | $0.48 \pm 0.02$ | $0.48 \pm 0.03$ | $0.48 \pm 0.02$ | $0.48 \pm 0.02$ | $0.47 \pm 0.02$ |
| IntegratedGradients | $0.55 \pm 0.13$ | $0.44 \pm 0.06$ | $0.58 \pm 0.18$ | $0.41 \pm 0.11$ | $0.59 \pm 0.19$ | $0.38 \pm 0.12$ |
| Saliency | $0.43 \pm 0.02$ | $0.48 \pm 0.02$ | $0.46 \pm 0.02$ | $0.47 \pm 0.03$ | $0.48 \pm 0.02$ | $0.47 \pm 0.02$ |
| InputXGradient | $0.40 \pm 0.04$ | $0.49 \pm 0.02$ | $0.43 \pm 0.04$ | $0.49 \pm 0.02$ | $0.46 \pm 0.03$ | $0.49 \pm 0.03$ |
| GNNExplainer | $0.46 \pm 0.03$ | $0.47 \pm 0.03$ | $0.50 \pm 0.06$ | $0.45 \pm 0.05$ | $0.52 \pm 0.07$ | $0.44 \pm 0.05$ |
| HIGNN | $0.34 \pm 0.04$ | $0.47 \pm 0.04$ | $0.41 \pm 0.06$ | $0.46 \pm 0.04$ | $0.45 \pm 0.05$ | $0.45 \pm 0.05$ |
| ProtGNN | $0.40 \pm 0.06$ | $0.42 \pm 0.07$ | $0.42 \pm 0.07$ | $0.42 \pm 0.07$ | $0.43 \pm 0.07$ | $0.42 \pm 0.07$ |
| PGExplainer | $0.35 \pm 0.04$ | $0.48 \pm 0.03$ | $0.41 \pm 0.04$ | $0.47 \pm 0.04$ | $0.45 \pm 0.03$ | $0.46 \pm 0.04$ |
| SMEAtom | $0.41 \pm 0.00$ | $0.52 \pm 0.04$ | $0.45 \pm 0.02$ | $0.51 \pm 0.04$ | $0.48 \pm 0.03$ | $0.51 \pm 0.03$ |
| SMEBrics | $0.38 \pm 0.02$ | $0.52 \pm 0.04$ | $0.45 \pm 0.03$ | $0.52 \pm 0.04$ | $0.49 \pm 0.04$ | $0.51 \pm 0.04$ |
| SMEMurcko | $0.38 \pm 0.02$ | $0.52 \pm 0.04$ | $0.44 \pm 0.03$ | $0.52 \pm 0.04$ | $0.49 \pm 0.04$ | $0.50 \pm 0.03$ |
| PGIB | $0.36 \pm 0.03$ | $0.43 \pm 0.07$ | $0.39 \pm 0.04$ | $0.41 \pm 0.07$ | $0.42 \pm 0.04$ | $0.38 \pm 0.07$ |
| SEAL ($\lambda = 2$) | $0.57 \pm 0.01$ | $\mathbf{0.00} \pm 0.00$ | $0.66 \pm 0.01$ | $\mathbf{0.00} \pm 0.00$ | $0.76 \pm 0.01$ | $\mathbf{0.00} \pm 0.00$ |
| SEAL ($\lambda = 1$) | $0.57 \pm 0.02$ | $\mathbf{0.00} \pm 0.00$ | $0.65 \pm 0.02$ | $\mathbf{0.00} \pm 0.00$ | $0.75 \pm 0.01$ | $\mathbf{0.00} \pm 0.00$ |
| SEAL ($\lambda = 0.5$) | $0.57 \pm 0.01$ | $\mathbf{0.00} \pm 0.00$ | $0.66 \pm 0.02$ | $\mathbf{0.00} \pm 0.00$ | $0.76 \pm 0.01$ | $\mathbf{0.00} \pm 0.00$ |
| SEAL ($\lambda = 10^{-1}$) | $0.59 \pm 0.01$ | $\mathbf{0.00} \pm 0.00$ | $0.68 \pm 0.01$ | $\mathbf{0.00} \pm 0.00$ | $0.77 \pm 0.01$ | $\mathbf{0.00} \pm 0.00$ |
| SEAL ($\lambda = 10^{-2}$) | $0.59 \pm 0.01$ | $\mathbf{0.00} \pm 0.00$ | $0.68 \pm 0.01$ | $\mathbf{0.00} \pm 0.00$ | $0.78 \pm 0.01$ | $\mathbf{0.00} \pm 0.00$ |
| SEAL ($\lambda = 10^{-3}$) | $0.63 \pm 0.03$ | $0.02 \pm 0.01$ | $0.72 \pm 0.03$ | $0.01 \pm 0.01$ | $0.80 \pm 0.03$ | $0.01 \pm 0.01$ |
| SEAL ($\lambda = 10^{-4}$) | $0.63 \pm 0.01$ | $0.09 \pm 0.03$ | $0.71 \pm 0.01$ | $0.07 \pm 0.02$ | $0.78 \pm 0.01$ | $0.05 \pm 0.02$ |
| SEAL ($\lambda = 0$) | $0.67 \pm 0.02$ | $0.15 \pm 0.02$ | $0.74 \pm 0.02$ | $0.15 \pm 0.01$ | $0.78 \pm 0.03$ | $0.14 \pm 0.01$ |
| SEALAtom ($\lambda = 2$) | $0.78 \pm 0.01$ | $0.01 \pm 0.01$ | $0.86 \pm 0.01$ | $\mathbf{0.00} \pm 0.00$ | $0.87 \pm 0.03$ | $\mathbf{0.00} \pm 0.00$ |
| SEALAtom ($\lambda = 1$) | $0.78 \pm 0.03$ | $0.01 \pm 0.01$ | $0.87 \pm 0.02$ | $\mathbf{0.00} \pm 0.00$ | $0.89 \pm 0.01$ | $\mathbf{0.00} \pm 0.00$ |
| SEALAtom ($\lambda = 0.5$) | $0.77 \pm 0.03$ | $0.05 \pm 0.03$ | $0.86 \pm 0.02$ | $\mathbf{0.00} \pm 0.00$ | $0.85 \pm 0.03$ | $\mathbf{0.00} \pm 0.00$ |
| SEALAtom ($\lambda = 10^{-1}$) | $0.78 \pm 0.01$ | $0.01 \pm 0.01$ | $0.87 \pm 0.01$ | $\mathbf{0.00} \pm 0.00$ | $0.89 \pm 0.01$ | $\mathbf{0.00} \pm 0.00$ |
| SEALAtom ($\lambda = 10^{-2}$) | $0.77 \pm 0.01$ | $0.00 \pm 0.00$ | $0.83 \pm 0.01$ | $\mathbf{0.00} \pm 0.00$ | $0.87 \pm 0.01$ | $\mathbf{0.00} \pm 0.00$ |
| SEALAtom ($\lambda = 10^{-3}$) | $\mathbf{0.83} \pm 0.02$ | $0.02 \pm 0.02$ | $\mathbf{0.89} \pm 0.03$ | $0.01 \pm 0.00$ | $\mathbf{0.91} \pm 0.03$ | $0.01 \pm 0.01$ |
| SEALAtom ($\lambda = 10^{-4}$) | $0.55 \pm 0.08$ | $0.57 \pm 0.15$ | $0.54 \pm 0.10$ | $0.58 \pm 0.15$ | $0.50 \pm 0.12$ | $0.54 \pm 0.13$ |
| SEALAtom ($\lambda = 0$) | $0.73 \pm 0.02$ | $0.23 \pm 0.16$ | $0.74 \pm 0.02$ | $0.28 \pm 0.14$ | $0.71 \pm 0.05$ | $0.30 \pm 0.12$ |

Table 21: Performance of model explanations on real-world datasets (Solubility). Explanations are evaluated using Fidelity metrics at 10%, 20%, and 30% masking thresholds, representing the proportion of most important atoms (nodes) either removed or retained during the evaluation.

| | Model | $\text{Fidelity}_{10}+\uparrow$ | $\text{Fidelity}_{10}-\downarrow$ | $\text{Fidelity}_{20}+\uparrow$ | $\text{Fidelity}_{20}-\downarrow$ | $\text{Fidelity}_{30}+\uparrow$ | $\text{Fidelity}_{30}-\downarrow$ |
|---|---|---|---|---|---|---|---|
| Solubility | Deconvolution | $2.56 \pm 0.87$ | $4.09 \pm 1.17$ | $2.82 \pm 0.94$ | $3.96 \pm 1.13$ | $3.20 \pm 1.03$ | $3.79 \pm 1.08$ |
| | GuidedBackprop | $\mathbf{3.77} \pm 1.10$ | $3.30 \pm 1.11$ | $\mathbf{4.02} \pm 1.19$ | $3.09 \pm 1.04$ | $\mathbf{4.24} \pm 1.29$ | $2.80 \pm 0.96$ |
| | IntegratedGradients | $2.33 \pm 0.80$ | $4.62 \pm 1.58$ | $2.56 \pm 0.87$ | $4.54 \pm 1.56$ | $2.88 \pm 0.96$ | $4.39 \pm 1.50$ |
| | Saliency | $3.22 \pm 0.99$ | $3.68 \pm 1.20$ | $3.46 \pm 1.07$ | $3.50 \pm 1.15$ | $3.75 \pm 1.15$ | $3.23 \pm 1.06$ |
| | InputXGradient | $3.14 \pm 0.93$ | $3.50 \pm 1.06$ | $3.41 \pm 1.02$ | $3.33 \pm 1.00$ | $3.74 \pm 1.14$ | $3.09 \pm 0.93$ |
| | GNNExplainer | $3.56 \pm 1.24$ | $3.64 \pm 0.99$ | $3.90 \pm 1.34$ | $3.47 \pm 0.93$ | $4.25 \pm 1.48$ | $3.24 \pm 0.85$ |
| | HIGNN | $0.46 \pm 0.08$ | $0.43 \pm 0.09$ | $0.49 \pm 0.08$ | $0.40 \pm 0.09$ | $0.53 \pm 0.09$ | $0.36 \pm 0.08$ |
| | PGExplainer | $3.21 \pm 1.05$ | $3.49 \pm 0.97$ | $3.44 \pm 1.13$ | $3.29 \pm 0.90$ | $3.72 \pm 1.20$ | $3.02 \pm 0.81$ |
| | SMEAtom | $1.01 \pm 0.65$ | $1.18 \pm 1.05$ | $1.07 \pm 0.70$ | $1.11 \pm 0.97$ | $1.17 \pm 0.85$ | $0.99 \pm 0.79$ |
| | SMEBrics | $0.96 \pm 0.57$ | $1.01 \pm 0.69$ | $1.02 \pm 0.60$ | $0.95 \pm 0.60$ | $1.09 \pm 0.69$ | $0.87 \pm 0.50$ |
| | SMEMurcko | $0.98 \pm 0.59$ | $1.09 \pm 0.85$ | $1.04 \pm 0.63$ | $1.01 \pm 0.75$ | $1.12 \pm 0.74$ | $0.91 \pm 0.60$ |
| | SEAL ($\lambda = 2$) | $1.12 \pm 0.25$ | $1.04 \pm 0.42$ | $1.20 \pm 0.28$ | $0.96 \pm 0.38$ | $1.34 \pm 0.32$ | $0.84 \pm 0.33$ |
| | SEAL ($\lambda = 1$) | $1.04 \pm 0.31$ | $0.83 \pm 0.39$ | $1.11 \pm 0.34$ | $0.78 \pm 0.36$ | $1.22 \pm 0.38$ | $0.69 \pm 0.31$ |
| | SEAL ($\lambda = 0.5$) | $1.08 \pm 0.31$ | $0.89 \pm 0.41$ | $1.15 \pm 0.33$ | $0.83 \pm 0.38$ | $1.26 \pm 0.37$ | $0.74 \pm 0.34$ |
| | SEAL ($\lambda = 10^{-1}$) | $0.77 \pm 0.20$ | $0.67 \pm 0.29$ | $0.83 \pm 0.23$ | $0.62 \pm 0.26$ | $0.92 \pm 0.27$ | $0.55 \pm 0.23$ |
| | SEAL ($\lambda = 10^{-2}$) | $0.64 \pm 0.12$ | $0.50 \pm 0.13$ | $0.68 \pm 0.13$ | $0.49 \pm 0.13$ | $0.73 \pm 0.14$ | $0.46 \pm 0.13$ |
| | SEAL ($\lambda = 10^{-3}$) | $1.24 \pm 0.20$ | $1.26 \pm 0.33$ | $1.37 \pm 0.21$ | $1.14 \pm 0.30$ | $1.55 \pm 0.25$ | $0.98 \pm 0.25$ |
| | SEAL ($\lambda = 10^{-4}$) | $0.78 \pm 0.14$ | $0.64 \pm 0.16$ | $0.84 \pm 0.15$ | $0.58 \pm 0.13$ | $0.91 \pm 0.17$ | $0.52 \pm 0.09$ |
| | SEAL ($\lambda = 0$) | $0.49 \pm 0.03$ | $0.55 \pm 0.07$ | $0.54 \pm 0.04$ | $0.53 \pm 0.08$ | $0.58 \pm 0.05$ | $0.48 \pm 0.07$ |
| | SEALAtom ($\lambda = 2$) | $0.41 \pm 0.04$ | $0.65 \pm 0.15$ | $0.44 \pm 0.04$ | $0.63 \pm 0.15$ | $0.46 \pm 0.05$ | $0.58 \pm 0.13$ |
| | SEALAtom ($\lambda = 1$) | $0.38 \pm 0.02$ | $0.56 \pm 0.14$ | $0.41 \pm 0.02$ | $0.54 \pm 0.13$ | $0.43 \pm 0.03$ | $0.50 \pm 0.12$ |
| | SEALAtom ($\lambda = 0.5$) | $0.39 \pm 0.03$ | $0.33 \pm 0.11$ | $0.41 \pm 0.04$ | $0.32 \pm 0.12$ | $0.43 \pm 0.04$ | $0.30 \pm 0.12$ |
| | SEALAtom ($\lambda = 10^{-1}$) | $0.37 \pm 0.03$ | $\mathbf{0.28} \pm 0.10$ | $0.40 \pm 0.04$ | $\mathbf{0.25} \pm 0.11$ | $0.41 \pm 0.05$ | $\mathbf{0.22} \pm 0.10$ |
| | SEALAtom ($\lambda = 10^{-2}$) | $0.86 \pm 0.34$ | $1.19 \pm 0.53$ | $0.93 \pm 0.36$ | $1.03 \pm 0.45$ | $1.01 \pm 0.41$ | $0.83 \pm 0.34$ |
| | SEALAtom ($\lambda = 10^{-3}$) | $0.80 \pm 0.42$ | $0.81 \pm 0.48$ | $0.86 \pm 0.46$ | $0.70 \pm 0.38$ | $0.93 \pm 0.55$ | $0.57 \pm 0.27$ |
| | SEALAtom ($\lambda = 10^{-4}$) | $0.97 \pm 0.37$ | $0.95 \pm 0.46$ | $1.04 \pm 0.41$ | $0.82 \pm 0.34$ | $1.15 \pm 0.48$ | $0.68 \pm 0.22$ |
| | SEALAtom ($\lambda = 0$) | $0.69 \pm 0.10$ | $0.54 \pm 0.10$ | $0.74 \pm 0.11$ | $0.51 \pm 0.08$ | $0.79 \pm 0.12$ | $0.47 \pm 0.06$ |

Table 22: Performance of model explanations on real-world datasets (MUTAG). Explanations are evaluated using Subgraph Explanation (SE) and Null Explanation (NE) metrics.

| | Model | SE ↑ | NE ↑ |
|---|---|---|---|
| | Deconvolution | $0.84 \pm 0.01$ | $\mathbf{0.77} \pm 0.02$ |
| | GuidedBackprop | $0.38 \pm 0.11$ | $0.60 \pm 0.07$ |
| | IntegratedGradients | $0.56 \pm 0.18$ | $0.53 \pm 0.03$ |
| | Saliency | $0.48 \pm 0.07$ | $0.63 \pm 0.03$ |
| | InputXGradient | $0.45 \pm 0.06$ | $0.63 \pm 0.06$ |
| | GNNExplainer | $0.48 \pm 0.03$ | $0.72 \pm 0.02$ |
| | ProtGNN | $0.47 \pm 0.06$ | $0.65 \pm 0.03$ |
| | HiGNN | $0.55 \pm 0.00$ | $0.70 \pm 0.02$ |
| | PGExplainer | $0.29 \pm 0.08$ | $0.68 \pm 0.03$ |
| | SMEAtom | $0.76 \pm 0.04$ | $0.36 \pm 0.07$ |
| | SMEBrics | $0.55 \pm 0.00$ | $\mathbf{0.77} \pm 0.01$ |
| MUTAG | SMEMurcko | $0.47 \pm 0.03$ | $0.60 \pm 0.03$ |
| | PGIB | $0.46 \pm 0.05$ | $0.50 \pm 0.04$ |
| | SEAL ($\lambda = 2$) | $0.85 \pm 0.02$ | $0.54 \pm 0.02$ |
| | SEAL ($\lambda = 1$) | $0.85 \pm 0.02$ | $0.55 \pm 0.01$ |
| | SEAL ($\lambda = 0.5$) | $0.85 \pm 0.02$ | $0.54 \pm 0.01$ |
| | SEAL ($\lambda = 10^{-1}$) | $0.84 \pm 0.02$ | $0.55 \pm 0.01$ |
| | SEAL ($\lambda = 10^{-2}$) | $0.87 \pm 0.02$ | $0.53 \pm 0.02$ |
| | SEAL ($\lambda = 10^{-3}$) | $0.88 \pm 0.01$ | $0.52 \pm 0.02$ |
| | SEAL ($\lambda = 10^{-4}$) | $0.90 \pm 0.01$ | $0.53 \pm 0.02$ |
| | SEAL ($\lambda = 0$) | $0.91 \pm 0.01$ | $0.51 \pm 0.02$ |
| | SEALAtom ($\lambda = 2$) | $0.65 \pm 0.02$ | $0.24 \pm 0.02$ |
| | SEALAtom ($\lambda = 1$) | $0.65 \pm 0.02$ | $0.24 \pm 0.02$ |
| | SEALAtom ($\lambda = 0.5$) | $0.65 \pm 0.01$ | $0.24 \pm 0.02$ |
| | SEALAtom ($\lambda = 10^{-1}$) | $0.65 \pm 0.01$ | $0.25 \pm 0.02$ |
| | SEALAtom ($\lambda = 10^{-2}$) | $\mathbf{0.97} \pm 0.00$ | $0.27 \pm 0.02$ |
| | SEALAtom ($\lambda = 10^{-3}$) | $\mathbf{0.97} \pm 0.00$ | $0.25 \pm 0.03$ |
| | SEALAtom ($\lambda = 10^{-4}$) | $0.86 \pm 0.02$ | $0.42 \pm 0.02$ |
| | SEALAtom ($\lambda = 0$) | $0.71 \pm 0.05$ | $0.47 \pm 0.03$ |

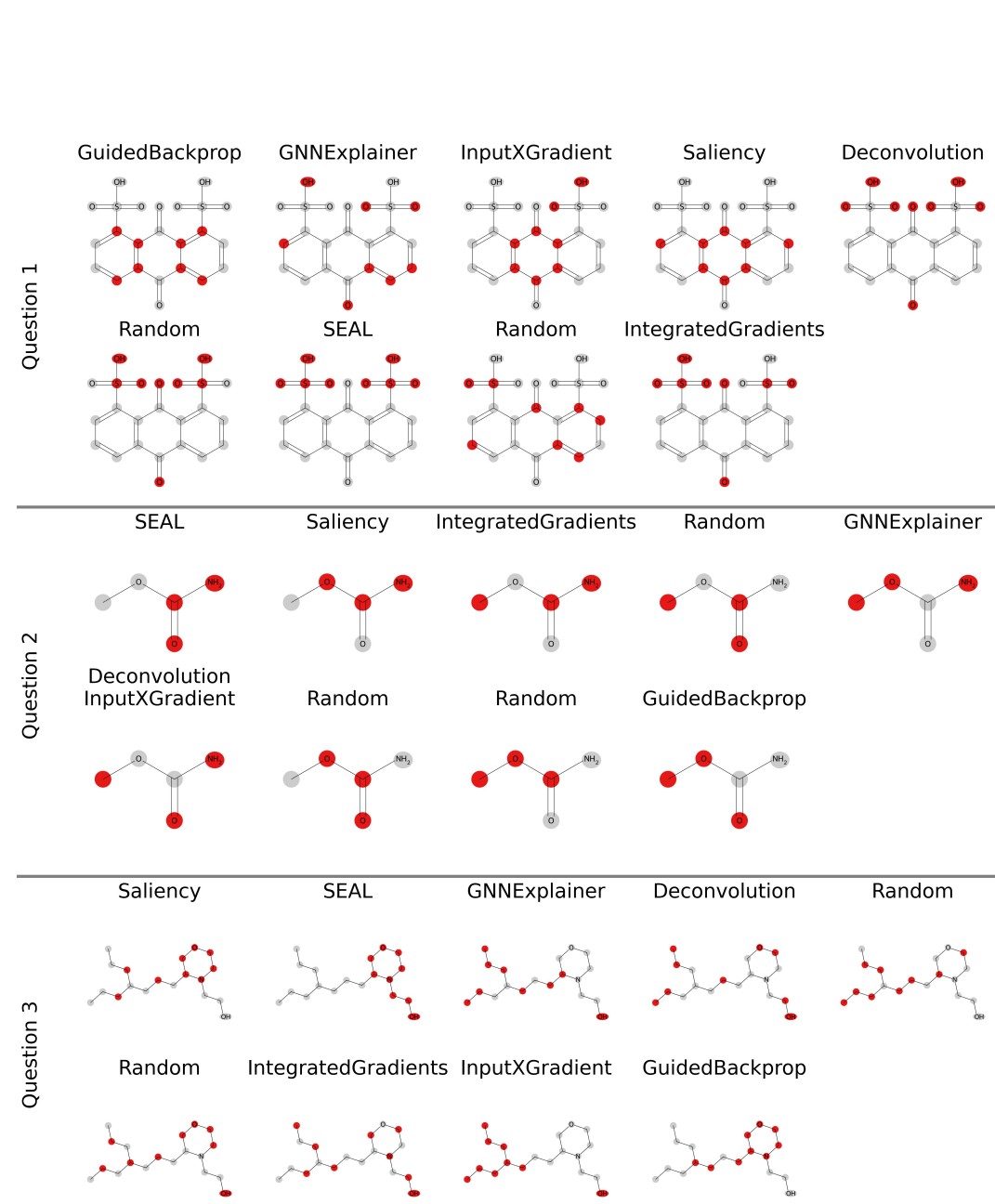

Figure 13: Node-level explanation examples from user study, The red color indicates that the highlighted atoms had a positive contribution to the compound's solubility.

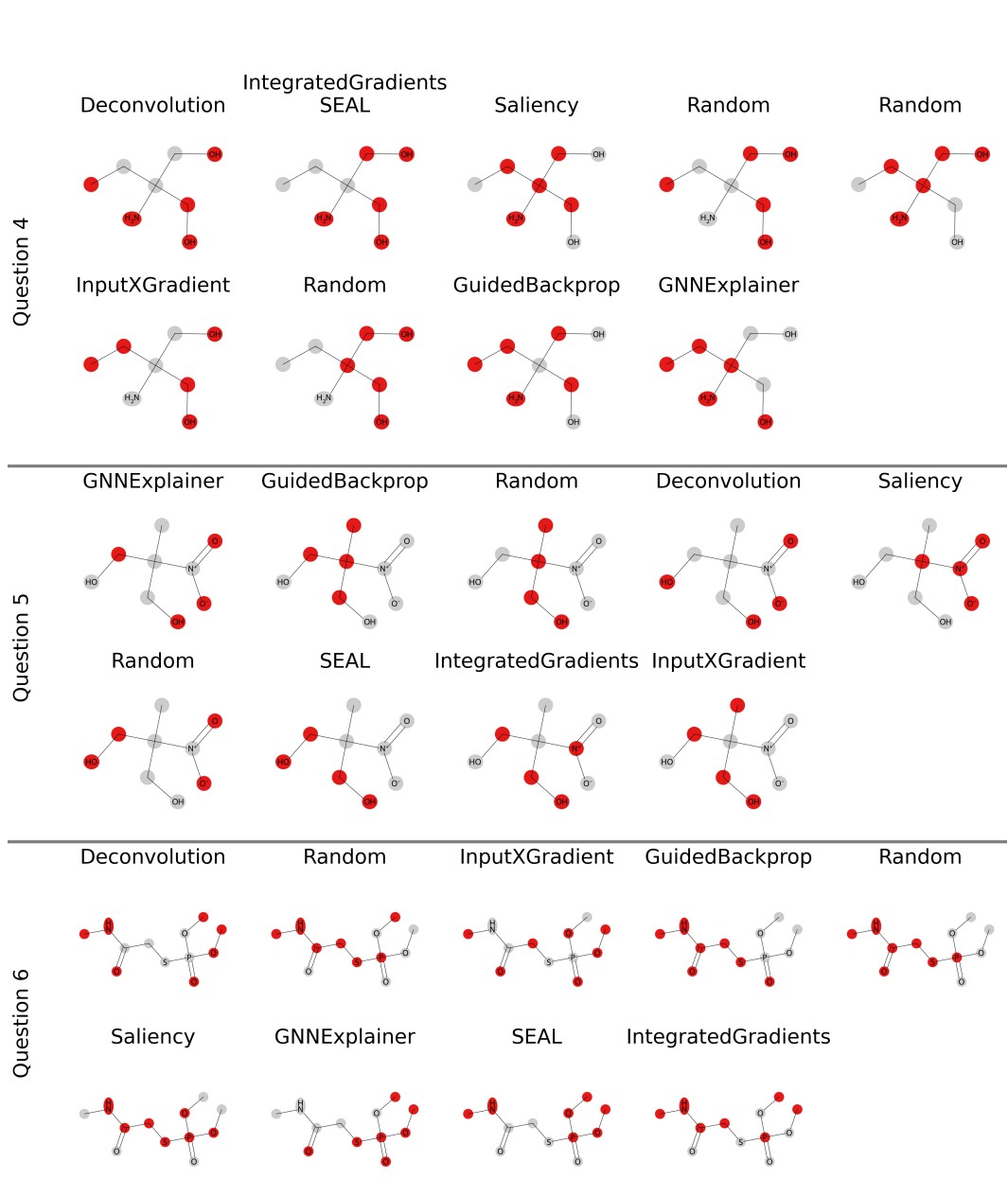

Figure 14: Node-level explanation examples from user study, The red color indicates that the highlighted atoms had a positive contribution to the compound's solubility.

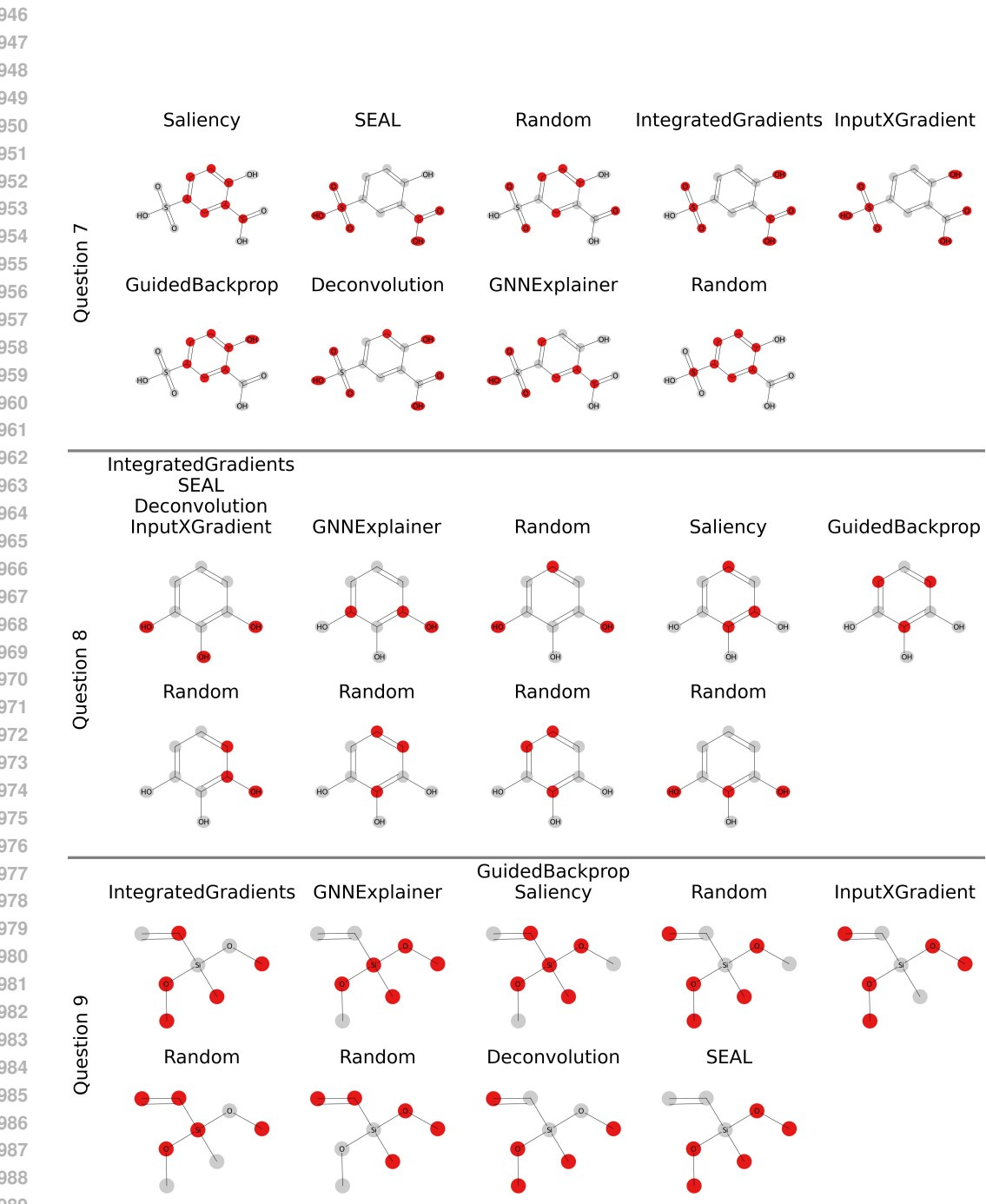

Figure 15: Node-level explanation examples from user study, The red color indicates that the highlighted atoms had a positive contribution to the compound's solubility.

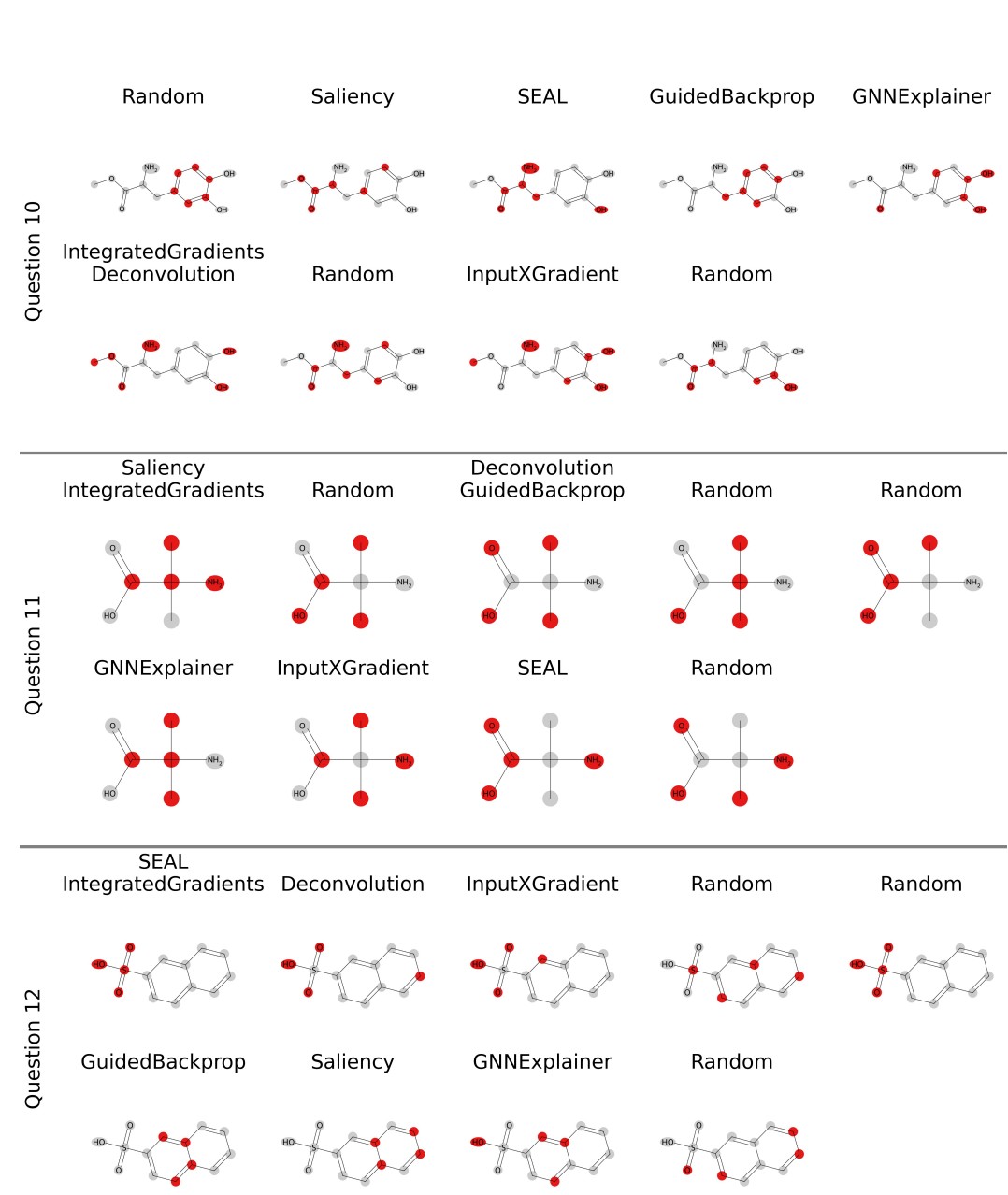

Figure 16: Node-level explanation examples from user study, The red color indicates that the highlighted atoms had a positive contribution to the compound's solubility.

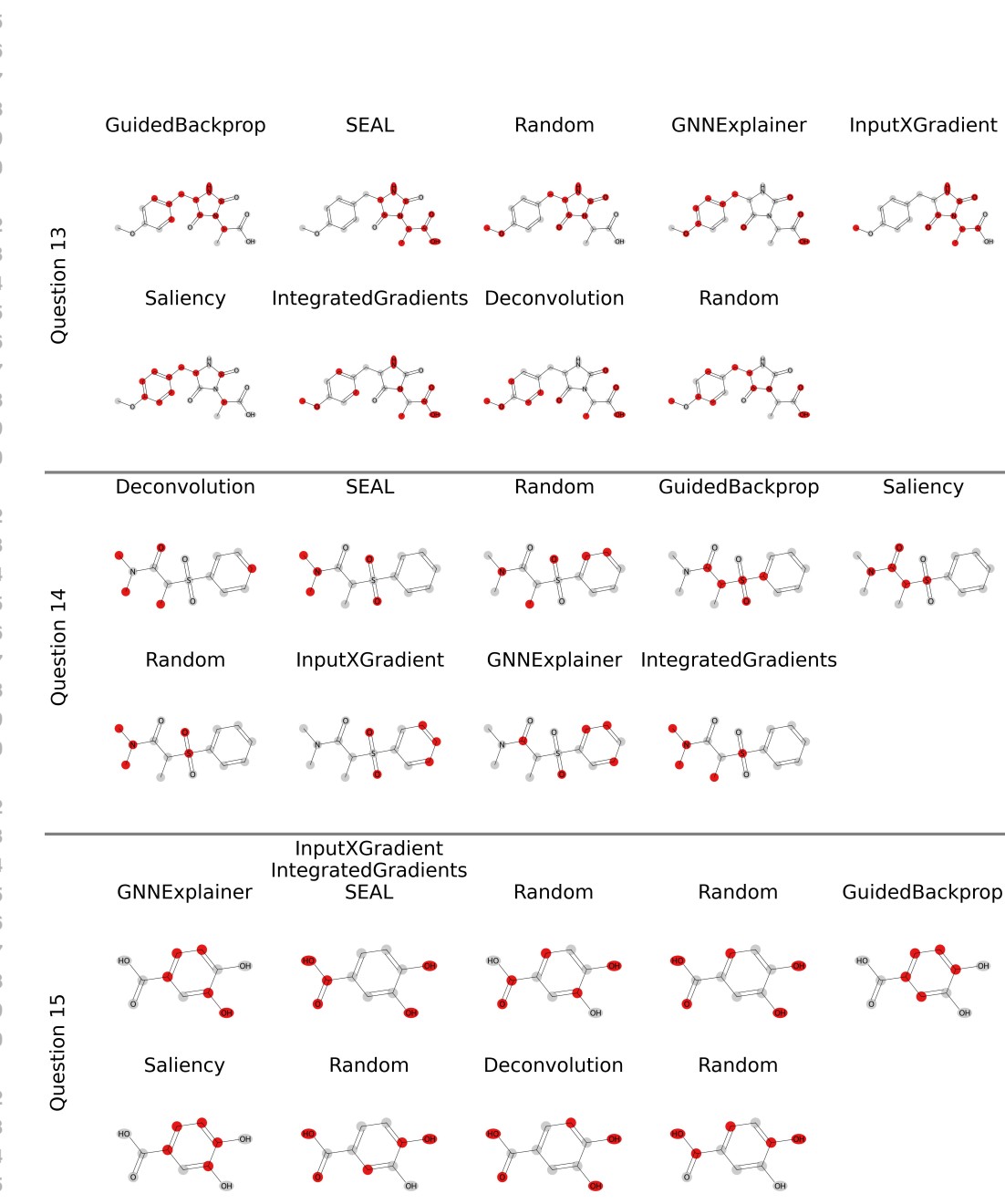

Figure 17: Node-level explanation examples from user study, The red color indicates that the highlighted atoms had a positive contribution to the compound's solubility.

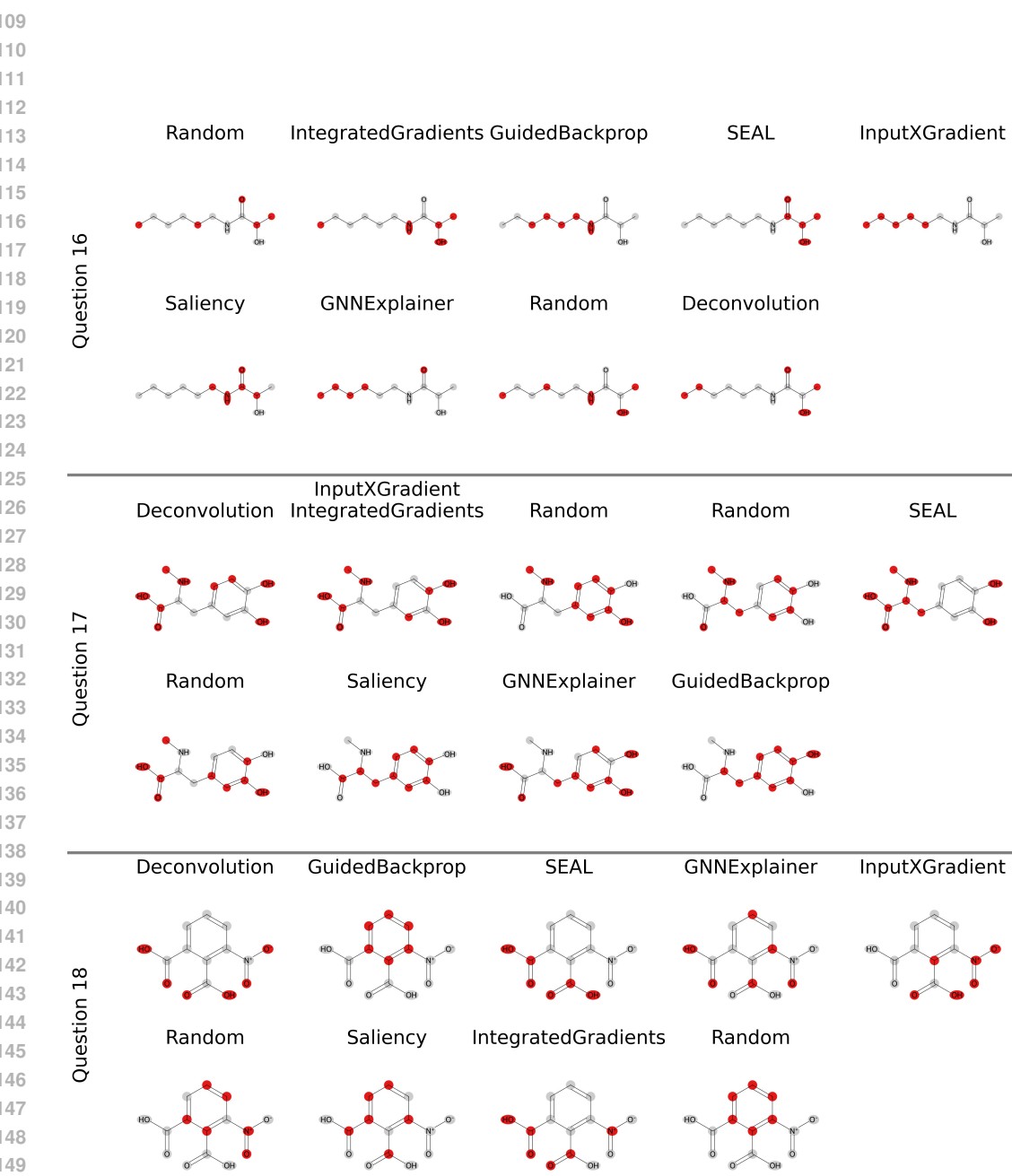

Figure 18: Node-level explanation examples from user study, The red color indicates that the high-lighted atoms had a positive contribution to the compound's solubility.

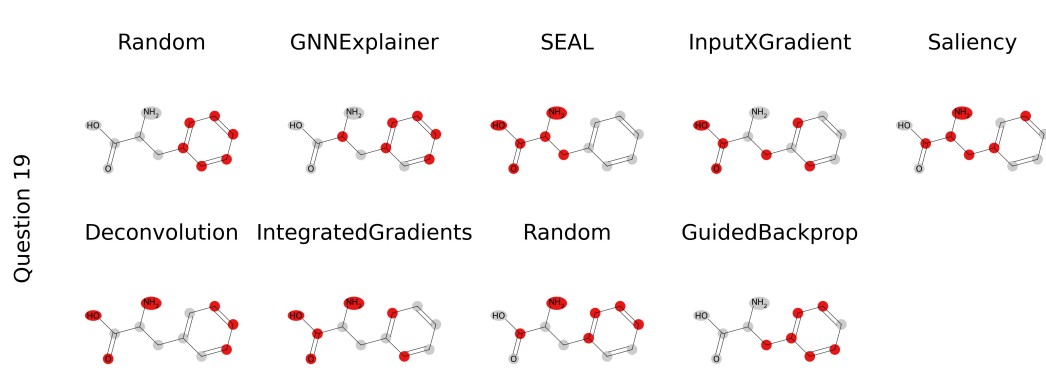

Figure 19: Node-level explanation examples from user study, The red color indicates that the highlighted atoms had a positive contribution to the compound's solubility.

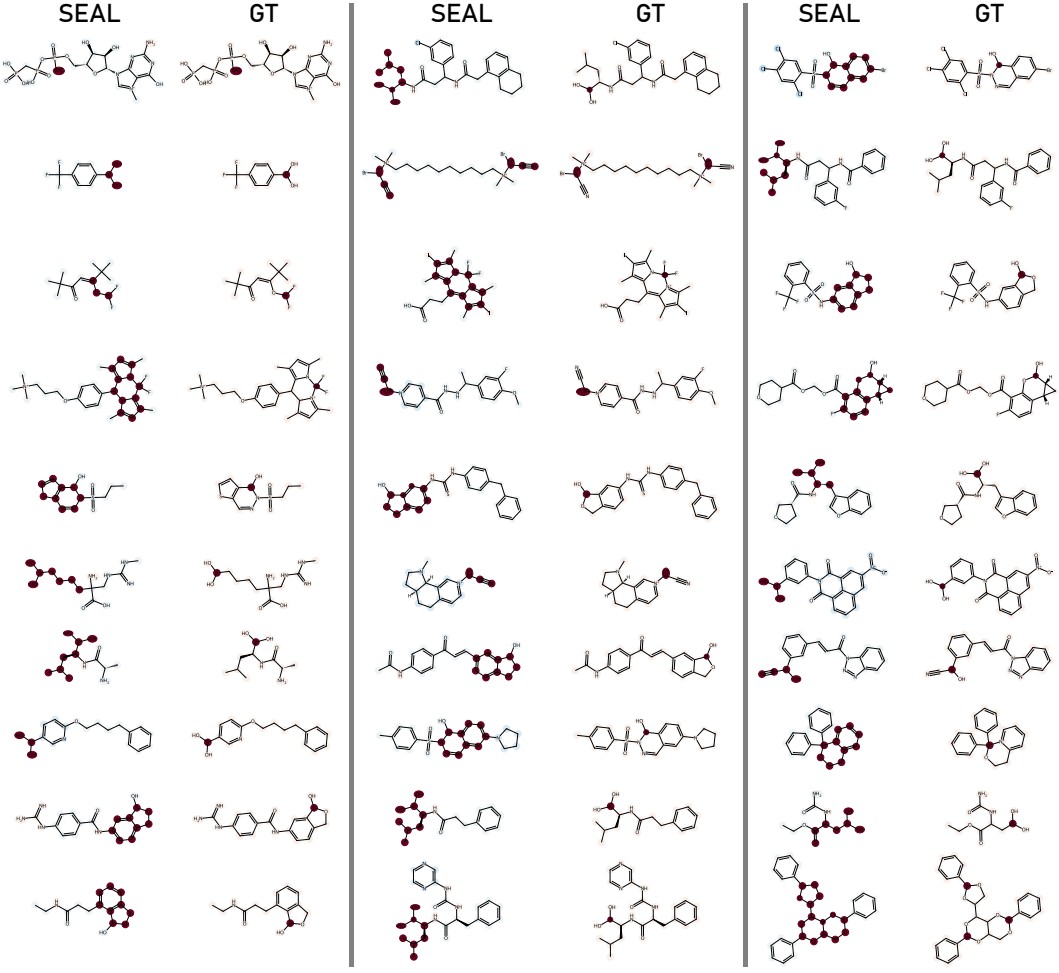

Figure 20: Node-level explanation examples of the SEAL method and Ground-Truth evaluated on the Boron (B) task for the positive target class. The red color indicates that the highlighted atoms had a positive contribution to the compound's positive prediction. Blue as a negative contribution.

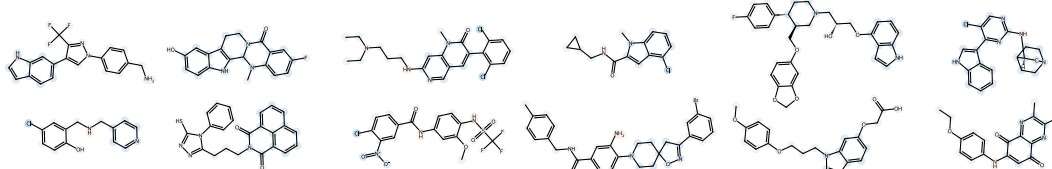

Figure 21: Node-level explanation examples of the SEAL method evaluated on the Boron (B) task for the negative target class. The red color indicates that the highlighted atoms had a positive contribution to the compound's positive prediction. Blue as a negative contribution.

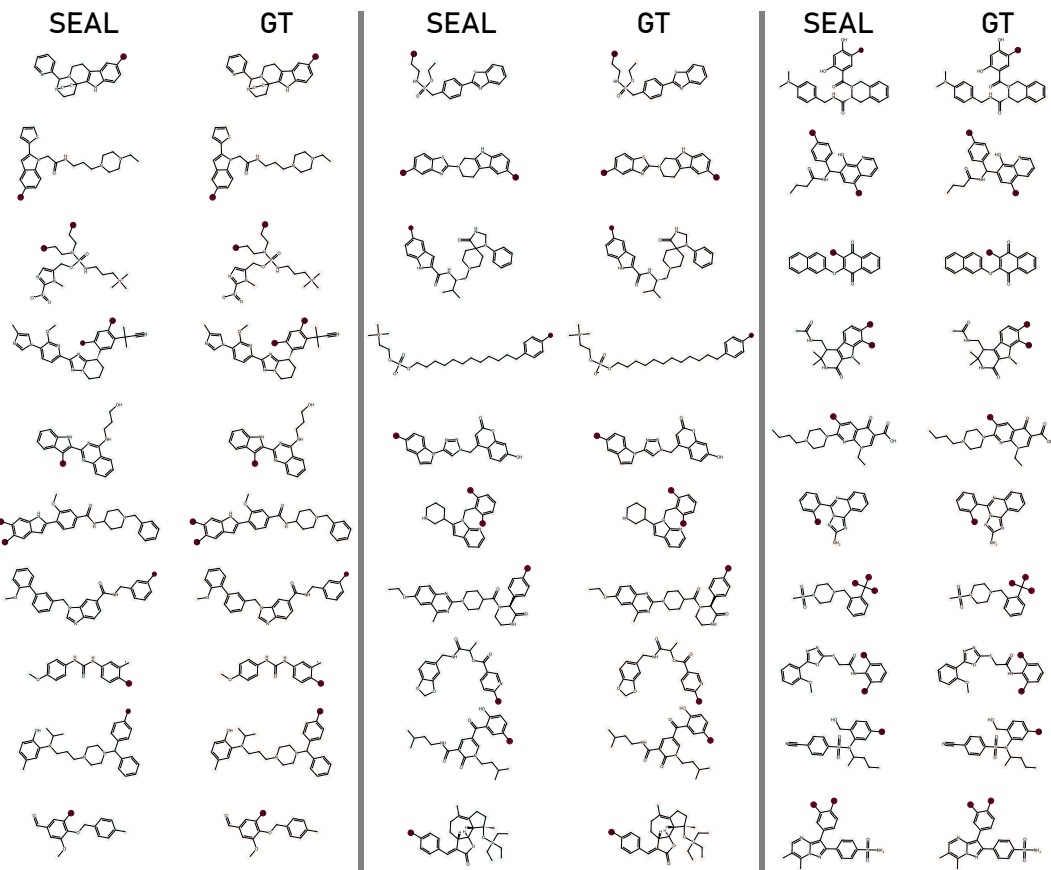

Figure 22: Node-level explanation examples of the SEAL method and Ground-Truth evaluated on Halogens (X) task for the positive target class. The red color indicates that the highlighted atoms had a positive contribution to the compound's positive prediction. Blue as a negative contribution.

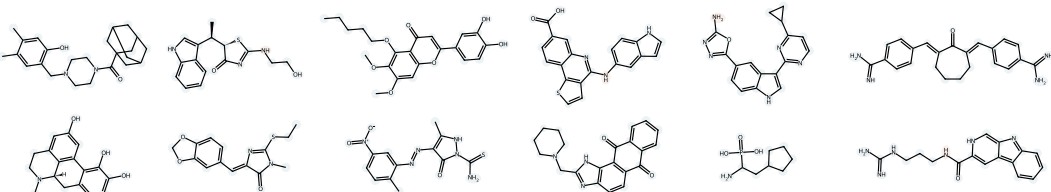

Figure 23: Node-level explanation examples of the SEAL method evaluated on the Halogens (X) task for the negative target class. The red color indicates that the highlighted atoms had a positive contribution to the compound's positive prediction. Blue as a negative contribution.

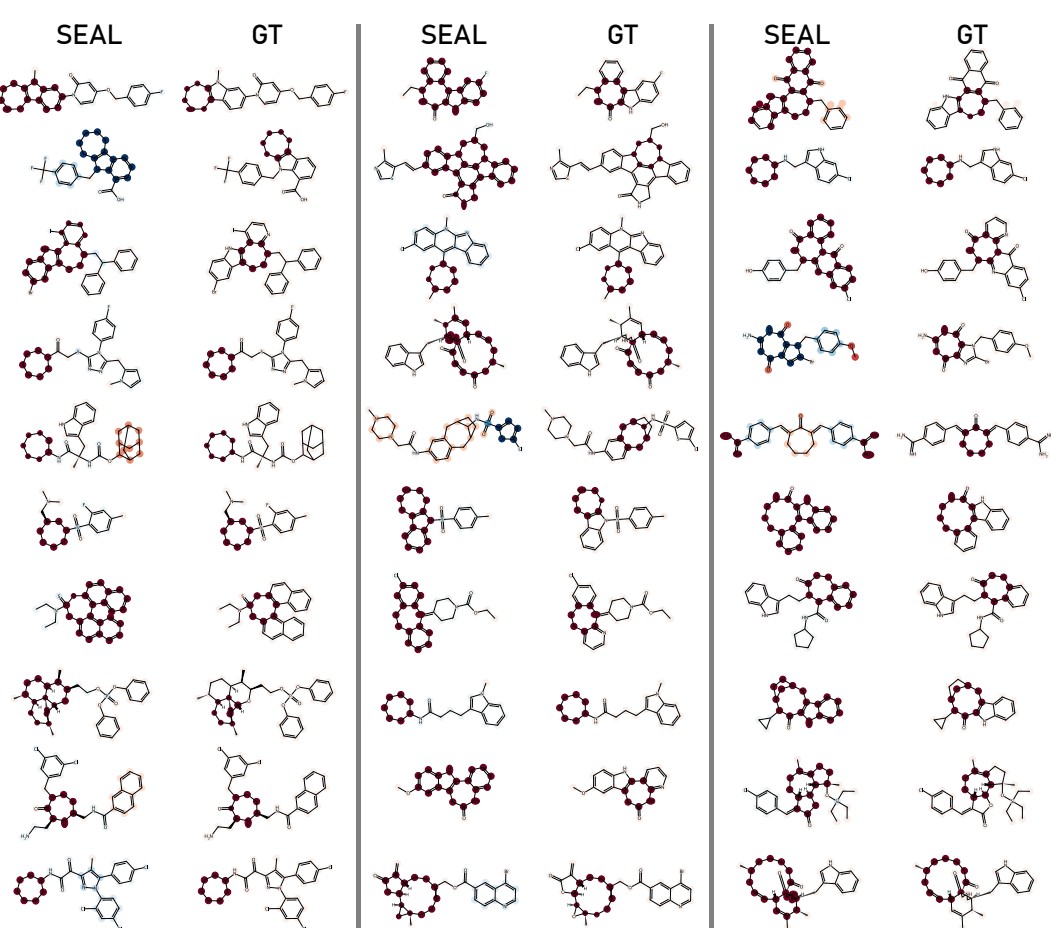

Figure 24: Node-level explanation examples of the SEAL method and Ground-Truth evaluated on the rings-max task for the positive target class. The red color indicates that the highlighted atoms had a positive contribution to the compound's positive prediction. Blue as a negative contribution.

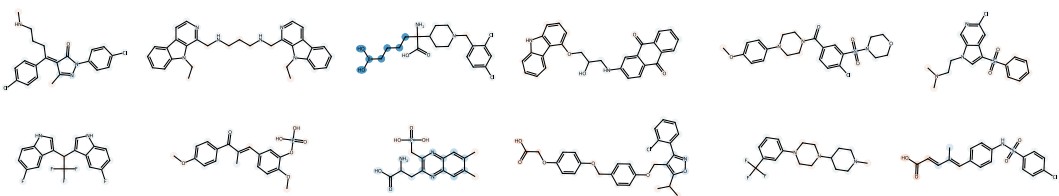

Figure 25: Node-level explanation examples of the SEAL method evaluated on the rings-max task for the negative target class. The red color indicates that the highlighted atoms had a positive contribution to the compound's positive prediction. Blue as a negative contribution.

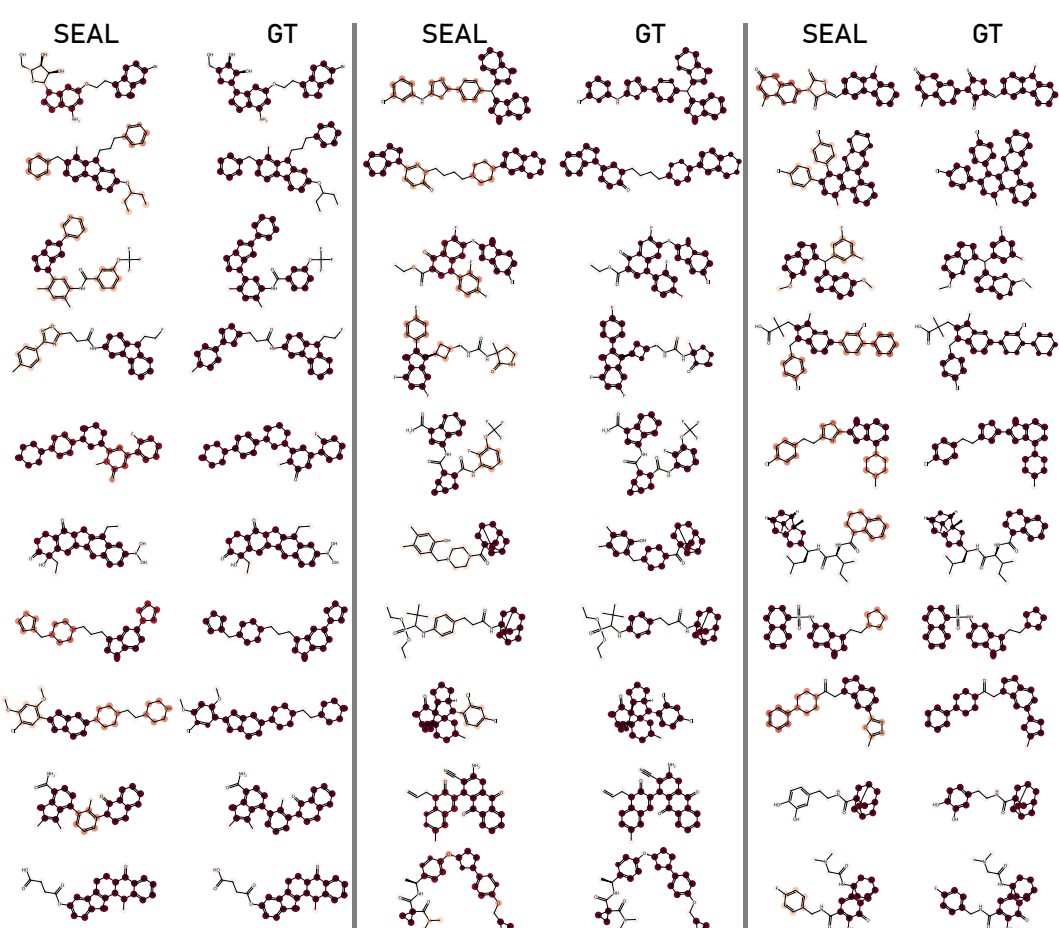

Figure 26: Node-level explanation examples of the SEAL method and Ground-Truth evaluated on the rings-count task for the positive target class. The red color indicates that the highlighted atoms had a positive contribution to the compound's positive prediction. Blue as a negative contribution.

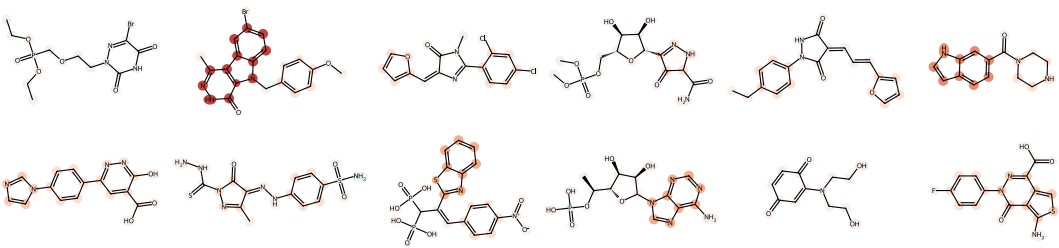

Figure 27: Node-level explanation examples of the SEAL method evaluated on the rings-count task for the negative target class. The red color indicates that the highlighted atoms had a positive contribution to the compound's positive prediction. Blue as a negative contribution.

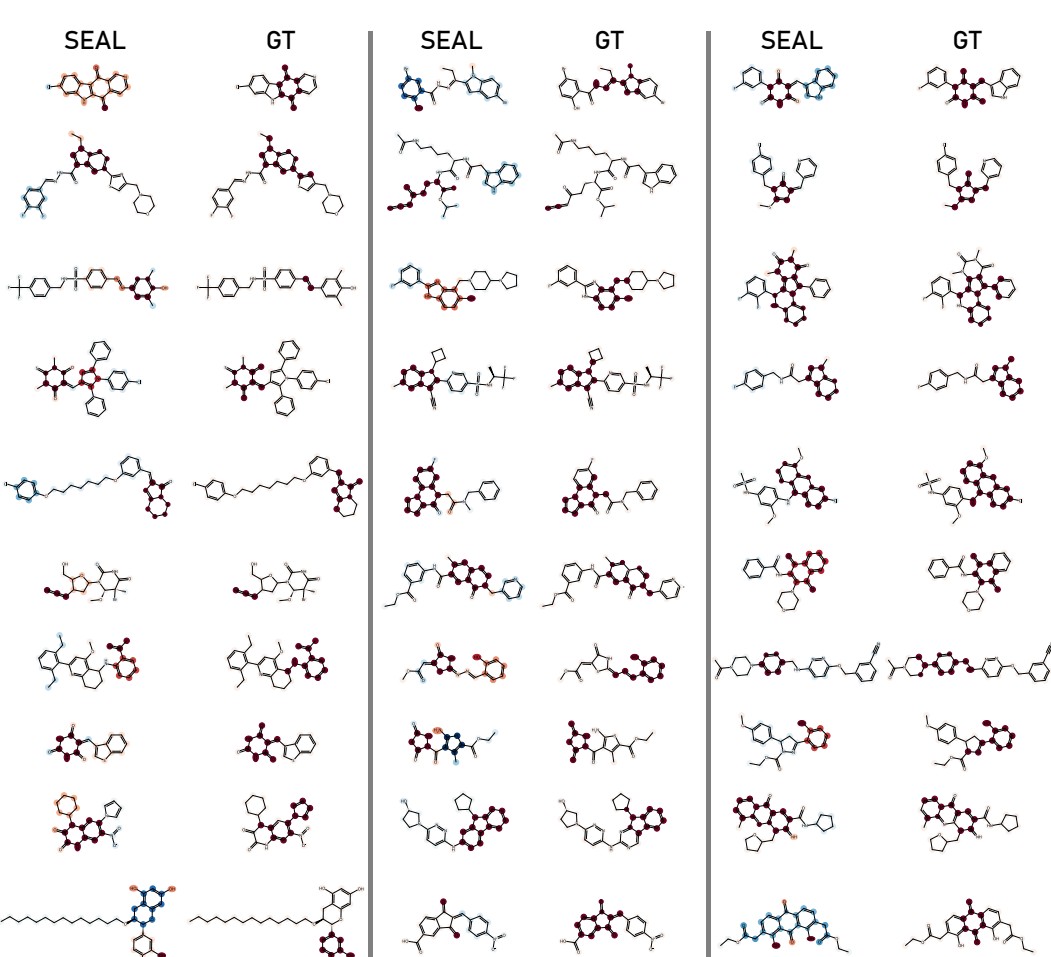

Figure 28: Node-level explanation examples of the SEAL method and Ground-Truth evaluated on the PAINS task for the positive target class. The red color indicates that the highlighted atoms had a positive contribution to the compound's positive prediction. Blue as a negative contribution.

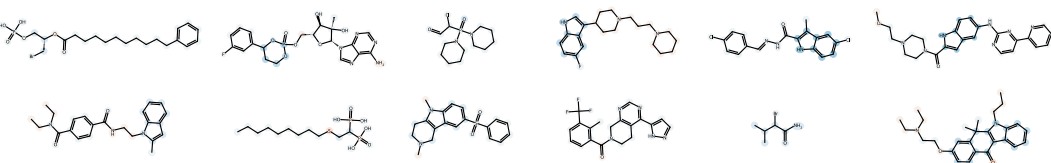

Figure 29: Node-level explanation examples of the SEAL method evaluated on the PAINS task for the negative target class. The red color indicates that the highlighted atoms had a positive contribution to the compound's positive prediction. Blue as a negative contribution.

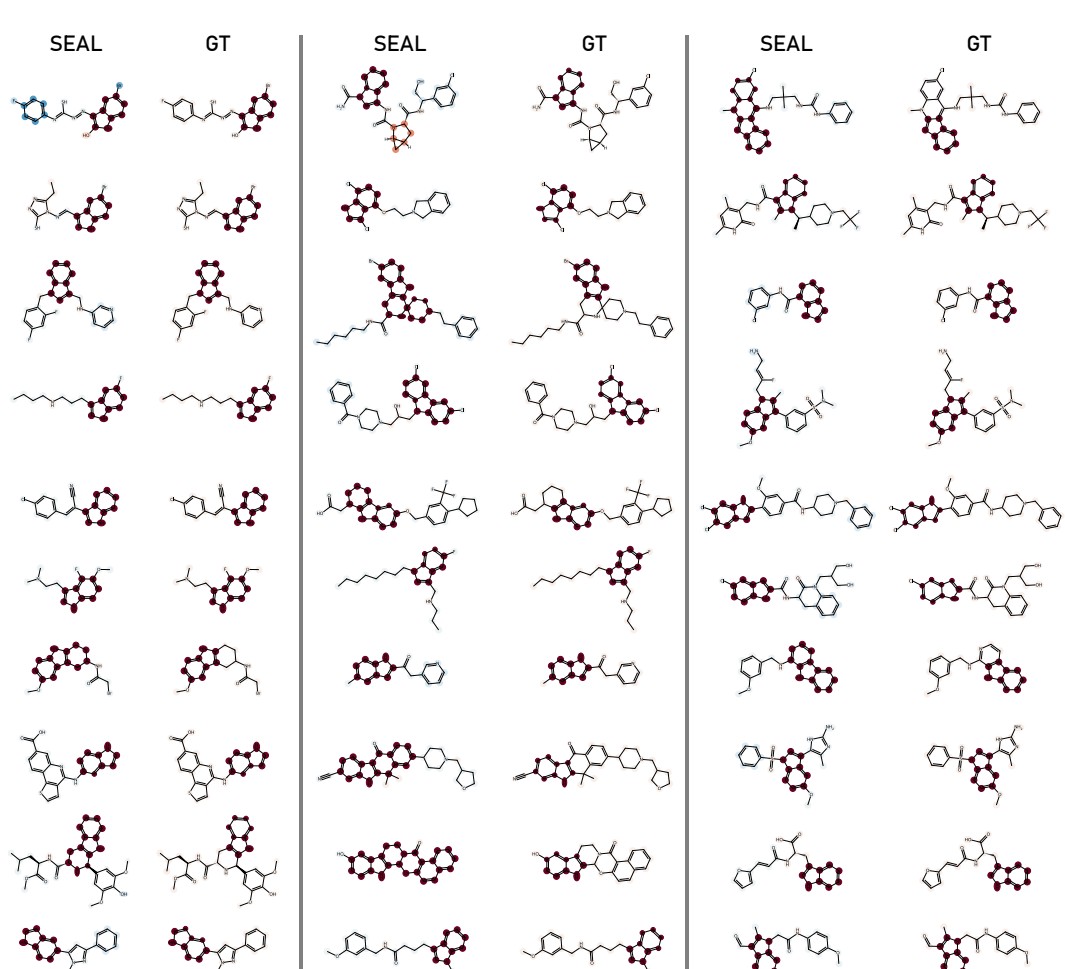

Figure 30: Node-level explanation examples of the SEAL method and Ground-Truth evaluated on the indole task for the positive target class. The red color indicates that the highlighted atoms had a positive contribution to the compound's positive prediction. Blue as a negative contribution.

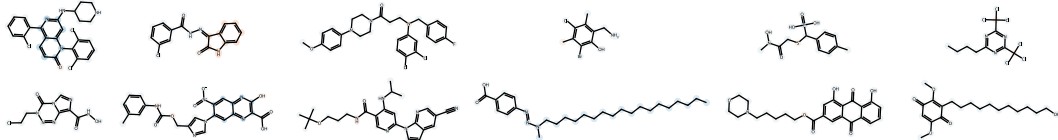

Figure 31: Node-level explanation examples of the SEAL method evaluated on the indole task for the negative target class. The red color indicates that the highlighted atoms had a positive contribution to the compound's positive prediction. Blue as a negative contribution.

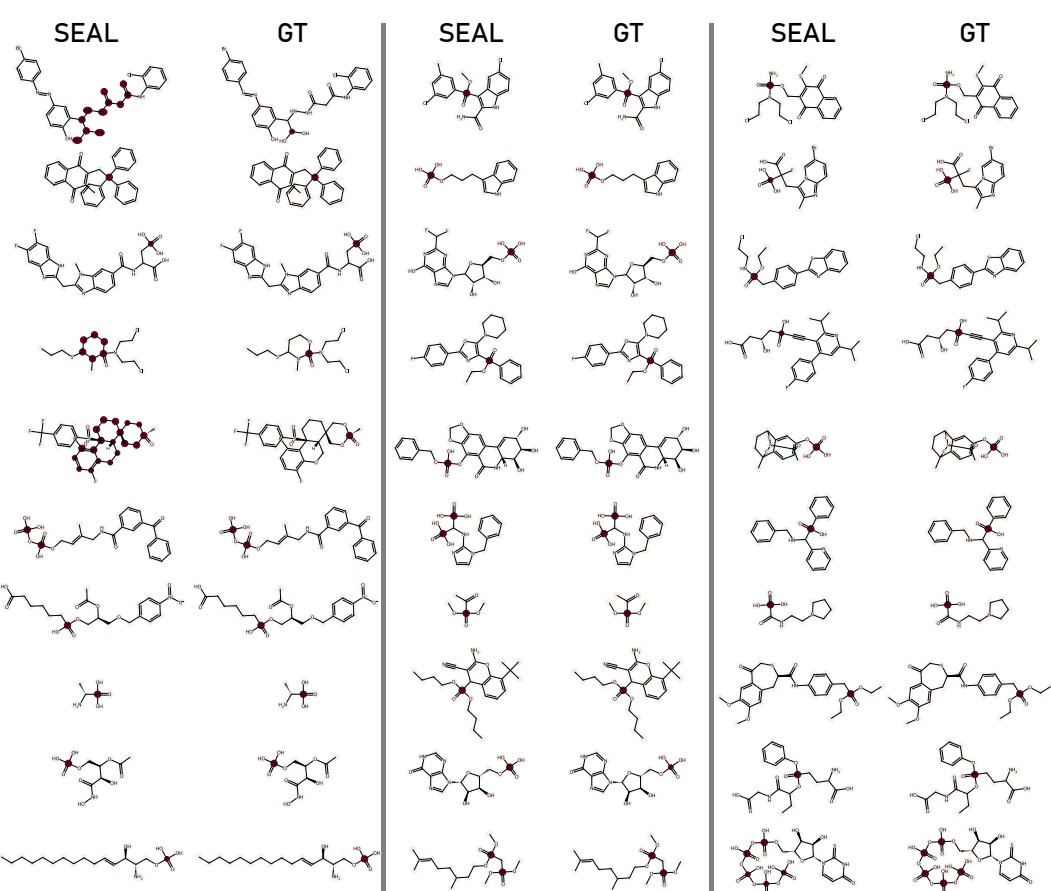

Figure 32: Node-level explanation examples of the SEAL method and Ground-Truth evaluated on the Phosphorus (P) task for the positive target class. The red color indicates that the highlighted atoms had a positive contribution to the compound's positive prediction. Blue as a negative contribution.

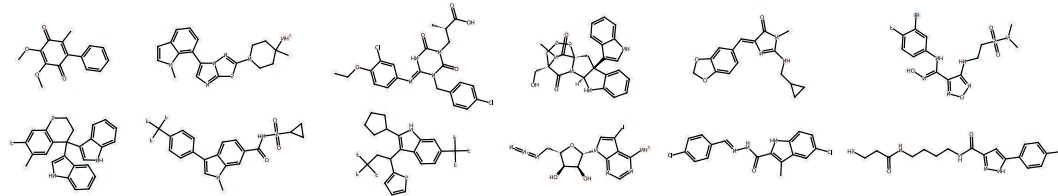

Figure 33: Node-level explanation examples of the SEAL method evaluated on the Phosphorus (P) task for the negative target class. The red color indicates that the highlighted atoms had a positive contribution to the compound's positive prediction. Blue as a negative contribution.

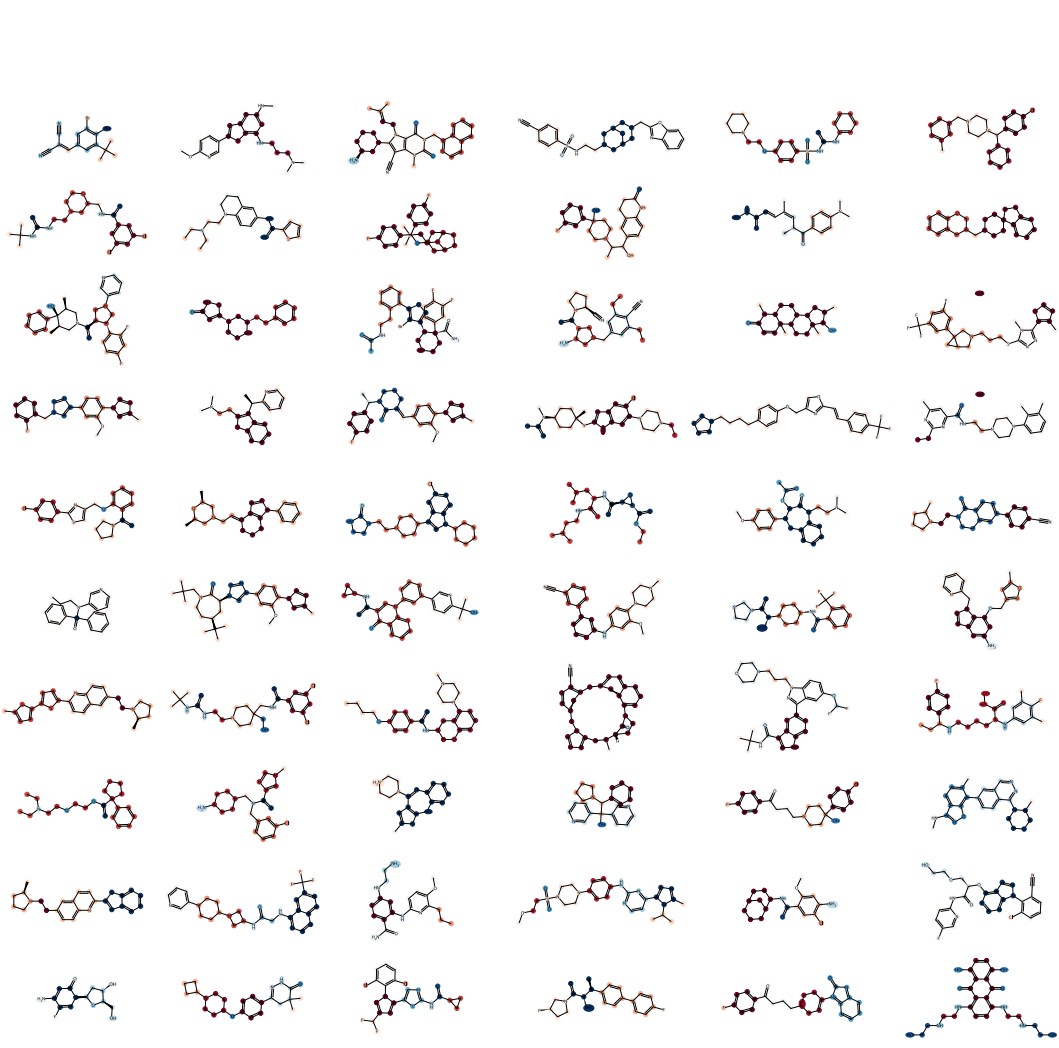

Figure 34: Node-level explanation examples of the SEAL method evaluated on the hERG dataset. The red color indicates that the highlighted atoms had a positive contribution to the positive prediction. Blue as a negative contribution.

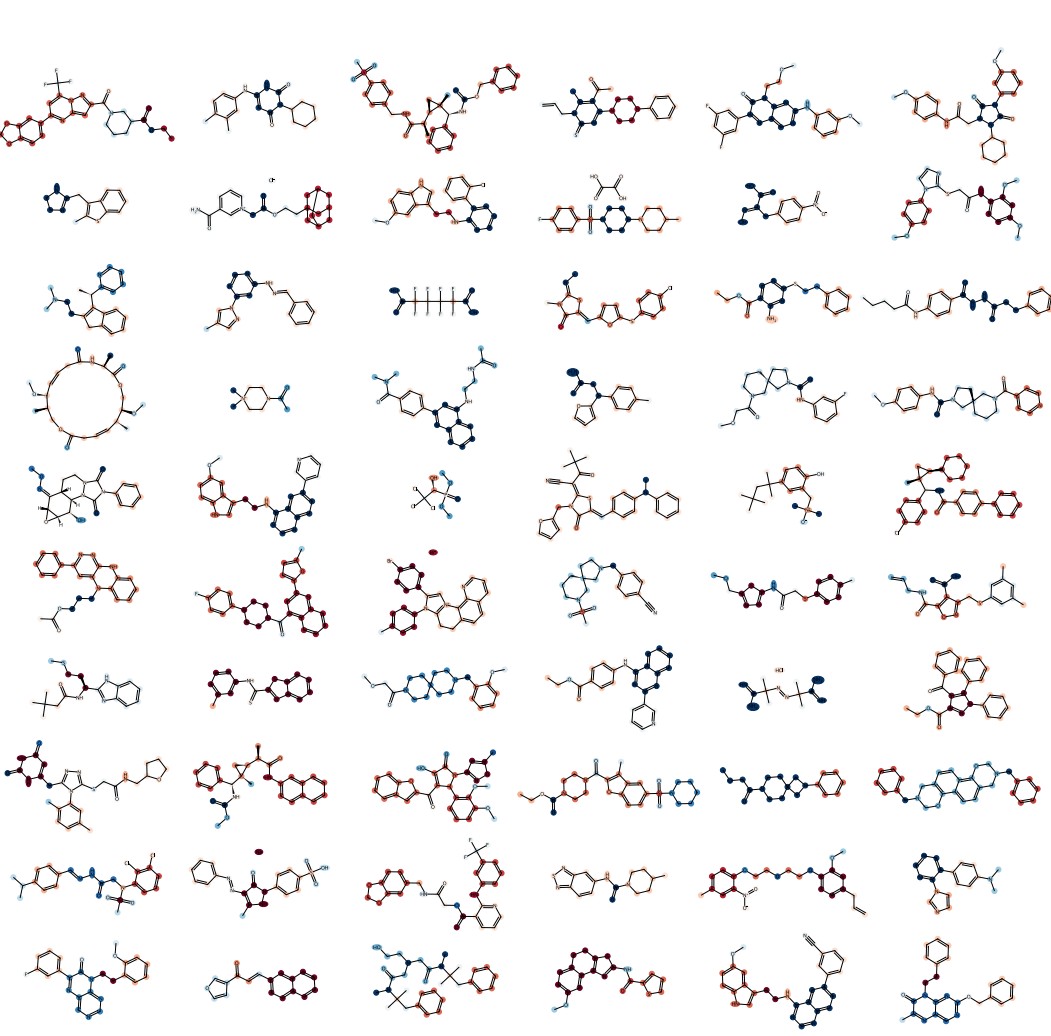

Figure 35: Node-level explanation examples of the SEAL method evaluated on the CYP2C9 dataset. The red color indicates that the highlighted atoms had a positive contribution to the positive prediction. Blue as a negative contribution.

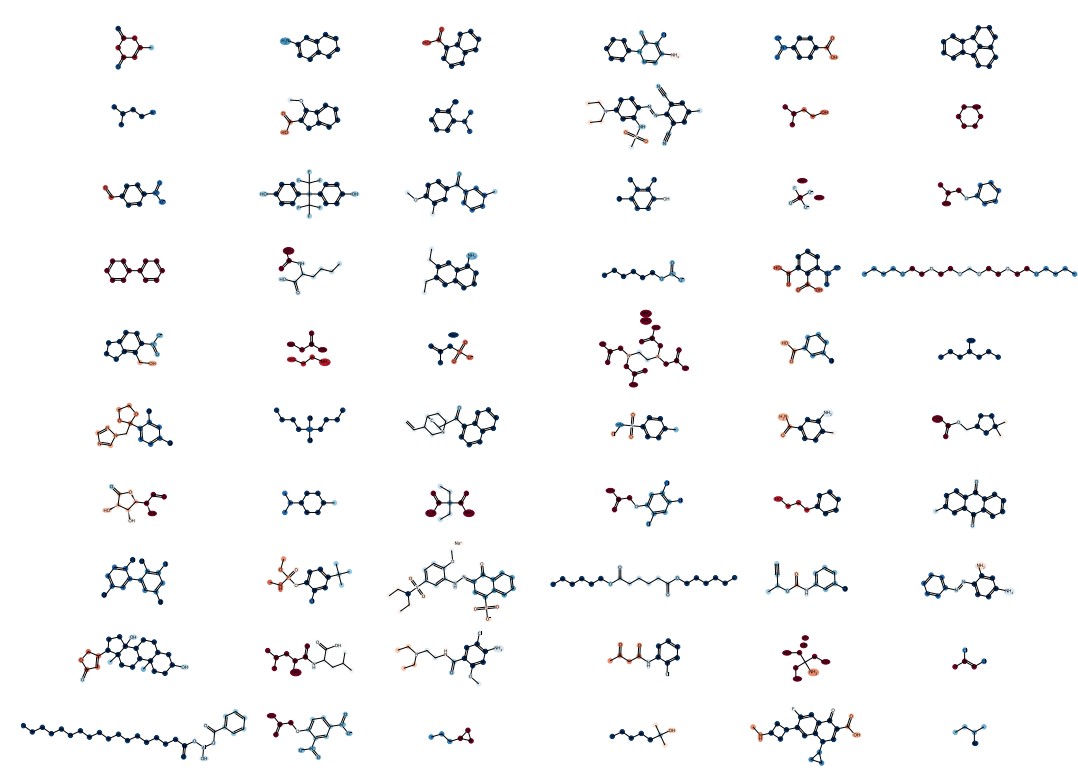

Figure 36: Node-level explanation examples of the SEAL method evaluated on the aqueous solubility dataset. The red color indicates that the highlighted atoms had a positive contribution to the positive prediction. Blue as a negative contribution.

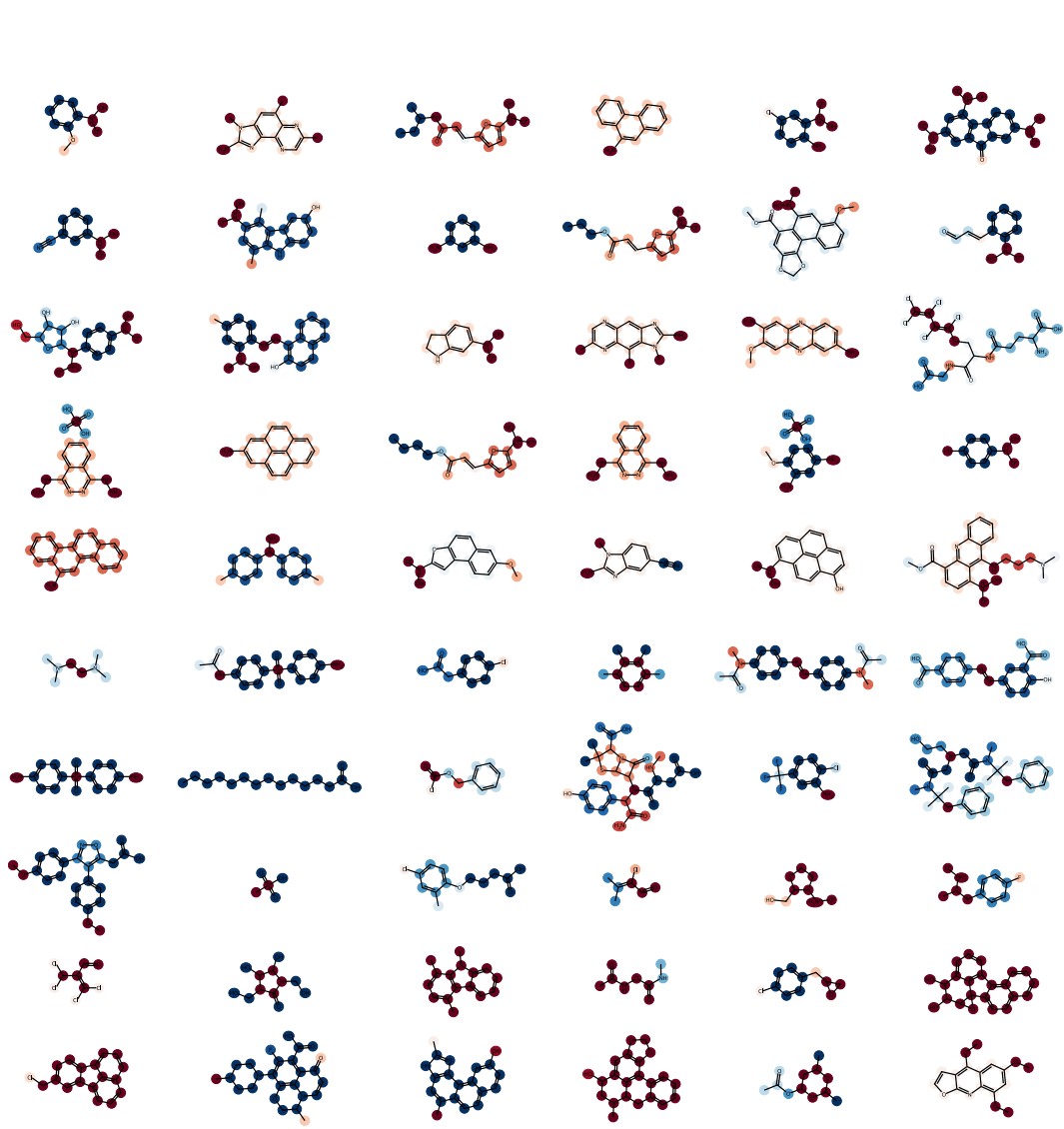

Figure 37: Node-level explanation examples of the SEAL method evaluated on the MUTAG dataset. The red color indicates that the highlighted atoms had a positive contribution to the positive prediction. Blue as a negative contribution.

