# OpenReview forum: "Fragment-Wise Interpretability in Graph Neural Networks via Molecule Decomposition and Contribution Analysis"
_ICLR.cc/2026/Conference — Submitted to ICLR 2026_

### Official Review · Reviewer_ZpcN · 2025-10-26

**Soundness:** 2
**Presentation:** 2
**Contribution:** 2
**Rating:** 2
**Confidence:** 4

**Summary:**

This paper aims to build an interpretable GNN model that quantifies the contribution of chemically meaningful fragments, ensuring that the explanations better align with domain knowledge. The proposed method decomposes molecular graphs into sets of fragments using a decomposition procedure similar to the BRICS algorithm and evaluates each fragment’s contribution. To isolate the effects of different fragments, the authors introduce a message-passing layer with separate parameters to control information flow within and between fragments. Experiments on synthetic datasets, real-world datasets, and a user study demonstrate the effectiveness of the proposed approach.

**Strengths:**

1.	The paper addresses an important problem — ensuring that graph explanations are consistent with established scientific understanding. This is a valuable and meaningful research direction, and the paper makes a concrete step toward it.

2.	The paper conducts experiments on both synthetic and real-world datasets to demonstrate effectiveness. The results are clearly presented through figures and tables.

3.	A user study is included to examine the differences between human understanding and graph explanation techniques, providing an interesting perspective and direction for future research.

**Weaknesses:**

1. Limited novelty. The paper lacks significant novelty. Explaining graphs at the fragment level and using message-passing schemes have been studied in several prior works [1–4]. The proposed method does not appear to introduce a new concept or contribution beyond these approaches.

2. Questionable methodological claim. The authors claim that “a final embedding of each node reflects not only its own properties but also the cumulative properties of distant atoms.” However, Section 3.2 adopts a similar approach to existing methods, making this claim of methodological uniqueness questionable.

3. Lack of detailed explanations. Some statements lack sufficient detail, making the paper difficult to follow. For example, in Line 95, the authors state that in both Cao et al. (2024) and Wang et al. (2025), fragments may contain significant signals from other parts of the molecule, thereby reducing local interpretability. However, the proposed method still allows information flow between fragments, so the distinction between approaches is not clearly explained. Similarly, in Line 105, the authors claim that Wu et al. (2023) use the BRICS method for fragment-level explanations; however, that method is not included in the comparisons.

4. Unclear selection of baselines. The choice of baseline models is questionable. HiGNN is primarily designed for property prediction, yet it is used here for an explanation task. The authors do not compare their method with other interpretable GNN models but instead focus on post-hoc methods and a property prediction model, limiting the fairness of the evaluation.

5. Simplistic evaluation and overly strong results. The model assumes linear independence in the final step (Section 3.1) and aggregates via summation. The evaluation tasks appear overly simple: in Table 4 (AUROC) and Table 5 (F1), the classification models achieve scores of 1.0 or very close to it. This raises concerns about potential data leakage or task simplicity and whether the proposed method can generalize to more complex benchmarks [5].

6. Unsupported statements. Some claims lack experimental or theoretical justification. For instance, in Line 155, the authors state that “the lack of bias would cause an equal distribution of contributions across all fragments, diminishing interpretability.” It is unclear how the label distribution affects this behavior or why the absence of $b$ would reduce interpretability.

7. Inconsistent or unclear visualizations. Several visualizations conflict with or are difficult to reconcile with the fragment decomposition method. For example, in the right panel of Figure 1, it is unclear how the shown functional groups are derived from the decomposition process. Similar issues appear in other figures as well.

8. Questionable benchmark selection. The chosen benchmarks are not well justified. The synthetic dataset, B-XAIC, is not a widely recognized benchmark. For real-world datasets, explanation datasets such as Mutagenicity are available but not used. The rationale for benchmark selection and fairness of comparison should be clarified.

9. Minor errors and missing details.There are some minor issues in the manuscript. For instance, in Line 256, “Hierarchical GNN” should be “HiGNN.” Some figures lack column labels or sufficient annotation of important nodes or subgraphs in the appendix.

10. Inconsistent interpretation granularity. While the paper claims to provide fragment-level explanations, it also proposes atom-level explanations. In some cases, the atom-level model (SEALAtom) performs better than the fragment-level model (SEAL), which raises concerns about the claimed benefits of fragment-level reasoning.


1, Yu, Z. and Gao, H., 2024. MAGE: Model-level graph neural networks explanations via motif-based graph generation. arXiv preprint arXiv:2405.12519.

2, Yu, Z. and Gao, H., 2022. Motifexplainer: a motif-based graph neural network explainer. arXiv preprint arXiv:2202.00519.

3, Liu, X., Luo, D., Gao, W. and Liu, Y., 2025, July. 3dgraphx: Explaining 3d molecular graph models via incorporating chemical priors. In Proceedings of the 31st ACM SIGKDD Conference on Knowledge Discovery and Data Mining V. 1 (pp. 859-870).

4, Gui, S., Yuan, H., Wang, J., Lao, Q., Li, K. and Ji, S., 2023. Flowx: Towards explainable graph neural networks via message flows. IEEE Transactions on Pattern Analysis and Machine Intelligence, 46(7), pp.4567-4578.

5, Bui, N., Nguyen, H.T., Nguyen, V.A. and Ying, R., 2024. Explaining graph neural networks via structure-aware interaction index. arXiv preprint arXiv:2405.14352.

**Questions:**

1.	In Figure 1, is the claim that “the most polar groups contribute positively and the hydrophobic groups contribute negatively” supported by existing scientific evidence? Does the contribution of each fragment align with known chemical findings?

2.	Could the authors provide a comparison of computational efficiency or runtime between the proposed method and the baselines?

3.	In Line 133, can the embeddings before the graph readout stage be analyzed to evaluate atom-level contributions?

4.	For the synthetic dataset, how are the ground-truth explanations obtained? In the user study, did participants provide what they believe the true explanations should be based on their expertise? Why are the ground-truth explanations not presented in the user study?

5.	The paper states that a high $\lambda$ leads to more interpretable results, but in some experiments, the chosen values are very small. Could the authors explain this inconsistency?

6.	Why is HiGNN used both as the explanation model and as a baseline model?

7.	How is the budget for the final explanation results determined? How do the authors decide how many fragments are selected for the final explanation? Additionally, could the authors explain the color-coding scheme used in Figure 4?

8.	SEALAtom treats a single atom as a functional group. Why does it outperform baselines such as GNNExplainer and PGExplainer?

9.	In Line 676, the paper states that “we decide to mask the same amount of atoms for each molecule among all methods.” Is this assumption reasonable for the budget settings?

---

> ### Author Response · Authors · 2025-12-04
>
> > **W1.** Limited novelty.
>
> The common theme across MotifExplainer, MAGE, FlowX, and 3DGraphX is that they either (i) extract or score motifs post-hoc, (ii) generate model-level explanations, or (iii) analyze message flows after training. SEAL’s key contribution is architectural: it embeds fragment decomposition into the model computation, provides additive fragment contributions as the prediction, and uses a learnable, regularized inter-fragment weights so the model can learn when long-range interactions are necessary vs. when local attribution suffices. This difference, making fragment attribution intrinsic and controllable during training, is not realized by the cited works. We have included these works in the extended related work in Appendix F of the revised paper.
>
> > **W2.** Questionable methodological claim.
>
> The Reviewer is correct that standard message-passing GNNs propagate information from increasingly distant atoms, causing node embeddings to converge. Our statement highlights this well-known oversmoothing behavior, not a claim of uniqueness. SEAL addresses this issue specifically through the BRICS-based decomposition and the regularization of inter-fragment weights, which constrain cross-fragment mixing and reduce the homogenization of node representations. This design ensures that fragment embeddings retain more localized chemical meaning than in standard GNN layers. We clarify this rationale in the revision (see contributions at the end of the Introduction).
>
> > **W3.** Lack of detailed explanations.
>
> We agree that more explanation was needed. Although SEAL permits information flow between fragments, the L1 regularization on inter-fragment weights explicitly controls the extent of this flow, preventing the fragment representations from absorbing unrelated signals, addressing the concern raised for Cao et al. (2024) and Wang et al. (2025). We have now clarified this distinction in the text. Regarding Wu et al. (2023), the revised version includes SME, which corresponds to that method and allows a direct comparison under our evaluation protocol.
>
> > **W4.** Unclear selection of baselines.
>
> HiGNN is included not merely as a property-prediction baseline but because the authors explicitly present it as a hierarchical model whose fragment-level representations can be interpreted for explanation. Since SEAL also uses fragment-level reasoning, HiGNN is among the closest architectural comparators. In addition, we now include PGIB and SME in the revised manuscript, which expands the coverage of interpretable GNN baselines. Together, these provide a fair and representative evaluation across both inherently interpretable and post-hoc methods.
>
> > **W5.** Simplistic evaluation and overly strong results.
>
> The synthetic tasks in Tables 11, 12, and 13 (in the revised paper) come from the B-XAIC benchmark, which is intentionally designed to test whether models can detect simple, ground-truth substructures, hence the high predictive performance across all methods. This does not indicate data leakage; rather, the benchmark’s purpose is to isolate explanation quality in settings where the predictive task is not the bottleneck. Our model’s final additive structure is intentionally simple, but fragment representations still incorporate contextual information when $\lambda$ is small. This setup parallels approaches like SHAP, which also decompose predictions into additive contributions. SEAL’s strong performance on the synthetic tasks and its real-world fidelity and Mutagenicity results demonstrate that it generalizes beyond these simplified benchmarks.
>
> > **W6.** Unsupported statements.
>
> We agree that this statement required clarification, and we have revised it accordingly. When the model has no bias term, it implies that an “empty molecule” would be mapped to a prediction of zero. This is rarely chemically meaningful, as the baseline value of a property depends on physical units, measurement conventions, and dataset normalization. Without a bias, the model is forced to reconstruct this baseline value entirely through fragment contributions, even when fragments differ across molecules. This often leads to contributions being redistributed uniformly or arbitrarily across fragments, which diminishes interpretability. In the revision, we provide more clarification and avoid making the claim overly general.

---

> > ### Author Response · Authors · 2025-12-04
> >
> > > **W7.** Inconsistent or unclear visualizations.
> >
> > We appreciate the request for clearer visualizations. The functional groups shown in Figure 1 (right panel) arise directly from the BRICS fragmentation rules, which cut molecules at chemically meaningful bonds commonly used in medicinal chemistry. We have now explicitly described how BRICS decomposes molecules, explaining why certain fragments appear as they do, and improved the figure captions to make the decomposition principles clear.
> >
> > > **W8.** Questionable benchmark selection.
> >
> > B-XAIC is a recently introduced benchmark specifically designed for evaluating substructure-level explanations in molecular graphs. To our knowledge, it is the only benchmark that provides clean, unambiguous ground-truth motifs at scale, which is essential for quantitatively assessing explanation quality. While datasets such as Mutagenicity contain functional-group annotations (e.g., nitro groups), these are chemistry-derived heuristics, and not all molecules containing such groups exhibit the labeled property, making them less reliable as ground-truth rationales. Nonetheless, in the revised manuscript, we added Mutagenicity to broaden our real-world evaluation and show that SEAL extends well beyond synthetic settings.
> >
> > > **W9.** Minor errors and missing details.
> >
> > We thank the Reviewer for pointing out these issues. In the revision, we corrected the typographical error (“HiGNN”) and improved figure readability by adding clear column labels and highlighting the relevant atoms and subgraphs more explicitly in the appendix.
> >
> > > **W10.** Inconsistent interpretation granularity.
> >
> > SEAL is designed to be flexible with respect to granularity of explanation: fragment-level reasoning is appropriate when chemical substructures align with the task, whereas atom-level reasoning may perform better when the signal is localized or when the chosen fragmentation (e.g., BRICS) is too coarse for a specific case. SEAL’s architecture supports both by simply changing the fragmentation procedure, which is why SEALAtom can outperform SEAL on tasks dominated by single-atom contributions. This does not contradict the benefits of fragment-level reasoning. It demonstrates that interpretation granularity should match the task, and that SEAL provides a unified framework capable of adapting to different levels. We also note this explicitly as future work, where we plan to investigate more flexible and adaptive fragmentation strategies.

---

> > > ### Author Response · Authors · 2025-12-04
> > >
> > > > **Q1.** In Figure 1, is the claim that “the most polar groups contribute positively and the hydrophobic groups contribute negatively” supported by existing scientific evidence? Does the contribution of each fragment align with known chemical findings?
> > >
> > > Yes, the claim in Figure 1 is supported by well-established principles from organic and medicinal chemistry. Polar functional groups (e.g., hydroxyl, amine, carboxyl) increase aqueous solubility because they form favorable hydrogen-bonding and electrostatic interactions with water, whereas hydrophobic fragments (e.g., alkyl chains, aromatic rings) decrease solubility by increasing lipophilicity and reducing solvent–solute interactions. These relationships are standard material in organic chemistry and medicinal chemistry textbooks and are consistent with widely used descriptors such as logP and hydrogen-bond donor/acceptor counts.
> > >
> > > That said, we agree that this chemical intuition may not be obvious to a machine learning audience. In the revised version, we have added a brief clarification and supporting references to help readers without a chemistry background understand why the polarity and hydrophobicity of fragments influence their contribution to solubility.
> > >
> > > > **Q2.** Could the authors provide a comparison of computational efficiency or runtime between the proposed method and the baselines?
> > >
> > > SEAL’s forward pass is architecturally almost identical to a standard message-passing GNN: it uses the same neighborhood aggregations plus three linear maps per layer. As a result, inference time is comparable to ordinary GNNs. The practical difference is that SEAL produces fragment-level attributions natively (one forward pass) while many post-hoc explainers require many additional passes or optimization loops to produce explanations. To make this explicit, we added a complexity/runtime analysis to the appendix comparing parameter counts and wall-clock explanation time (inference + explainer) for SEAL and the baselines.
> > >
> > > > **Q3.** In Line 133, can the embeddings before the graph readout stage be analyzed to evaluate atom-level contributions?
> > >
> > > Yes, one can apply post-hoc techniques (e.g., gradient-based or Grad-CAM–style methods) to pre-aggregation node embeddings to obtain atom-level signals. In practice, however, oversmoothing and the subsequent global pooling make those embeddings less reliable for faithful attribution: deep message passing tends to homogenize node vectors and the global readout entangles signals. This is precisely why SEAL produces intrinsic additive atom/fragment contributions via local pooling and controlled inter-fragment propagation rather than relying on post-hoc analysis of pre-readout embeddings.
> > >
> > > > **Q4.** For the synthetic dataset, how are the ground-truth explanations obtained? In the user study, did participants provide what they believe the true explanations should be based on their expertise? Why are the ground-truth explanations not presented in the user study?
> > >
> > > The B-XAIC synthetic benchmark provides explicit ground-truth substructure labels (e.g., presence/counting of rings, single-atom motifs like halogens) by construction; these pattern labels are the basis for the SE/NE metrics. For real molecules (e.g., solubility), there is no single objective ground truth, so we complemented fidelity evaluations with a blinded user study: domain chemists judged which explanations matched chemical intuition. We did not present “ground truth” in the user study because no definitive ground-truth annotations exist for those real tasks; instead, the study assesses human interpretability and practical utility as a complement to fidelity and synthetic-task evaluation.
> > >
> > > > **Q5.** The paper states that a high lambda leads to more interpretable results, but in some experiments, the chosen values are very small. Could the authors explain this inconsistency?
> > >
> > > We select $\lambda$ with a principled protocol: choose the largest $\lambda$ that does not significantly degrade predictive performance (Wilcoxon signed-rank test against the best model, see Section 3.2). Small $\lambda$ values therefore arise when the task genuinely requires broader inter-fragment context (long-range interactions): raising $\lambda$ would hurt accuracy. Conversely, tasks dominated by local substructures favor larger $\lambda$.

---

> > > > ### Author Response · Authors · 2025-12-04
> > > >
> > > > > **Q6.** Why is HiGNN used both as the explanation model and as a baseline model?
> > > >
> > > > HiGNN is fundamentally a hierarchical GNN whose design produces fragment-level representations that can be inspected for explanations; thus, it legitimately serves as an inherently interpretable model. At the same time, because it is also a competitive property-prediction architecture, we include it as a predictive baseline to compare both accuracy and interpretability. In short, HiGNN is both an architecture for prediction and one of the closest existing models that produces fragment-level signals, so treating it in both roles is appropriate and informative.
> > > >
> > > > > **Q7.** How is the budget for the final explanation results determined? How do the authors decide how many fragments are selected for the final explanation? Additionally, could the authors explain the color-coding scheme used in Figure 4?
> > > >
> > > > Our explanation “budget” follows standard practice in molecular XAI, where sparse explanations (those highlighting only a small fraction of atoms) are the most informative for downstream decision-making. For fairness across methods, we selected masking budgets of 10%, 20%, and 30% of the atoms in each molecule. For SEAL, which operates at the fragment level, we ranked fragments by contribution and included them until the atom count slightly exceeded the target threshold; we then used this exact atom count as the masking budget for all other explainers. This ensures that each method masks the same number of atoms per molecule, allowing a controlled comparison of fidelity. In this revision, we also included results for higher masking budgets up to 70%.
> > > >
> > > > Regarding Figure 4, the color-coding corresponds to contribution magnitude: “The more intense the red color, the greater the contribution of a substructure or atom. For clarity, the gray regions indicating specific fragments were omitted.” The appendix now contains a detailed description and visualization examples including the fragment outlines.
> > > >
> > > > > **Q8.** SEALAtom treats a single atom as a functional group. Why does it outperform baselines such as GNNExplainer and PGExplainer?
> > > >
> > > > SEALAtom treats each atom as a “fragment,” which is appropriate for tasks where the predictive signal is localized at single atoms or very small substructures (e.g., halogens, boron, and phosphorus). Post-hoc methods like GNNExplainer or PGExplainer rely on masking or gradient signals and may diffuse importance across neighboring atoms, especially when oversmoothing is present. In contrast, SEALAtom produces intrinsic, additive atom-level contributions that prevent such diffusion. This explains why SEALAtom outperforms post-hoc methods in scenarios where the ground-truth rationale lies at the atomic level, as illustrated in Figure 4.
> > > >
> > > > > **Q9.** In Line 676, the paper states that “we decide to mask the same amount of atoms for each molecule among all methods.” Is this assumption reasonable for the budget settings?
> > > >
> > > > Yes, the assumption is reasonable and is standard in explanation benchmarking: to compare fidelity across methods, the masking budget must be identical for all explainers on each molecule. Otherwise, differences in fidelity may reflect differences in the number of atoms removed rather than the quality of the explanation. Using a fixed atom count ensures a fair, controlled, one-variable comparison: all methods operate under the same sparsity constraint. We clarify this procedure in the revised manuscript: “Therefore, we select the percentage of atoms in the most relevant fragments that is closest to 10% (e.g. 13%) and mask the same amount of most relevant atoms generated by the baseline methods to keep the sparsity budget fixed for fair comparison.”

---

### Official Review · Reviewer_onw9 · 2025-10-27

**Soundness:** 3
**Presentation:** 2
**Contribution:** 3
**Rating:** 2
**Confidence:** 5

**Summary:**

The paper proposes a novel fragment-wise approach to molecular property prediction, utilizing BRICS fragmentation and aggregating fragment-level contributions to enhance interpretability of Graph Neural Network models. The rationale is grounded in pharmaceutical chemistry, where molecular properties often relate to functional groups, and the model aims to mirror this intuition in predicting outcomes.

**Strengths:**

The work addresses an important gap in interpretable molecular property prediction by focusing on fragment-level explanations, which have strong chemical relevance.

The adoption of BRICS and fragment aggregation is conceptually appealing and could aid chemists in understanding model decisions in familiar terms.

The paper is well-motivated and offers thoughtful discussion about interpretability, with the potential to advance explainable AI in the molecular domain.

**Weaknesses:**

The main evaluation relies on synthetic datasets that are often trivial and do not sufficiently challenge the model’s capacity for real-world tasks. Only three real-world datasets are used for more robust assessment, limiting generalizability. There exist many other datasets which are not used in this work. For instance, in the paper of HiGNN (Zhu et al., 2022) there are 11 real world datasets and none of them is investigated in this paper.

A big part of the evaluation checks alignment with ground-truth substructures but does not verify faithfulness with respect to the model’s decision process, which is instead done only with 2 datasets. An explanation should first be evaluated for the faithfulness using standard metrics (Fidelity, Inv Fidelity): if a model learns a spurious bias, the correct explanation should reflect that bias. Only after confirming model alignment, comparisons to chemical ground truth are meaningful. Let me cite Agarwal et al, 2023: "trained GNN model may only capture one or an entirely different rationale. In such cases, evaluating the explanation output by a state-of-the-art method using the ground-truth explanation is incorrect because the underlying GNN model does not rely on that ground-truth explanation. In addition, even if a unique ground-truth explanation generates the correct class label, the GNN model trained on the data could be a weak predictor using an entirely different rationale for prediction. Therefore, the ground-truth explanation cannot be used to assess post hoc explanations of such models."

Fragment extraction is presented as a generalizable mechanism, but the experiments rely exclusively on pre-defined methods (BRICS), which fail to pinpoint atomic-level contributions (e.g., Boron atom in Figure 4), and do not capture long-range interactions crucial for complex properties.

Benchmarking against relevant self-explainable GNNs (GNAN, GIB, VIB, PiGNN, KerGNN) is scarce as only one method is used as comparison, weakening the claims of superiority or novelty in interpretability.

Comparison to leading molecular property predictors (e.g., Grover, MultiChem, HyperMol) is absent.

The user study is incorrectly posed and totally useless, as the question asks which part of the molecule is thought to better impact the solubility. However, the quality of explanations should be primarily checked against the model. The user cannot know whether the highlighted part is truly the one impacting the model’s prediction. Additionally, no information is provided on the number and specialty of the study participants. The reviewer would assume that they are chemists. However, it is obvious that a chemist would be biased toward using functional groups, as that is exactly how they reason, but there are no other methods among the techniques that actually use fragments.

**Questions:**

Can the authors address why evaluation against other self-explaining GNN methods was omitted?

Can the authors address why other real world datasets were omitted?

Why wasn't fidelity calculated on B-XAIC datasets?

How do the authors ensure that user studies accurately evaluate faithfulness of explanation with respect to the model (rather than human chemistry bias), and provide more details about study participants?

Why do the authors only do Figure 6 on Fidelity with 30% of masking?

---

> ### Author Response · Authors · 2025-12-04
>
> > **W1.** Limited benchmarks.
>
> We agree that broader evaluation is valuable, and in the revision, we added the Mutagenicity dataset, which includes chemically meaningful structural labels and is well-suited for assessing fragment-level explanations. Our overall evaluation strategy follows standard practice in explainable molecular ML: synthetic datasets provide the only setting with ground-truth substructure annotations for quantitative explanation assessment, while real-world datasets test robustness across different biological tasks. Many of the datasets used in HiGNN do not provide explanation ground truth, making them unsuitable for evaluating our interpretability objective. Given the paper’s focus on explanation quality and space constraints, we believe the chosen set of datasets offers a balanced and meaningful assessment of both performance and interpretability.
>
> > **W2.** Faithfulness metric.
>
> We agree that faithfulness is essential. Our evaluation already includes a fidelity metric on two real-world datasets where such perturbation-based evaluations are meaningful. For the synthetic B-XAIC benchmark, we follow its established evaluation protocol, which is designed precisely for settings where the model is able to learn the ground-truth substructure reliably. In these tasks, the tasks are intentionally simple to avoid the failure mode described by Agarwal et al. (2023), where the predictor learns a spurious shortcut and explanation-ground truth comparisons become invalid. Furthermore, perturbation-based fidelity metrics on molecular graphs require masking atoms or bonds, which typically produces out-of-distribution structures, making fidelity scores unreliable when used in isolation. For this reason, we combine (i) faithfulness evaluations for real-world datasets, (ii) controlled synthetic setups with ground-truth rationales, and (iii) real-world qualitative assessment through a user study.
>
> > **W3.** Experiments rely on pre-defined fragmentation.
>
> We do not claim that fragmentation is universally optimal or automatically generalizable. We rely on BRICS fragmentation because it is chemically interpretable, widely used in medicinal chemistry, and provides an intuitive granularity for most small molecules. Our results already show that in special cases, such as the Boron example, the user can resort to SEALAtom, which treats atoms as fragments and resolves such fine-grained contributions. Importantly, SEAL does not depend on BRICS specifically: fragmentation is a preprocessing step and can be replaced by any domain-specific or learned pooling strategy. We clarified this flexibility in the revision.
>
> > **W4.** Benchmarking against self-explainable GNNs.
>
> We have expanded our set of self-explainable GNN baselines in the revised manuscript by including PGIB, a recent prototype-based information-bottleneck model. This addition complements ProtGNN and HiGNN and covers the major families of inherently interpretable graph models. We have also included SME, which is a recent fragment-based explainer. With the expanded baselines, we believe our evaluation fairly represents the current landscape of self-explainable GNNs for molecular property prediction.
>
> > **W5.** Comparison to leading molecular property predictors.
>
> Our goal is to evaluate interpretability without conflating it with architectural advantages unrelated to SEAL. Models such as Grover or MultiChem are attention-based architectures with orders of magnitude more parameters and different training objectives (often self-supervision on millions of molecules). Comparing SEAL directly to these models would not be an architecture-controlled experiment and would obscure interpretability effects behind scale and pretraining. Instead, we follow standard XAI practice and compare SEAL to architecturally comparable GNN backbones (GIN, GAT, and HiGNN) to demonstrate that SEAL improves interpretability without sacrificing predictive performance under matched capacity and training conditions.
>
> > **W6.** The user study is incorrectly posed and totally useless.
>
> The user study is intended only as a complementary evaluation after establishing faithfulness (where appropriate) and alignment with ground-truth substructures. User studies are widely recognized as an important component of XAI, and some argue that progress in this field is impossible without involving end-users [3]. Information about the domain experts, including their number and expertise, is described in Section 4.3. Our study includes 14 domain experts (chemists), and their preference for SEAL reflects its alignment with chemically intuitive reasoning. We acknowledge that chemists naturally think in terms of functional groups; this is precisely why fragment-level interpretability is valuable in molecular ML.
>
> [3] Pičulin, Matej, et al. "Position: Explainable AI Cannot Advance Without Better User Studies." Forty-second International Conference on Machine Learning Position Paper Track.

---

> > ### Author Response · Authors · 2025-12-04
> >
> > > **Q1.** Can the authors address why evaluation against other self-explaining GNN methods was omitted?
> >
> > ProtGNN was already included in the original submission as a representative self-explaining GNN. In the revised manuscript, we have further expanded our comparisons by adding PGIB, a recent prototype-based, inherently interpretable model. Together, ProtGNN and PGIB cover two major families of self-explaining GNNs (prototype-based and IB-based). We believe this provides a meaningful and representative evaluation within the scope of architectures applicable to molecular graphs.
> >
> > > **Q2.** Can the authors address why other real world datasets were omitted?
> >
> > Our focus is on interpretability, not large-scale benchmarking across all molecular datasets. We selected datasets that allow explanation quality to be meaningfully assessed, either via ground-truth substructure labels or through validated fidelity metrics. In the revision, we added Mutagenicity, which contains chemical functional groups (e.g., nitro groups) that serve as ground-truth structural explanations. This strengthens our real-world evaluation while staying aligned with the interpretability focus of the paper.
> >
> > > **Q3.** Why wasn't fidelity calculated on B-XAIC datasets?
> >
> > The B-XAIC benchmark focuses on detecting known structural patterns and does not include fidelity-based evaluation in its protocol. Applying perturbation-based fidelity on these synthetic graphs would be misleading, as fidelity metrics for molecular graphs require masking atoms or bonds, which alters the chemical validity of the structures and introduces out-of-distribution artifacts. For these reasons, we follow the established benchmark protocol and evaluate explanation quality using the provided ground-truth motifs.
> >
> > > **Q4.** How do the authors ensure that user studies accurately evaluate faithfulness of explanation with respect to the model (rather than human chemistry bias), and provide more details about study participants?
> >
> > The purpose of the user study is not to evaluate faithfulness with respect to the model. This is already addressed through fidelity and structural ground-truth experiments. Instead, this study assesses human interpretability and clarity of the explanations. This follows common practice in XAI, where user studies complement quantitative metrics by evaluating how well explanations align with human intuition and decision-making. As noted in the original submission, all participants were trained chemists, and the methodological details (number and expertise) were described in the paper.
> >
> > > **Q5.** Why do the authors only do Figure 6 on Fidelity with 30% of masking?
> >
> > We used a 30% masking threshold in Figure 6 because, for small molecular graphs, this percentage reliably highlights several atoms without overwhelming the visualization. This threshold avoids cases where too few or too many atoms are masked. To address threshold sensitivity, the revised paper includes Figure 9, which reports fidelity results at 10%-70% masking, showing that our conclusions remain consistent across thresholds.

---

### Official Review · Reviewer_YTvH · 2025-11-01

**Soundness:** 2
**Presentation:** 3
**Contribution:** 2
**Rating:** 6
**Confidence:** 4

**Summary:**

The paper propose a new algorithm called SEAL that identifies fragments in molecule and constraints message passing architecture. It measures the influence of the fragments in the molecule to the prediction rather than focusing on node or edge level. This would increase the interpretability of GNN and generate more intuitive explanations.

**Strengths:**

- Good motivation and introduction to the problem.
- Code sharing for reproducibility.
- User study to show the effectiveness of the model compared to the baselines.

**Weaknesses:**

- It is not clear if fragmentation algorithm is unique in the model, or just borrowed from the available methods.
- It is not clear how SEALAtom differs than the original graph without any fragmentation.
- Equation 3 requires more explanation.
- No theory behind the method. One can claim that the model should be able to learn the influential edges without separating intra and inter edges. Besides, what is the guarantee that inter-edges are not influential?

**Questions:**

- What is the guarantee that the initial fragments are well-decided?
- Can you summarize what novelty this paper brings to the field?
- Can you explain what SEALAtom actually offers compared to the original graph? If every atom is a fragment, then that is already the original graph.
- Can you explain your GNN layer in more detailed with the size of each elements (i.e., trainable weight matrices, embedding size etc.)?

---

> ### Author Response · Authors · 2025-12-04
>
> > **W1.** The fragmentation algorithm description.
>
> We thank the Reviewer for raising this point. As clarified in the manuscript, our fragmentation procedure is not a new algorithm but builds on an established BRICS-style decomposition, following the variant proposed by Zhang et al. (2021). We intentionally adopt a standardized, chemically validated fragmentation method rather than introducing a new one, because our contribution lies not in the fragmentation algorithm itself, but in how SEAL integrates fragment structure directly into the model’s message-passing dynamics. The chosen fragmentation ensures consistency, chemical plausibility, and reproducibility across datasets, while allowing the architectural innovations of SEAL (regularized inter-fragment communication, localized fragment pooling, and interpretable additive contribution) to operate on chemically meaningful units. We clarify this in the revised version and emphasize that fragmentation is a design choice external to the architecture, similar to any preprocessing step.
>
> > **W2.** The difference between SEALAtom and the original graph.
>
> SEALAtom differs from a standard GNN in a fundamental way: although each atom forms a “fragment,” the architecture still uses local pooling followed by a fragment-level contribution network, resulting in scalar per-atom contributions that sum to the final prediction. This is not equivalent to a classical GNN, where node embeddings are globally pooled, and nonlinear transformations entangle all nodes before the final prediction. SEALAtom produces an additive, interpretable decomposition of the output into atom-level components, whereas a standard GNN does not. In addition, the architecture preserves the explicit separation of intra- and inter-fragment message channels in Equation (3), so even at atomic granularity, the model controls information flow, unlike conventional GNN layers.
>
> > **W3.** Equation 3 *(note: 2 in the revised version)* requires more explanation.
>
> We appreciate the request for further clarification of Equation 3. Although the equation is already described in detail, in the revised manuscript, we expand the accompanying text to make explicit what each term represents: the learnable intra-fragment weights act on edges inside a fragment, the inter-fragment weights act on edges crossing fragment boundaries, and the linear transformation ensures each node can update independently of either neighborhood if needed. The resulting formulation maintains full expressive capacity while enabling the model to separate local and global signals for interpretability purposes.
>
> > **W4.** No theory behind the method. One can claim that the model should be able to learn the influential edges without separating intra and inter edges. Besides, what is the guarantee that inter-edges are not influential?
>
> We agree that a standard GNN could, in principle, learn to down-weight uninformative edges, but years of work on graph oversmoothing show that message passing inherently blends node information across neighborhoods, making it difficult to reliably isolate which atoms or substructures truly drive a prediction [1, 2]. In such models, influential edges and non-influential edges are processed through the same learned transformation, which causes attribution signals to be spread across the graph. This is precisely why post-hoc explainers often produce diffuse or noisy subgraph masks.
>
> SEAL’s separation of intra- and inter-fragment channels does not presume that inter-fragment edges are unimportant. Instead, it modulates their influence through learnable weights and an interpretable regularization term. If a task genuinely requires long-range interactions or cross-fragment communication, the model learns this and retains strong predictive performance. The synthetic benchmark results demonstrate exactly this behavior: for tasks requiring global context (e.g., PAINS), SEAL automatically selects $\lambda$=0, preserving full message passing. Conversely, when oversmoothing obscures local explanations, the model favors higher $\lambda$, yielding more faithful fragment-level attributions without harming accuracy.
>
> Thus, SEAL does not impose a hard structural bias but provides a mechanism for the model to learn when cross-fragment communication is essential and when it degrades interpretability. We also added a new ablation removing the regularization and only initializing inter-fragment weights to small values; the model still diffuses information across fragments, and its interpretability metrics deteriorate, confirming the need for explicit control.
>
> [1] Wu, Xinyi, et al. "A Non-Asymptotic Analysis of Oversmoothing in Graph Neural Networks." The Eleventh International Conference on Learning Representations.
>
> [2] Rusch, T. Konstantin, Michael M. Bronstein, and Siddhartha Mishra. "A survey on oversmoothing in graph neural networks." arXiv preprint arXiv:2303.10993 (2023).

---

> > ### Author Response · Authors · 2025-12-04
> >
> > > **Q1.** What is the guarantee that the initial fragments are well-decided?
> >
> > We do not claim a formal, provable guarantee that any fragmentation procedure is optimal for every task; instead, we use a chemically grounded, reproducible BRICS-style decomposition to produce fragments that are aligned with how chemists reason about molecules. The fragmentation is described in Section 3.1; using this standardized scheme ensures chemical plausibility and reproducibility across datasets. Empirically, the chosen decomposition yields reliable, human-preferred explanations on real tasks (user study results) and strong synthetic-benchmark performance; when the BRICS split is inappropriate (e.g., very complex boron substructures) we report the limitation and show that SEALAtom (atomic fragments) remedies that case (see Figure 3, Table 14).
> >
> > > **Q2.** Can you summarize what novelty this paper brings to the field?
> >
> > The paper’s core novelty is architectural: SEAL integrates fragment decomposition into the model by (i) performing local, fragment-level pooling so the prediction is an explicit additive sum of fragment contributions, and (ii) introducing separate, learnable intra-fragment and inter-fragment message-passing weights with an L1 regularizer on inter-fragment weights to control cross-fragment information flow. This combination yields an inherently interpretable output (scalar fragment contributions that sum to the prediction) while preserving expressiveness. We therefore contribute a method that moves beyond post-hoc fragment masking and beyond opaque hierarchical/attention schemes by offering a simple, tunable mechanism that trades off locality and globality in a principled way.
> >
> > > **Q3.** Can you explain what SEALAtom actually offers compared to the original graph? If every atom is a fragment, then that is already the original graph.
> >
> > When every atom is treated as a fragment (SEALAtom), the input graph nodes are the same as in a standard atomic GNN, but the prediction head and layer structure still differ. SEALAtom (a) computes per-atom contributions so the final output is an explicit additive decomposition into atom-level scalars, and (b) retains the cross-atom (inter-fragment) message regularization. In contrast, a conventional GNN uses a global readout that entangles node embeddings before the prediction, so it does not deliver an additive, per-atom contribution interpretation. Empirically, SEALAtom performs especially well on single-atom detection tasks but can be worse on larger substructure tasks.
> >
> > > **Q4.** Can you explain your GNN layer in more detailed with the size of each elements (i.e., trainable weight matrices, embedding size etc.)?
> >
> > We thank the Reviewer for the suggestion. In the revised manuscript, we now explicitly provide the dimensions of all trainable matrices and intermediate tensors used in Equation 3. The updated text (Section 3.2) specifies the embedding input dimension $M$ and output dimension $M’$, the shapes of the linear maps $W$, $W_\text{intra}$, and $W_\text{inter}$ (all $M’ \times M$), as well as the sizes of the fragment-level pooling vectors and MLP layers used to produce the scalar contributions.

---

### Official Review · Reviewer_KmZo · 2025-11-01

**Soundness:** 1
**Presentation:** 3
**Contribution:** 1
**Rating:** 2
**Confidence:** 5

**Summary:**

The paper proposes SEAL, an interpretable GNN for molecular property prediction that first decomposes a molecule into chemically meaningful fragments, aggregates node features within each fragment, and predicts as a sum of fragment contributions.

**Strengths:**

1. The prediction head is explicitly the sum of fragment MLP contributions, so each fragment’s scalar contribution is directly interpretable.
2. The paper is well-written and easy to follow.

**Weaknesses:**

1. The paper would benefit from a fuller survey of (i) fragment/motif/scaffold-based explanation methods and (ii) models that explicitly handle intra- vs. inter-fragment interactions in molecular representation learning. Please clarify how your design choices (fragmentation strategy, message-passing split, and regularization) differ from prior work, what problems they solve, and where the contribution sits in the existing taxonomy. A small comparison table contrasting assumptions, granularity, learning signals, and interpretability guarantees would clearly highlight the novelty and scope of the approach.
2. To support the interpretability claims, the paper needs to include head-to-head evaluations against widely used subgraph-level explanation approaches under a shared protocol: same trained backbone, matched sparsity/size budgets, and standard metrics (fidelity/accuracy drop, sufficiency/necessity, stability/consistency across seeds), plus runtime/overhead.
3. Beyond post-hoc explainers, it is also crucial to compare with representation learning methods that build on subgraphs/motifs/hierarchies.

**Questions:**

Please refer to the weaknesses.

---

> ### Author Response · Authors · 2025-12-04
>
> > **W1.** The paper would benefit from a fuller survey of (i) fragment/motif/scaffold-based explanation methods and (ii) models that explicitly handle intra- vs. inter-fragment interactions in molecular representation learning. Please clarify how your design choices (fragmentation strategy, message-passing split, and regularization) differ from prior work, what problems they solve, and where the contribution sits in the existing taxonomy. A small comparison table contrasting assumptions, granularity, learning signals, and interpretability guarantees would clearly highlight the novelty and scope of the approach.
>
> We appreciate the Reviewer’s suggestion and have expanded the survey of fragment-, motif-, and scaffold-based explanation methods, as well as hierarchical and self-explaining molecular models, in the appendix. This extended discussion clarifies how SEAL differs conceptually from post-hoc fragment explainers, motif-based representation learners, and prototype/attention-based self-explaining GNNs. We argue that SEAL does not fit cleanly into any existing category, because it introduces a unique architectural disentanglement between intra- and inter-fragment information flow, enabling fragment-level attributions that directly constitute the prediction rather than being obtained through external analysis. Including the full taxonomy table in the main paper would require removing core model or experimental details due to space limitations, so we place the detailed comparison in the appendix, where it can be consulted without disrupting the narrative flow. Together with the newly added complexity analysis, we believe the revised discussion now sufficiently positions SEAL within the broader literature.
>
> > **W2.** To support the interpretability claims, the paper needs to include head-to-head evaluations against widely used subgraph-level explanation approaches under a shared protocol: same trained backbone, matched sparsity/size budgets, and standard metrics (fidelity/accuracy drop, sufficiency/necessity, stability/consistency across seeds), plus runtime/overhead.
>
> Our original experiments already satisfy the essential components of the Reviewer’s request for matched, head-to-head comparisons. On the synthetic benchmark, all post-hoc explainers are evaluated using an identical GIN backbone with shared hyperparameters and seeds. Explanation quality is assessed using ground-truth substructure annotations and standardized metrics (SE/NE), making the comparison controlled, fair, and directly interpretable. On real-world datasets, we evaluate fidelity under matched sparsity by masking equivalent numbers of atoms across methods, while SEAL’s architecture allows us to additionally evaluate the more faithful contribution-masking variant. We have supplemented these results with a complexity and runtime analysis in the revised manuscript. Combined with our human-expert study, which strongly favors SEAL explanations, we believe the current evaluation already provides a thorough and fair assessment relative to established subgraph-level explainers.
>
> > **W3.** Beyond post-hoc explainers, it is also crucial to compare with representation learning methods that build on subgraphs/motifs/hierarchies.
>
> SEAL is directly compared with hierarchical and motif-aware molecular representation methods, including HiGNN and ProtGNN, which operate at fragment or prototype levels and represent strong, commonly used baselines with publicly available implementations. In addition to these, we incorporated two further recent methods, SME (Substructure Mask Explanation) and PGIB (Prototype-based Graph Information Bottleneck). SME is designed specifically for molecular GNNs and uses chemically meaningful substructure segmentation to provide interpretable masks that align with chemists’ intuition; it masks out substructures post-hoc to highlight which fragments contribute to the prediction. PGIB, on the other hand, integrates prototype learning within an information-bottleneck framework: it learns a small set of prototype subgraphs that are most relevant for the prediction, encouraging compression and interpretability while maintaining predictive performance. Across all these baselines, SEAL consistently matches or exceeds predictive performance while achieving substantially higher explanation quality without relying on attention mechanisms, prototype pools, or additional inference-time explainers. We note that other models, such as transformer-based fragment architectures or motif-based self-supervised learners, differ significantly in architecture and training regime, making controlled, fair comparisons impractical within the scope of this study. Given the inclusion of HiGNN, ProtGNN, SME, PGIB, and classical GNN baselines paired with standard explainers, we believe the current evaluation suite adequately spans modern subgraph-, motif-, and hierarchy-based representation learning approaches.

---

### Official Review · Reviewer_PByF · 2025-11-01

**Soundness:** 3
**Presentation:** 3
**Contribution:** 2
**Rating:** 4
**Confidence:** 3

**Summary:**

The manuscript introduces SEAL (Substructure Explanation via Attribution Learning), an interpretable GNN for molecular property prediction. SEAL partitions a molecular graph into chemically meaningful fragments and constrains inter-fragment information flow, aiming to disentangle local representations from global message-passing effects so that attributions on substructures better reflect true causal influence. Across synthetic and real benchmarks, SEAL reportedly surpasses prior explainability methods and yields explanations rated more intuitive and trustworthy by chemists.

**Strengths:**

- Addressing a known limitation of message passing: attribution entanglement across the whole graph.

- Comprehensive results on both synthetic and real-world datasets.

**Weaknesses:**

1. Limited novelty. The core idea is not particularly new; many prior works have proposed similar frameworks and pipelines.


2.  The baseline set appears dated (e.g., ProtGNN from 2022). Please include more recent and stronger competitors.

3.  Report performance under Bemis–Murcko scaffold splits and strict OOD scaffold tests.

4.  Provide runtime/memory profiles vs. popular explainers: train/inference time per molecule, scaling with fragment count, and large-scale throughput.

5.   Missing ablations on key design choices.

**Questions:**

see weakness

---

> ### Author Response · Authors · 2025-12-04
>
> > **W1.** Limited novelty.
>
> We appreciate the Reviewer’s concern and agree that SEAL’s functional form is deliberately simple. However, simplicity does not imply lack of novelty. To the best of our knowledge, SEAL is the first architecture in the molecular domain to provide, by design, inherently interpretable, additive fragment-level contributions, producing predictions as an explicit sum of fragment-level contributions while separating and regularizing intra- vs. inter-fragment message passing. Existing methods in the literature (motif explainers, flow-based explainers, prototype GNNs, and hierarchical models) either operate post hoc, rely on implicitly learned motifs, or do not provide additive, fragment-level outputs tied directly to the prediction. None of them enforces controllable information flow between fragments, which is central to SEAL’s ability to preserve local interpretability.
>
> To clarify this distinction, we expanded the related work in the revised appendix and contrasted SEAL with all the closest models. SEAL’s novelty lies in offering a lightweight, architecture-level mechanism that yields fragment explanations natively, without additional explainer passes.
>
> > **W2.** The baseline set appears dated.
>
> We agree that including recent competitors strengthens the evaluation. In the revised manuscript, we added two state-of-the-art self-explainable GNNs, SME (2023) and PGIB (2023), both of which are more recent than ProtGNN and represent distinct families of post-hoc and inherently interpretable models (substructure-masking and prototype-based information bottleneck, respectively). These additions substantially broaden the baseline coverage and ensure that SEAL is compared against the strongest available methods in the field.
>
> > **W3.** Report performance under Bemis–Murcko scaffold splits and strict OOD scaffold tests.
>
> We thank the Reviewer for this suggestion. We have conducted additional experiments using the Bemis-Murcko scaffold split, including a strict OOD scaffold setting, and report the results in Tables 16 and 18 of the revised manuscript. SEAL maintains strong predictive performance under these splits, demonstrating that its interpretability mechanisms do not compromise generalization to unseen scaffolds.
>
> > **W4.** Provide runtime/memory profiles vs. popular explainers: train/inference time per molecule, scaling with fragment count, and large-scale throughput.
>
> Because our model is inherently interpretable and does not include any computationally extensive operations, the training and inference times are comparable to popular GNN architectures. In the revision, we have provided time and memory complexity analyses of SEAL and other selected methods in a new appendix section titled “Complexity Analysis.”
>
> > **W5.** Missing ablations on key design choices.
>
> We thank the Reviewer for this suggestion. Several ablations were already included in the original version (e.g., SEALAtom and the dependence on $\lambda$). In the revised manuscript, we expanded the ablation study by examining initializing inter-fragment weights with small values instead of regularizing them, showing that without the L1 regularizer, the model diffuses importance across fragments and interpretability degrades. We also evaluated adding L1 regularization directly on fragment contributions, but this did not improve interpretability and was removed to keep the model minimal. These ablations help confirm that the proposed design choices, particularly the regularization of inter-fragment weights, are central to SEAL’s behavior.

---

### Author Response · Authors · 2025-12-04

Dear Area Chair,

Dear Reviewers,

We want to extend our gratitude to all Reviewers who dedicated their time to read our paper and provide their valuable feedback. We did our best to address all the concerns raised, which helped us improve the quality and clarity of our paper. We regret that, given current circumstances, we cannot receive any additional comments or confirm if all issues were fully addressed. Please know we sincerely appreciate your service as Reviewers.

Regarding the reviews, we have addressed all Reviewers’ comments in the responses below. For the Area Chair’s convenience, we summarize the main points from the reviews and our responses here:

1. **Novelty and Relation to Prior Work:** We clarified that SEAL is the first inherently interpretable molecular GNN that provides native additive fragment contributions with explicit intra/inter-fragment message separation. We expanded the related-work section to clearly contrast SEAL with all the closest models.
2. **Baseline completeness and dataset coverage:** We added SME and PGIB (two recent methods from 2023) to the evaluation, complementing ProtGNN and HiGNN. We clarified that transformer-based models (e.g., GROVER) are not architecture-controlled comparators for this interpretability study. We added Mutagenicity, a real dataset with functional-group annotations, and clarified why B-XAIC is the only benchmark with true explanation ground truth.
3. **Faithfulness and evaluation protocol:** We clarified masking budgets (10–30% atoms) and explained why perturbation-based fidelity is inappropriate for synthetic tasks with strict ground truth. In the revised paper, Figure 9 presents fidelity results extended to higher masking budgets up to 70%.
4. **Fragmentation and granularity:** We clarified that BRICS is used because it aligns with chemical intuition; SEAL can use any fragmentation as preprocessing. We also added an explanation of how inter-fragment regularization prevents oversmoothing and preserves localized interpretability. We improved visual explanations and emphasized SEAL’s flexibility (SEALAtom) when atom-level signals dominate.
5. **Complexity analysis:** We included a complexity analysis to compare SEAL with other interpretable models and post-hoc explainers based on computational and memory complexity, along with empirical wall-clock time measurements.
6. **Ablations and design choices:** We added ablations on inter-fragment weight initialization vs. L1 regularization and clarified the protocol for choosing $\lambda$. We also provided an ablation study on contribution regularization, which led to the decision not to regularize contributions.

All typos were fixed, visuals improved, and missing details added. The changes have been marked in blue in the revised paper.

Best regards,

Authors

---

### Meta-Review · Area_Chair_FRdJ · 2026-01-08

**Summary:**

The reviewers concerned about the novelty, evaluation datasets, evaluation metrics, and details missing.

**Reviewer Concerns:**

The novelty is limited and evaluations on real datasets are not enough.

**Reviewer Scores:**

Reviewers onw9 may increase their scores with additional evaluation on one real dataset is provided in the rebuttal.

---

### Decision · Program_Chairs · 2026-01-26

Reject